# Bayesian Deep Convolutional Networks with Many Channels are Gaussian Processes

**Roman Novak** [†], **Lechao Xiao** [†][*], **Jaehoon Lee** [‡][*] **Yasaman Bahri** [‡] **Greg Yang** [°],

**Jiri Hron** [◇], **Daniel A. Abolafia,** **Jeffrey Pennington,** **Jascha Sohl-Dickstein**

Google Brain, °Microsoft Research AI, ◇ Department of Engineering, University of Cambridge

{romann, xlc, jaehlee, yasamanb, °gregyang@microsoft.com,
◇jh2084@cam.ac.uk, danabo, jpennin, jaschasd}@google.com

## Abstract

There is a previously identified equivalence between wide fully connected neural networks (FCNs) and Gaussian processes (GPs). This equivalence enables, for instance, test set predictions that would have resulted from a fully Bayesian, infinitely wide trained FCN to be computed without ever instantiating the FCN, but by instead evaluating the corresponding GP. In this work, we derive an analogous equivalence for multi-layer convolutional neural networks (CNNs) both with and without pooling layers, and achieve state of the art results on CIFAR10 for GPs without trainable kernels. We also introduce a Monte Carlo method to estimate the GP corresponding to a given neural network architecture, even in cases where the analytic form has too many terms to be computationally feasible.

Surprisingly, in the absence of pooling layers, the GPs corresponding to CNNs with and without weight sharing are identical. As a consequence, translation equivariance, beneficial in finite channel CNNs trained with stochastic gradient descent (SGD), is guaranteed to play no role in the Bayesian treatment of the infinite channel limit – a qualitative difference between the two regimes that is not present in the FCN case. We confirm experimentally, that while in some scenarios the performance of SGD-trained finite CNNs approaches that of the corresponding GPs as the channel count increases, with careful tuning SGD-trained CNNs can significantly outperform their corresponding GPs, suggesting advantages from SGD training compared to fully Bayesian parameter estimation.

## 1 Introduction

Neural networks (NNs) demonstrate remarkable performance (He et al., 2016; Oord et al., 2016; Silver et al., 2017; Vaswani et al., 2017), but are still only poorly understood from a theoretical perspective (Goodfellow et al., 2015; Choromanska et al., 2015; Pascanu et al., 2014; Zhang et al., 2017). NN performance is often motivated in terms of model architectures, initializations, and training procedures together specifying biases, constraints, or implicit priors over the class of functions learned by a network. This induced structure in learned functions is believed to be well matched to structure inherent in many practical machine learning tasks, and in many real-world datasets. For instance, properties of NNs which are believed to make them well suited to modeling the world include: hierarchy and compositionality (Lin et al., 2017; Poggio et al., 2017), Markovian dynamics (Tiňo et al., 2004; 2007), and equivariances in time and space for RNNs (Werbos, 1988) and CNNs (Fukushima & Miyake, 1982; Rumelhart et al., 1985) respectively.

The recent discovery of an equivalence between deep neural networks and GPs (Lee et al., 2018; Matthews et al., 2018b) allow us to express an analytic form for the prior over functions encoded by deep NN architectures and initializations. This transforms an implicit prior over functions into an *explicit prior*, which can be analytically interrogated and reasoned about.[1]

Previous work studying these Neural Network-equivalent Gaussian Processes (NN-GPs) has established the correspondence only for fully connected networks (FCNs). Additionally, previous work has not used analysis of NN-GPs to gain specific insights into the equivalent NNs.

---

[*]Google AI Residents (g.co/airesidency). [†],[‡] Equal contribution.

[1]While there is broad literature on empirical interpretation of finite CNNs (Zeiler & Fergus, 2014; Simonyan et al., 2014; Long et al., 2014; Olah et al., 2017), it is commonly only applicable to fully trained networks.

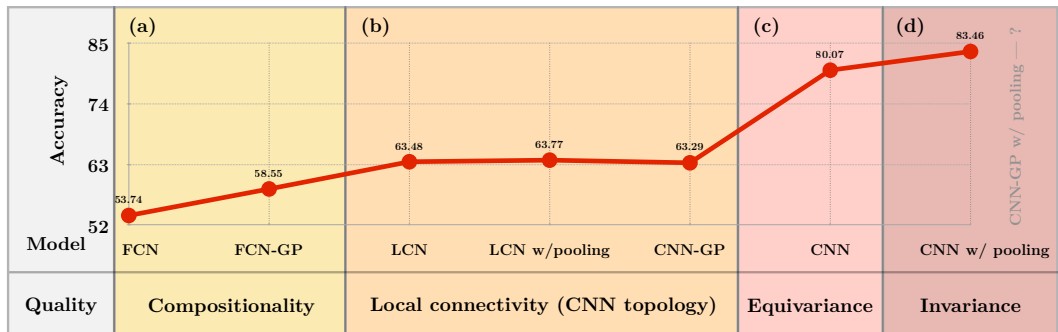

Figure 1: **Disentangling the role of network topology, equivariance, and invariance on test performance, for SGD-trained and infinitely wide Bayesian networks (NN-GPs).** Accuracy (%) on CIFAR10 of different models of the same depth, nonlinearity, and weight and bias variances. **(a)** Fully connected models – FCN (fully connected network) and FCN-GP (infinitely wide Bayesian FCN) underperform compared to **(b)** LCNs (locally connected network, a CNN without weight sharing) and CNN-GP (infinitely wide Bayesian CNN), which have a hierarchical local **topology** beneficial for image recognition. As derived in §5.1: (i) weight sharing has no effect in the Bayesian treatment of an infinite width CNN (CNN-GP performs similarly to an LCN), and (ii) pooling has no effect on generalization of an LCN model (LCN and LCN with pooling perform nearly identically). **(c)** Local connectivity combined with **equivariance** (CNN) is enabled by weight sharing in an SGD-trained finite model, allowing for a significant improvement. **(d)** Finally, **invariance** enabled by weight sharing and pooling allows for the best performance. Due to computational limitations, the performance of CNN-GPs with pooling – which also possess the property of invariance – remains an open question. Values are reported for 8-layer ReLU models. See §G.6 for experimental details, Figure 7, and Table 1 for more model comparisons.

In the present work, we extend the equivalence between NNs and NN-GPs to deep *Convolutional* Neural Networks (CNNs), both with and without pooling. CNNs are a particularly interesting architecture for study, since they are frequently held forth as a success of motivating NN design based on invariances and equivariances of the physical world (Cohen & Welling, 2016) – specifically, designing a NN to respect translation equivariance (Fukushima & Miyake, 1982; Rumelhart et al., 1985). As we will see in this work, absent pooling, this quality of equivariance has no impact in the Bayesian treatment of the infinite channel (number of convolutional filters) limit (Figure 1).

The specific novel contributions of the present work are:

1. We show analytically that CNNs with many channels, trained in a fully Bayesian fashion, correspond to an NN-GP (§2, §3). We show this for CNNs both with and without pooling, with arbitrary convolutional striding, and with both same and valid padding. We prove convergence as the number of channels in hidden layers approach infinity simultaneously (i.e. $\min\left\{n^1, \ldots, n^L\right\} \to \infty$, see §E.4 for details), extending the result of Matthews et al. (2018a) under mild conditions on the nonlinearity derivative. Our results also provide a rigorous proof of the assumption made in Xiao et al. (2018) that pre-activations in hidden layers are i.i.d. Gaussian.

2. We show that in the absence of pooling, the NN-GP for a CNN and a Locally Connected Network (LCN) are identical (Figure 1, §5.1). An LCN has the same local connectivity pattern as a CNN, but without weight sharing or translation equivariance.

3. We experimentally compare trained CNNs and LCNs and find that under certain conditions both perform similarly to the respective NN-GP (Figure 6, b, c). Moreover, both architectures tend to perform better with increased channel count, suggesting that similarly to FCNs (Neyshabur et al., 2015; Novak et al., 2018) CNNs benefit from overparameterization (Figure 6, a, b), corroborating a similar trend observed in Canziani et al. (2016, Figure 2). However, we also show that careful tuning of hyperparameters allows finite CNNs trained with SGD to outperform their corresponding NN-GPs by a significant margin. We ex-

perimentally disentangle and quantify the contributions stemming from local connectivity, equivariance, and invariance in a convolutional model in one such setting (Figure 1).

4. We introduce a Monte Carlo method to compute NN-GP kernels for situations (such as CNNs with pooling) where evaluating the NN-GP is otherwise computationally infeasible (§4).

We stress that we do not evaluate finite width Bayesian networks nor do we make any claims about their performance relative to the infinite width GP studied here or finite width SGD-trained networks. While this is an interesting subject to pursue (see Matthews et al. (2018a); Neal (1994)), it is outside of the scope of this paper.

## 1.1 RELATED WORK

In early work on neural network priors, Neal (1994) demonstrated that, in a fully connected network with a single hidden layer, certain natural priors over network *parameters* give rise to a Gaussian process prior over *functions* when the number of hidden units is taken to be infinite. Follow-up work extended the conditions under which this correspondence applied (Williams, 1997; Le Roux & Bengio, 2007; Hazan & Jaakkola, 2015). An exactly analogous correspondence for infinite width, finite depth *deep* fully connected networks was developed recently in Lee et al. (2018); Matthews et al. (2018b), with Matthews et al. (2018a) extending the convergence guarantees from ReLU to any linearly bounded nonlinearities and monotonic width growth rates. In this work we further relax the conditions to absolutely continuous nonlinearities with exponentially bounded derivative and any width growth rates.

The line of work examining signal propagation in random deep networks (Poole et al., 2016; Schoenholz et al., 2017; Yang & Schoenholz, 2017; 2018; Hanin & Rolnick, 2018; Chen et al., 2018; Yang et al., 2018) is related to the construction of the GPs we consider. They apply a mean field approximation in which the pre-activation signal is replaced with a Gaussian, and the derivation of the covariance function with depth is the same as for the kernel function of a corresponding GP. Recently, Xiao et al. (2017b; 2018) extended this to convolutional architectures without pooling. Xiao et al. (2018) also analyzed properties of the convolutional kernel at large depths to construct a *phase diagram* which will be relevant to NN-GP performance, as discussed in §B.

Compositional kernels coming from wide convolutional and fully connected layers also appeared outside of the GP context. Cho & Saul (2009) derived closed-form compositional kernels for rectified polynomial activations (including ReLU). Daniely et al. (2016) proved approximation guarantees between a network and its corresponding kernel, and show that empirical kernels will converge as the number of channels increases.

There is a line of work considering stacking of GPs, such as *deep GP*s (Lawrence & Moore, 2007; Damianou & Lawrence, 2013). These no longer correspond to GPs, though they can describe a rich class of probabilistic models beyond GPs. Alternatively, *deep kernel learning* (Wilson et al., 2016b;a; Bradshaw et al., 2017) utilizes GPs with base kernels which take in features produced by a deep neural network (often a CNN), and train the resulting model end-to-end. Finally, van der Wilk et al. (2017) incorporates convolutional structure into GP kernels, with follow-up work stacking multiple such GPs (Kumar et al., 2018; Blomqvist et al., 2018) to produce a deep convolutional GP (which is no longer a GP). Our work differs from all of these in that our GP corresponds exactly to a fully Bayesian CNN in the infinite channel limit, when all layers are taken to be of infinite size. We remark that while alternative models, such as deep GPs, do include infinite-sized layers in their construction, they do not treat all layers in this way – for instance, through insertion of bottleneck layers which are kept finite. While it remains to be seen exactly which limit is applicable for understanding realistic CNN architectures in practice, the limit we consider is natural for a large class of CNNs, namely those for which all layers sizes are large and rather comparable in size. Deep GPs, on the other hand, correspond to a potentially richer class of models, but are difficult to analytically characterize and suffer from higher inference cost.

Borovykh (2018) analyzes the convergence of CNN outputs at different spatial locations (or different timepoints for a temporal CNN) to a GP for a single input example. Thus, while they also consider a GP limit (and perform an experiment comparing posterior GP predictions to an SGD-trained CNN),

they do not address the dependence of network outputs on multiple input examples, and thus their model is unable to generate predictions on a test set consisting of new input examples.

In concurrent work, Garriga-Alonso et al. (2018) derive an NN-GP kernel equivalent to one of the kernels considered in our work. In addition to explicitly specifying kernels corresponding to pooling and vectorizing, we also compare the NN-GP performance to finite width SGD-trained CNNs and analyze the differences between the two models.

## 2 MANY-CHANNEL BAYESIAN CNNS ARE GAUSSIAN PROCESSES

### 2.1 PRELIMINARIES

**General setup.** For simplicity of presentation we consider 1D convolutional networks with circularly-padded activations (identically to Xiao et al. (2018)). Unless specified otherwise, no pooling anywhere in the network is used. If a model (NN or GP) is mentioned explicitly as "with pooling", it always corresponds to a single global average pooling layer at the top. Our analysis is straightforward to extend to higher dimensions, using zero (same) or no (valid) padding, strided convolutions, and pooling in intermediary layers (§C). We consider a series of $L + 1$ convolutional layers, $l = 0, \ldots, L$.

**Random weights and biases.** The parameters of the network are the convolutional filters and biases, $\omega_{ij,\beta}^l$ and $b_i^l$, respectively, with outgoing (incoming) channel index $i$ ($j$) and filter relative spatial location $\beta \in [\pm k] \equiv \{-k, \ldots, 0, \ldots, k\}$.[2] Assume a Gaussian prior on both the filter weights and biases,

$$\omega_{ij,\beta}^l \sim \mathcal{N}\left(0, v_\beta \frac{\sigma_\omega^2}{n^l}\right), \qquad\qquad b_i^l \sim \mathcal{N}\left(0, \sigma_b^2\right). \tag{1}$$

The weight and bias variances are $\sigma_\omega^2, \sigma_b^2$, respectively. $n^l$ is the number of channels (filters) in layer $l$, $2k + 1$ is the filter size, and $v_\beta$ is the fraction of the receptive field variance at location $\beta$ (with $\sum_\beta v_\beta = 1$). In experiments we utilize uniform $v_\beta = 1/(2k + 1)$, but nonuniform $v_\beta \neq 1/(2k + 1)$ should enable kernel properties that are better suited for ultra-deep networks, as in Xiao et al. (2018).

**Inputs, pre-activations, and activations.** Let $\mathcal{X}$ denote a set of input images (training set, validation set, or both). The network has activations $y^l(x)$ and pre-activations $z^l(x)$ for each input image $x \in \mathcal{X} \subset \mathbb{R}^{n^0 d}$, with input channel count $n^0 \in \mathbb{N}$, number of pixels $d \in \mathbb{N}$, where

$$y_{i,\alpha}^l(x) \equiv \begin{cases} x_{i,\alpha} & l = 0 \\ \phi\left(z_{i,\alpha}^{l-1}(x)\right) & l > 0 \end{cases}, \qquad z_{i,\alpha}^l(x) \equiv \sum_{j=1}^{n^l} \sum_{\beta=-k}^{k} \omega_{ij,\beta}^l y_{j,\alpha+\beta}^l(x) + b_i^l. \tag{2}$$

We emphasize the dependence of $y_{i,\alpha}^l(x)$ and $z_{i,\alpha}^l(x)$ on the input $x$. $\phi : \mathbb{R} \to \mathbb{R}$ is a nonlinearity (with elementwise application to higher-dimensional inputs). Similarly to Xiao et al. (2018), $y^l$ is assumed to be circularly-padded and the spatial size $d$ hence remains constant throughout the network in the main text (a condition relaxed in §C and Remark E.3). See Figures 2 and 3 for a visual depiction of our notation.

**Activation covariance.** A recurring quantity in this work will be the empirical uncentered covariance matrix $K^l$ of the activations $y^l$, defined as

$$\left[K^l\right]_{\alpha,\alpha'}(x, x') \equiv \frac{1}{n^l} \sum_{i=1}^{n^l} y_{i,\alpha}^l(x) y_{i,\alpha'}^l(x'). \tag{3}$$

$K^l$ is a random variable indexed by two inputs $x, x'$ and two spatial locations $\alpha, \alpha'$ (the dependence on layer widths $n^1, \ldots, n^l$, as well as weights and biases, is implied and by default not stated explicitly). $K^0$, the empirical uncentered covariance of inputs, is deterministic.

**Shapes and indexing.** Whenever an index is omitted, the variable is assumed to contain all possible entries along the respective dimension. For example, $y^0$ is a vector of size $|\mathcal{X}| n^0 d$, $\left[K^l\right]_{\alpha,\alpha'}$ is a matrix of shape $|\mathcal{X}| \times |\mathcal{X}|$, and $z_j^l$ is a vector of size $|\mathcal{X}| d$.

---

[2]We will use Roman letters to index channels and Greek letters for spatial location. We use letters $i, j, i', j'$, etc to denote channel indices, $\alpha, \alpha'$, etc to denote spatial indices and $\beta, \beta'$, etc for filter indices.

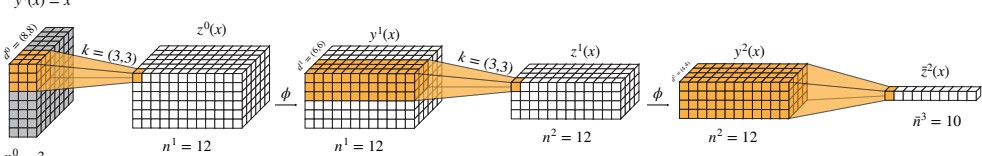

Figure 2: **A sample 2D CNN classifier annotated according to notation in §2.1, §3.** The network transforms $n^0 \times d^0 = 3 \times 8 \times 8$-dimensional inputs $y^0(x) = x \in \mathcal{X}$ into $\bar{n}^3 = 10$-dimensional logits $\bar{z}^2(x)$. Model has two convolutional layers with $k = (3, 3)$-shaped filters, nonlinearity $\phi$, and a fully connected layer at the top ($y^2(x) \rightarrow \bar{z}^2(x)$, §3.1). Hidden (pre-)activations have $n^1 = n^2 = 12$ filters. As $\min\{n^1, n^2\} \rightarrow \infty$, the prior of this CNN will approach that of a GP indexed by inputs $x$ and target class indices from 1 to $\bar{n}^3 = 10$. The covariance of such GP can be computed as $\left[\frac{\sigma_\omega^2}{6 \times 6} \sum_{\alpha=(1,1)}^{d^2} \left[(\mathcal{C} \circ \mathcal{A})^2 (K^0)\right]_{\alpha,\alpha} + \sigma_b^2\right] \otimes I_{\bar{n}^3}$, where the sum is over the $\{1, \ldots, 4\}^2$ hypercube (see §2.2, §3.1, §C). Presented is a CNN with stride 1 and no (valid) padding, i.e. the spatial shape of the input shrinks as it propagates through it $\left(d^0 = (8, 8) \rightarrow d^1 = (6, 6) \rightarrow d^2 = (4, 4)\right)$. Note that for notational simplicity 1D CNN and circular padding with $d^0 = d^1 = d^2 = d$ is assumed in the text, yet our formalism easily extends to the model displayed (§C). Further, while displayed (pre-)activations have 3D shapes, in the text we treat them as 1D vectors (§2.1).

Our work concerns proving that the top-layer pre-activations $z^L$ converge in distribution to an $|\mathcal{X}| n^{L+1}d$-variate normal random vector with a particular covariance matrix of shape $\left(|\mathcal{X}| n^{L+1}d\right) \times \left(|\mathcal{X}| n^{L+1}d\right)$ as $\min\{n^1, \ldots, n^L\} \rightarrow \infty$. We emphasize that only the channels in *hidden* layers are taken to infinity, and $n^{L+1}$, the number of channels in the top-layer pre-activations $z^L$, remains fixed. For convergence proofs, we always consider $z^l$, $y^l$, as well as any of their indexed subsets like $z_j^l$, $y_{i,\alpha}^l$ to be 1D vector random variables, while $K^l$, as well as any of its indexed subsets (when applicable, e.g. $\left[K^l\right]_{\alpha,\alpha'}$, $\left[K^l\right](x, x')$) to be 2D matrix random variables.

## 2.2 Correspondence between Gaussian processes and Bayesian deep CNNs with infinitely many channels

We next consider the prior over outputs $z^L$ computed by a CNN in the limit of infinitely many channels in the hidden (excluding input and output) layers, $\min\{n^1, \ldots, n^L\} \rightarrow \infty$, and derive its equivalence to a GP with a compositional kernel. This section outlines an argument showing that $z^L$ is normally distributed conditioned on previous layer activation covariance $K^L$, which itself becomes deterministic in the infinite limit. This allows to conclude convergence in distribution of the outputs to a Gaussian with the respective deterministic covariance limit. This section omits many technical details elaborated in §E.4.

### 2.2.1 A single convolutional layer is a GP conditioned on the uncentered covariance matrix of the previous layer's activations

As can be seen in Equation 2, the pre-activations $z^l$ are a linear transformation of the multivariate Gaussian $\{\omega^l, b^l\}$, specified by the previous layer's activations $y^l$. A linear transformation of a multivariate Gaussian is itself a Gaussian with a covariance matrix that can be derived straightforwardly. Specifically,

$$\left(z^l | y^l\right) \sim \mathcal{N}\left(0, \mathcal{A}\left(K^l\right) \otimes I_{n^{l+1}}\right), \tag{4}$$

where $I_{n^{l+1}}$ is an $n^{l+1} \times n^{l+1}$ identity matrix, and $\mathcal{A}\left(K^l\right)$ is the covariance of the pre-activations $z_i^l$ and is derived in Xiao et al. (2018). Precisely, $\mathcal{A} : \text{PSD}_{|\mathcal{X}|d} \rightarrow \text{PSD}_{|\mathcal{X}|d}$ is an affine transformation (a cross-correlation operator followed by a shifting operator) on the space of positive semi-definite $|\mathcal{X}|d \times |\mathcal{X}|d$ matrices defined as follows:

$$[\mathcal{A}(K)]_{\alpha,\alpha'}(x, x') \equiv \sigma_b^2 + \sigma_\omega^2 \sum_\beta v_\beta [K]_{\alpha+\beta,\alpha'+\beta}(x, x'). \tag{5}$$

$\mathcal{A}$ preserves positive semi-definiteness due to Equation 4. Notice that the covariance matrix in Equation 4 is block diagonal due to the fact that separate channels $\left\{z_i^l\right\}_{i=1}^{n^{l+1}}$ are i.i.d. conditioned on $y^l$, due to i.i.d. weights and biases $\left\{\omega_i^l, b_i^l\right\}_{i=1}^{n^{l+1}}$.

We further remark that per Equation 4 the normal distribution of $\left(z^l | y^l\right)$ only depends on $K^l$, hence the random variable $\left(z^l | K^l\right)$ has the same distribution by the law of total expectation:

$$\left(z^l | K^l\right) \sim \mathcal{N}\left(0, \mathcal{A}\left(K^l\right) \otimes I_{n^{l+1}}\right). \tag{6}$$

### 2.2.2 Activation covariance matrix becomes deterministic with increasing channel count

It follows from Equation 6 that the summands in Equation 3 are i.i.d. conditioned on *fixed* $K^{l-1}$. Subject to weak restrictions on the nonlinearity $\phi$, we can apply the weak law of large numbers and conclude that the covariance matrix $K^l$ becomes deterministic in the infinite channel limit in layer $l$ (note that pre-activations $z^l$ remain stochastic). Precisely,

$$\forall K^{l-1} \in \text{PSD}_{|\mathcal{X}|d} \quad \left(K^l | K^{l-1}\right) \xrightarrow[n^l \to \infty]{P^3} (\mathcal{C} \circ \mathcal{A})\left(K^{l-1}\right) \quad \text{(in probability)}, \tag{7}$$

where $\mathcal{C}$ is defined for any $|\mathcal{X}|d \times |\mathcal{X}|d$ PSD matrix $K$ as

$$[\mathcal{C}(K)]_{\alpha, \alpha'}(x, x') \equiv \mathbb{E}_{u \sim \mathcal{N}(0,K)}\left[\phi\left(u_\alpha(x)\right) \phi\left(u_{\alpha'}(x')\right)\right]. \tag{8}$$

The decoupling of the kernel "propagation" into $\mathcal{C}$ and $\mathcal{A}$ is highly convenient since $\mathcal{A}$ is a simple affine transformation of the kernel (see Equation 5), and $\mathcal{C}$ is a well-studied map in literature (see §G.4), and for nonlinearities such as ReLU (Nair & Hinton, 2010) and the error function (erf) $\mathcal{C}$ can be computed in closed form as derived in Cho & Saul (2009) and Williams (1997) respectively. We refer the reader to Xiao et al. (2018, Lemma A.1) for complete derivation of the limiting value in Equation 7.

A less obvious result is that, under slightly stronger assumptions on $\phi$, the top-layer activation covariance $K^L$ becomes *unconditionally* (dependence on observed deterministic inputs $y^0$ is implied) deterministic as channels in all hidden layers grow to infinity simultaneously:

$$K^L \xrightarrow[\min\{n^1,\ldots,n^L\} \to \infty]{P} K_\infty^L \equiv (\mathcal{C} \circ \mathcal{A})^L\left(K^0\right), \tag{9}$$

i.e. $K_\infty^L$ is $(\mathcal{C} \circ \mathcal{A})$ applied $L$ times to $K^0$, the deterministic input covariance. We prove this in §E.4 (Theorem E.5). See Figure 3 for a depiction of the correspondence between neural networks and their infinite width limit covariances $K_\infty^l$.

### 2.2.3 A conditionally normal random variable becomes normal if its covariance becomes deterministic

§2.2.1 established that $\left(z^L | K^L\right)$ is Gaussian, and §2.2.2 established that its covariance matrix $\mathcal{A}\left(K^L\right) \otimes I_{n^{L+1}}$ converges in probability to a deterministic $\mathcal{A}\left(K_\infty^L\right) \otimes I_{n^{l+1}}$ in the infinite channel limit (since $K^L \xrightarrow{P} K_\infty^L$, and $\mathcal{A}(\cdot) \otimes I_{n^{L+1}} : \mathbb{R}^{|\mathcal{X}|d \times |\mathcal{X}|d} \to \mathbb{R}^{n^{l+1}|\mathcal{X}|d \times n^{l+1}|\mathcal{X}|d}$ is continuous). As we establish in §E.4 (Theorem E.6), this is sufficient to conclude with the following result.

**Result.** If $\phi : \mathbb{R} \to \mathbb{R}$ is absolutely continuous and has an exponentially bounded derivative, i.e. $\exists a, b \in \mathbb{R} : |\phi'(x)| \leq a \exp(bx)$ a.e. (almost everywhere), then the following convergence in distribution holds:

$$\left(z^L | y^0\right) \xrightarrow[\min\{n^1,\ldots,n^L\} \to \infty]{\mathcal{D}} \mathcal{N}\left(0, \mathcal{A}\left(K_\infty^L\right) \otimes I_{n^{L+1}}\right). \tag{10}$$

For more intuition behind Equation 10 and an informal proof please consult §E.3.

---

[3] The weak law of large numbers allows convergence in probability of individual entries of $K^l$. However, due to the finite dimensionality of $K^l$, joint convergence in probability follows.

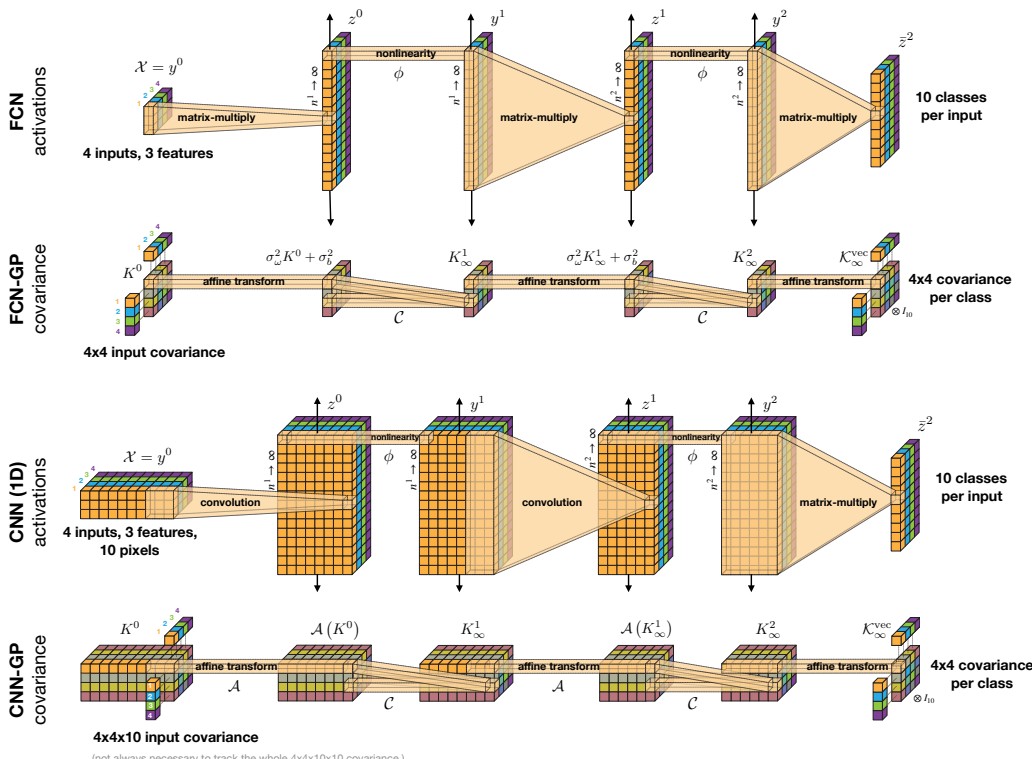

Figure 3: **Visualization of sample finite neural networks and their infinitely wide Bayesian counterparts (NN-GPs).** Presented are networks with nonlinearity $\phi$, $L = 2$ hidden layers that regress $\bar{n}^3 = 10$-dimensional outputs $\bar{z}^2(x)$ for each of the 4 (**1**, **2**, **3**, **4**) inputs $x$ from the dataset $\mathcal{X}$. A hierarchical computation performed by a NN corresponds to a hierarchical transformation of the respective covariance matrix (resulting in $\mathcal{K}_\infty^{\text{vec}} \otimes I_{10}$). **Top two rows:** a fully connected network (FCN, above) and the respective FCN-GP (below). **Bottom two rows:** a 1D CNN with no (valid) padding, and the respective CNN-GP. The analogy between the two models (NN and GP) is qualitatively similar to the case of FCN, modulo additional spatial dimensions in the internal activations and the respective additional entries in intermediary covariance matrices. Note that in general $d = 10$ pixels would induce a $10 \times 10$-fold increase in the number of covariance entries (in $K^0, K_\infty^1, \dots$) compared to the respective FCN, yet without pooling only a small fraction of them (displayed) needs be computed to obtain the top-layer GP covariance $\mathcal{K}_\infty^{\text{vec}} \otimes I_{10}$ (see §3.1).

## 3  TRANSFORMING A GP OVER SPATIAL LOCATIONS INTO A GP OVER CLASSES

In §2.2 we showed that in the infinite channel limit a deep CNN is a GP indexed by input samples, output spatial locations, and output channel indices. Further, its covariance matrix $K_\infty^L$ can be computed in closed form. Here we show that transformations to obtain class predictions that are common in CNN classifiers can be represented as either *vectorization* or *projection* (as long as we treat classification as regression (see §G), identically to Lee et al. (2018)). Both of these operations preserve the GP equivalence and allow the computation of the covariance matrix of the respective GP (now indexed by input samples and target classes) as a simple transformation of $K_\infty^L$.

In the following we will denote $\bar{n}^{L+2}$ as the number of target classes, $\bar{z}^{L+1}(x) \in \mathbb{R}^{\bar{n}^{L+2}}$ as the outputs of the CNN-based classifier, and $\bar{\omega}^{L+1}, \bar{b}^{L+1}$ as the last (number $L + 1$) layer weights and biases. For each class $i$ we will denote the empirical uncentered covariance of the outputs as

$$\mathcal{K}_i \equiv \left(\bar{z}_i^{L+1}\right)\left(\bar{z}_i^{L+1}\right)^T, \tag{11}$$

a random variable of shape $|\mathcal{X}| \times |\mathcal{X}|$. We will show that in the scenarios considered below in §3.1 and §3.2, the outputs $\left(\bar{z}_i^{L+1}|y^0\right)$ will converge in distribution to a multivariate Gaussian (i.i.d.

for every class $i$) as $\min\{n^1, \ldots, n^{L+1}\} \to \infty$ with covariance denoted as $\mathcal{K}_\infty \in \mathbb{R}^{|\mathcal{X}| \times |\mathcal{X}|}$, or, jointly, $\left(\bar{z}^{L+1} | y^0\right) \xrightarrow{\mathcal{D}} \mathcal{N}\left(0, \mathcal{K}_\infty \otimes I_{\bar{n}^{L+2}}\right)$. As in §2.2, we postpone the formal derivation until §E.4 (Theorem E.6).

## 3.1 Vectorization

One common readout strategy is to vectorize (flatten) the output of the last convolutional layer $z^L(x) \in \mathbb{R}^{n^{L+1}d}$ into a vector[4] $\text{vec}\left[z^L(x)\right] \in \mathbb{R}^{n^{L+1}d}$ and stack a fully connected layer on top:

$$\bar{z}_i^{L+1}(x) \equiv \sum_{j=1}^{n^{L+1}d} \bar{\omega}_{ij}^{L+1} \phi\left(\text{vec}\left[z^L(x)\right]\right)_j + \bar{b}_i^{L+1}, \tag{12}$$

where the weights $\bar{\omega}^{L+1} \in \mathbb{R}^{\bar{n}^{L+2} \times n^{L+1}d}$ and biases $\bar{b}^{L+1} \in \mathbb{R}^{\bar{n}^{L+2}}$ are i.i.d. Gaussian, $\bar{\omega}_{ij}^{L+1} \sim \mathcal{N}\left(0, \sigma_\omega^2/(n^{L+1}d)\right), \bar{b}_i^{L+1} \sim \mathcal{N}\left(0, \sigma_b^2\right)$.

It is easy to verify that $\left(\bar{z}_i^{L+1} | K^{L+1}\right)$ is a multivariate Gaussian with covariance:

$$\mathbb{E}\left[\mathcal{K}_i^{\text{vec}} | K^{L+1}\right] = \mathbb{E}\left[\left(\bar{z}_i^{L+1}\right)\left(\bar{z}_i^{L+1}\right)^T | K^{L+1}\right] = \frac{\sigma_\omega^2}{d} \sum_\alpha \left[K^{L+1}\right]_{\alpha,\alpha} + \sigma_b^2. \tag{13}$$

The argument of §2.2 can then be extended to conclude that in the limit of infinite width $\left(\bar{z}_i^{L+1} | y^0\right)$ converges in distribution to a multivariate Gaussian (i.i.d. for each class $i$) with covariance

$$\mathcal{K}_\infty^{\text{vec}} = \frac{\sigma_\omega^2}{d} \sum_\alpha \left[K_\infty^{L+1}\right]_{\alpha,\alpha} + \sigma_b^2. \tag{14}$$

A sample 2D CNN using this readout strategy is depicted in Figure 2, and a sample correspondence between a FCN, FCN-GP, CNN, and CNN-GP is depicted in Figure 3.

Note that as observed in Xiao et al. (2018), to compute the summands $\left[K_\infty^{L+1}\right]_{\alpha,\alpha}(x, x')$ in Equation 13, one needs only the corresponding terms $\left[K_\infty^L\right]_{\alpha,\alpha}(x, x')$. Consequently, we only need to compute $\left\{\left[K_\infty^l\right]_{\alpha,\alpha}(x, x') : x, x' \in \mathcal{X}, \alpha \in \{1 \ldots d\}\right\}_{l=0,\ldots,L}$ and the memory cost is $\mathcal{O}\left(|\mathcal{X}|^2 d\right)$ (or $\mathcal{O}(d)$ per covariance entry in an iterative or distributed setting). Note that this approach ignores pixel-pixel covariances and produces a GP corresponding to a locally connected network (see §5.1).

## 3.2 Projection

Another readout approach is a projection to collapse the spatial dimensions. Let $h \in \mathbb{R}^d$ be a deterministic projection vector, $\bar{\omega}_{ij}^{L+1} \sim \mathcal{N}\left(0, \sigma_\omega^2/n^{L+1}\right)$, and $\bar{b}^{L+1}$ be the same as in §3.1.

Define the output to be

$$\bar{z}_i^{L+1}(x) \equiv \sum_{j=1}^{n^{L+1}} \bar{\omega}_{ij}^{L+1}\left(\sum_{\alpha=1}^d \phi\left(z^L(x)\right)_{j,\alpha} h_\alpha\right) + \bar{b}_i^{L+1}, \quad \text{leading to (analogously to §3.1)} \tag{15}$$

$$\mathcal{K}_\infty^h \equiv \sigma_\omega^2 \sum_{\alpha,\alpha'} h_\alpha h_{\alpha'} \left[K_\infty^{L+1}\right]_{\alpha,\alpha'} + \sigma_b^2. \tag{16}$$

Examples of this approach include

1. **Global average pooling:** take $h = \frac{1}{d}\mathbf{1_d}$. Then

$$\mathcal{K}_\infty^{\text{pool}} \equiv \frac{\sigma_\omega^2}{d^2} \sum_{\alpha,\alpha'} \left[K_\infty^{L+1}\right]_{\alpha,\alpha'} + \sigma_b^2. \tag{17}$$

---

[4]Note that since per our notation described in §2.1 $z^L(x)$ is already considered a 1D vector, here this operation simply amounts to re-indexing $z^L(x)$ with one channel index $j$ instead of two (channel $i$ and pixel $\alpha$).

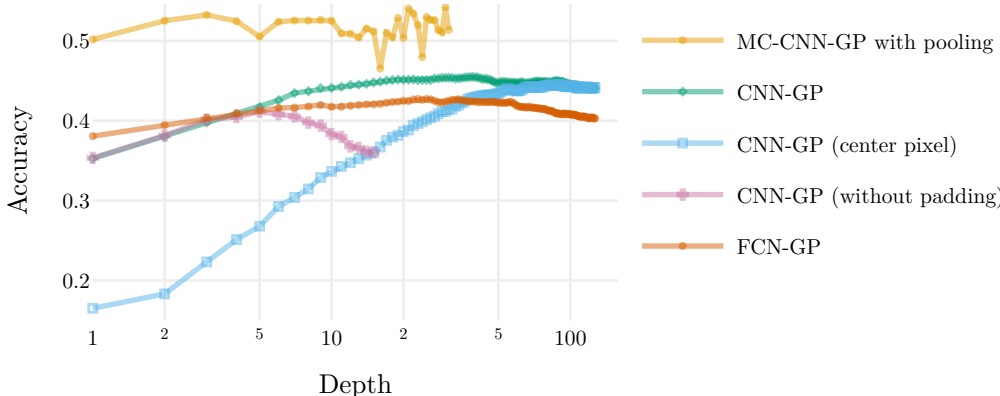

Figure 4: **Different dimensionality collapsing strategies described in §3.** Validation accuracy of an **MC-CNN-GP with pooling** (§3.2.1) is consistently better than other models due to translation invariance of the kernel. **CNN-GP with zero padding** (§3.1) outperforms an analogous **CNN-GP without padding** as depth increases. At depth 15 the spatial dimension of the output without padding is reduced to $1 \times 1$, making the **CNN-GP without padding** equivalent to the **center pixel selection strategy** (§3.2.2) – which also performs worse than the **CNN-GP** (we conjecture, due to overfitting to centrally-located features) but approaches the latter (right) in the limit of large depth, as information becomes more uniformly spatially distributed (Xiao et al., 2018). **CNN-GPs** generally outperform **FCN-GP**, presumably due to the local connectivity prior, but can fail to capture nonlinear interactions between spatially-distant pixels at shallow depths (left). Values are reported on a 2K/4K train/validation subset of CIFAR10. See §G.3 for experimental details.

This approach corresponds to applying global average pooling right after the last convolutional layer.[5] This approach takes all pixel-pixel covariances into consideration and makes the kernel translation invariant. However, it requires $\mathcal{O}\left(|\mathcal{X}|^2 d^2\right)$ memory to compute the sample-sample covariance of the GP (or $\mathcal{O}\left(d^2\right)$ per covariance entry in an iterative or distributed setting). It is impractical to use this method to analytically evaluate the GP, and we propose to use a Monte Carlo approach (see §4).

2. **Subsampling one particular pixel:** take $h = e_\alpha$,

$$\mathcal{K}^{e_\alpha}_\infty \equiv \sigma_\omega^2 \left[K^{L+1}_\infty\right]_{\alpha,\alpha} + \sigma_b^2. \tag{18}$$

This approach makes use of only one pixel-pixel covariance, and requires the same amount of memory as vectorization (§3.1) to compute.

We compare the performance of presented strategies in Figure 4. Note that all described strategies admit stacking additional FC layers on top while retaining the GP equivalence, using a derivation analogous to §2.2 (Lee et al., 2018; Matthews et al., 2018b).

## 4  MONTE CARLO EVALUATION OF INTRACTABLE GP KERNELS

We introduce a Monte Carlo estimation method for NN-GP kernels which are computationally impractical to compute analytically, or for which we do not know the analytic form. Similar in spirit to traditional random feature methods (Rahimi & Recht, 2007), the core idea is to instantiate many random *finite* width networks and use the empirical uncentered covariances of activations to estimate the Monte Carlo-GP (MC-GP) kernel,

$$\left[K^l_{n,M}\right]_{\alpha,\alpha'}(x, x') \equiv \frac{1}{Mn} \sum_{m=1}^{M} \sum_{c=1}^{n} y^l_{c\alpha}(x; \theta_m)\, y^l_{c\alpha'}(x'; \theta_m) \tag{19}$$

---

[5] Spatially local average pooling in intermediary layers can be constructed in a similar fashion (§C). We focus on global average pooling in this work to more effectively isolate the effects of pooling from other aspects of the model like local connectivity or equivariance.

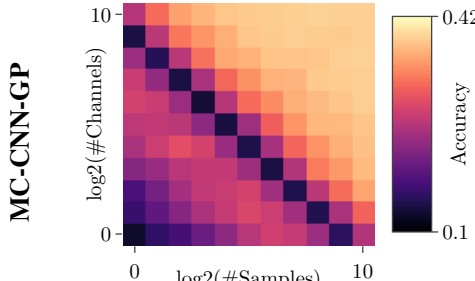 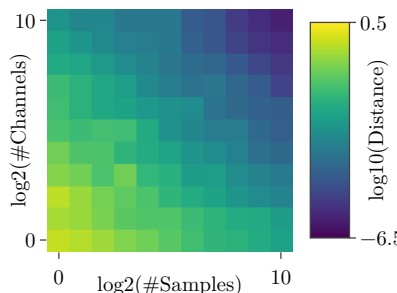

Figure 5: **Convergence of NN-GP Monte Carlo estimates.** **Validation accuracy** (left) of an MC-CNN-GP increases with $n \times M$ (channel count $\times$ number of samples) and approaches that of the exact CNN-GP (not shown), while the **distance** (right) to the exact kernel decreases. The dark band in the left plot corresponds to ill-conditioning of $K_{n,M}^{L+1}$ when the number of outer products contributing to $K_{n,M}^{L+1}$ approximately equals its rank. Values reported are for a 3-layer model applied to a 2K/4K train/validation subset of CIFAR10 downsampled to $8 \times 8$. See Figure 10 for similar results with other architectures and §G.2 for experimental details.

where $\theta$ consists of $M$ draws of the weights and biases from their prior distribution, $\theta_m \sim p(\theta)$, and $n$ is the width or number of channels in hidden layers. The MC-GP kernel converges to the analytic kernel with increasing width, $\lim_{n \to \infty} K_{n,M}^l = K_\infty^l$ in probability.

For finite width networks, the uncertainty in $K_{n,M}^l$ is $\mathrm{Var}\left[K_{n,M}^l\right] = \mathrm{Var}_\theta\left[K_n^l(\theta)\right]/M$. From Daniely et al. (2016, Theorems 2, 3), we know that for well-behaved nonlinearities (that include ReLU and erf considered in our experiments) $\mathrm{Var}_\theta\left[K_n^l(\theta)\right] \propto \frac{1}{n}$, which leads to $\mathrm{Var}_\theta[K_{n,M}^l] \propto \frac{1}{Mn}$. For finite $n$, $K_{n,M}^l$ is also a biased estimate of $K_\infty^l$, where the bias depends solely on network width. We do not currently have an analytic form for this bias, but we can see in Figures 5 and 10 that for the hyperparameters we probe it is small relative to the variance. In particular, $\left\|K_{n,M}^l(\theta) - K_\infty^L\right\|_F^2$ is nearly constant for constant $Mn$. We thus treat $Mn$ as the effective sample size for the Monte Carlo kernel estimate. Increasing $M$ and reducing $n$ can reduce memory cost, though potentially at the expense of increased compute time and bias.

In a non-distributed setting, the MC-GP reduces the memory requirements to compute $\mathcal{GP}^{\mathrm{pool}}$ from $\mathcal{O}\left(|\mathcal{X}|^2 d^2\right)$ to $\mathcal{O}\left(|\mathcal{X}|^2 + n^2 + nd\right)$, making the evaluation of CNN-GPs with pooling practical.

## 5 DISCUSSION

### 5.1 BAYESIAN CNNS WITH MANY CHANNELS ARE IDENTICAL TO LOCALLY CONNECTED NETWORKS, IN THE ABSENCE OF POOLING

Locally Connected Networks (LCNs) (Fukushima, 1975; Lecun, 1989) are CNNs without weight sharing between spatial locations. LCNs preserve the connectivity pattern, and thus topology, of a CNN. However, they do not possess the equivariance property of a CNN – if an input is translated, the latent representation in an LCN will be completely different, rather than also being translated.

The CNN-GP predictions without spatial pooling in §3.1 and §3.2.2 depend only on sample-sample covariances, and do not depend on pixel-pixel covariances. LCNs destroy pixel-pixel covariances: $\left[K_\infty^L\right]_{\alpha,\alpha'}^{\mathrm{LCN}}(x,x') = 0$, for $\alpha \neq \alpha'$ and all $x,x' \in \mathcal{X}$ and $L > 0$. However, LCNs preserve the covariances between input examples at every pixel: $\left[K_\infty^L\right]_{\alpha,\alpha}^{\mathrm{LCN}}(x,x') = \left[K_\infty^L\right]_{\alpha,\alpha}^{\mathrm{CNN}}(x,x')$. As a result, in the absence of pooling, LCN-GPs and CNN-GPs are identical. Moreover, LCN-GPs with pooling are identical to CNN-GPs with vectorization of the top layer (under suitable scaling of $y^{L+1}$). We confirm these findings experimentally in trained networks in the limit of large width in Figures 1 and 6 (b), as well as by demonstrating convergence of MC-GPs of the respective architectures to the same CNN-GP (modulo scaling of $y^{L+1}$) in Figures 5 and 10.

## 5.2 POOLING LEVERAGES EQUIVARIANCE TO PROVIDE INVARIANCE

The only kernel leveraging pixel-pixel covariances is that of the CNN-GP with pooling. This enables the predictions of this GP and the corresponding CNN to be invariant to translations (modulo edge effects) – a beneficial quality for an image classifier. We observe strong experimental evidence supporting the benefits of invariance throughout this work (Figures 1, 4, 5, 6 (b); Table 1), in both CNNs and CNN-GPs.

## 5.3 FINITE CHANNEL SGD-TRAINED CNNS CAN OUTPERFORM INFINITE CHANNEL BAYESIAN CNNS, IN THE ABSENCE OF POOLING

In the absence of pooling, the benefits of equivariance and weight sharing are more challenging to explain in terms of Bayesian priors on class predictions (since without pooling equivariance is not a property of the outputs, but only of intermediary representations). Indeed, in this work we find that the performance of finite width SGD-trained CNNs often approaches that of their CNN-GP counterpart (Figure 6, b, c),[6] suggesting that in those cases equivariance does not play a beneficial role in SGD-trained networks.

However, as can be seen in Figures 1, 6 (c), 7, and Table 1, the best CNN *overall* outperforms the best CNN-GP by a significant margin – an observation specific to CNNs and not FCNs or LCNs.[7] We observe this gap in performance especially in the case of $\mathrm{ReLU}$ networks trained with a large learning rate. In Figure 1 we demonstrate this large gap in performance by evaluating different models with equivalent architecure and hyperparameter settings, chosen for good SGD-trained CNN performance.

We conjecture that equivariance, a property lacking in LCNs and the Bayesian treatment of the infinite channel CNN limit, contributes to the performance of SGD-trained finite channel CNNs with the correct settings of hyperparameters. Nonetheless, more work is needed to disentangle and quantify the separate contributions of stochastic optimization and finite width effects,[8] and differences in performance between CNNs with weight sharing and their corresponding CNN-GPs.

## 6 CONCLUSION

In this work we have derived a Gaussian process that corresponds to fully Bayesian multi-layer CNNs with infinitely many channels. The covariance of this GP can be efficiently computed either in closed form or by using Monte Carlo sampling, depending on the architecture.

The CNN-GP achieves state of the art results for GPs without trainable kernels on CIFAR10. It can perform competitively with CNNs (that fit the training set) of equivalent architecture and weight priors, which makes it an appealing choice for small datasets, as it eliminates all training-related hyperparameters. However, we found that the best *overall* performance, at least in the absence of pooling, is achieved by finite SGD-trained CNNs and not by their infinite Bayesian counterparts. We hope our work stimulates future research towards understanding the distribution over functions induced by model architecture and training approach, and what aspects of this distribution are important for model performance.

Another natural extension of our work is the study of other deep learning architectures in the infinitely wide limit. After the publication of this paper, Yang (2019) devised a unifying framework proving the GP convergence for even more models (such as ones using batch normalization, (self-)attention, LSTM) with slightly different assumptions on the nonlinearity.

---

[6]This observation is conditioned on the respective NN fitting the training set to $100\%$. Underfitting breaks the correspondance to an NN-GP, since train set predictions of such a network no longer correspond to the true training labels. Properly tuned underfitting often also leads to better generalization (Table 1).

[7]Performing an analogous large-dataset comparison between CNNs and CNN-GPs *with pooling* was not computationally feasible. Their relative performance remains an interesting open question for future research.

[8]We remark that concerns about GPs not being able to learn hierarchical representations have been raised in the literature (Matthews et al., 2018a, Section 7), (Neal, 1995, Chapter 5), (MacKay, 2003, Section 45.7) However, practical impact of these assertions have not been extensively investigated empirically or theoretically, and we hope that our work stimulates research in this direction.

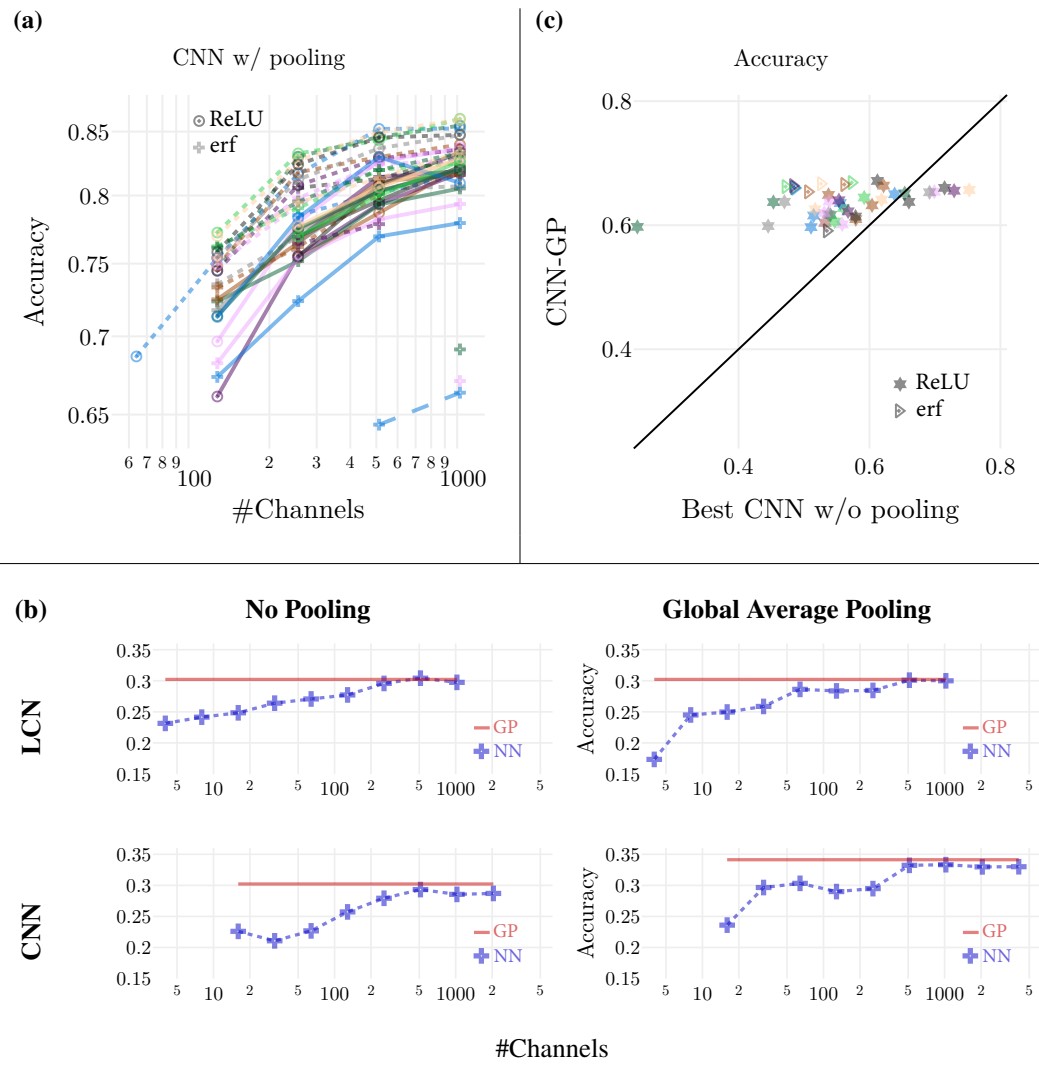

Figure 6: **(a): SGD-trained CNNs often perform better with increasing number of channels.** Each line corresponds to a particular choice of architecture and initialization hyperparameters, with best learning rate and weight decay selected independently for each number of channels ($x$-axis). **(b): SGD-trained CNNs often approach the performance of their corresponding CNN-GP with increasing number of channels.** All models have the same architecture except for pooling and weight sharing, as well as training-related hyperparameters such as learning rate, weight decay and batch size, which are selected for each number of channels ($x$-axis) to maximize validation performance ($y$-axis) of a neural network. As the number of channels grows, best validation accuracy increases and approaches accuracy of the respective GP (solid horizontal line). **(c): However, the best-performing SGD-trained CNNs can outperform their corresponding CNN-GPs.** Each point corresponds to the test accuracy of: ($y$-axis) a specific CNN-GP; ($x$-axis) the best (on validation) CNN with the same architectural hyper-parameters selected among the 100%-accurate models on the full training CIFAR10 dataset with different learning rates, weight decay and number of channels. While CNN-GP appears competitive against 100%-accurate CNNs (above the diagonal), the best CNNs *overall* outperform CNN-GPs by a significant margin (below the diagonal, right). For further analysis of factors leading to similar or diverging behavior between SGD-trained finite CNNs and infinite Bayesian CNNs see Figures 1, 7, and Table 1. **Experimental details:** all networks have reached 100% training accuracy on CIFAR10. Values in (b) are reported on an 0.5K/4K train/validation subset downsampled to $8 \times 8$ for computational reasons. See §G.5 and §G.1 for full experimental details of (a, c) and (b) plots respectively.

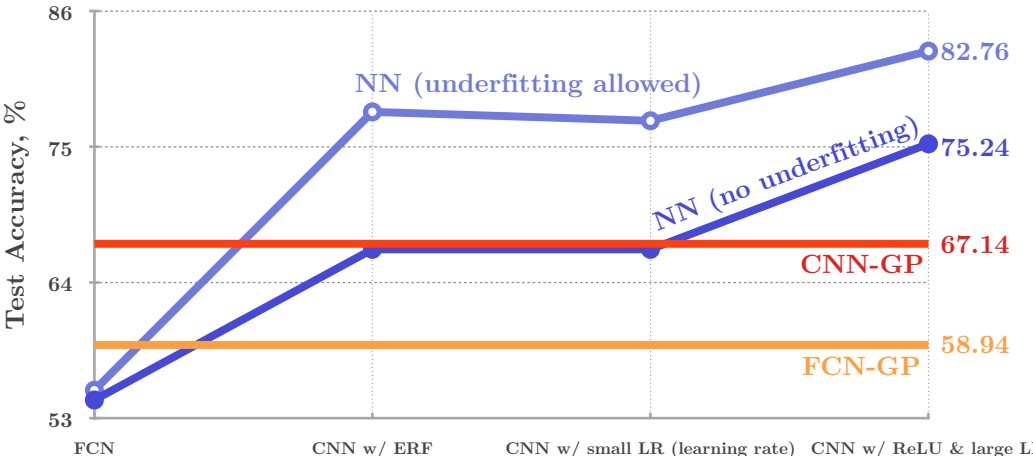

Figure 7: **Aspects of architecture and inference influencing test performance.** Test accuracy (vertical axis, %) for the best model within each model family (horizontal axis), maximizing validation accuracy over depth, width, and training and initialization hyperpameters ("No underfitting" means only models that achieved 100% training accuracy were considered). CNN-GP outperforms SGD-trained models optimized with a small learning rate to 100% train accuracy. When SGD optimization is allowed to underfit the training set, there is a significant improvement in generalization. Further, when ReLU nonlinearities are paired with large learning rates, the performance of SGD-trained models again improves relative to CNN-GPs, suggesting a beneficial interplay between ReLUs and fast SGD training. These differences in performance between CNNs and CNN-GPs are not observed between FCNs and FCN-GPs, or between LCNs and LCN-GPs (Figure 1), suggesting that *equivariance* is the underlying factor responsible for the improved performance of finite SGD-trained CNNs relative to infinite Bayesian CNNs without pooling. Further comparison between SGD-trained CNNs and CNN-GPs *with pooling* (omitted due to computational limitations), where both models do share the property of invariance, would be an interesting direction for future research. See Table 1 for further results on other datasets, as well as comparison to GP performance in prior literature. See §G.5 for experimental details.

| Model | CIFAR10 | MNIST | Fashion-MNIST |
|---|---|---|---|
| CNN with pooling | **14.85 (15.65)** | – | – |
| CNN with ReLU and large learning rate | 24.76 (17.64) | – | – |
| CNN-GP | 32.86 | 0.88 | **7.40** |
| CNN with small learning rate | 33.31(22.89) | – | – |
| CNN with erf (any learning rate) | 33.31(22.17) | – | – |
| Convolutional GP (van der Wilk et al., 2017) | 35.40 | 1.17 | – |
| ResNet GP (Garriga-Alonso et al., 2018) | – | **0.84** | – |
| Residual CNN-GP (Garriga-Alonso et al., 2018) | – | 0.96 | – |
| CNN-GP (Garriga-Alonso et al., 2018) | – | 1.03 | – |
| FCN-GP | 41.06 | 1.22 | 8.22 |
| FCN-GP (Lee et al., 2018) | 44.34 | 1.21 | – |
| FCN | 45.52 (44.73) | – | – |

Table 1: **Best achieved test error for different model classes (vertical axis) and datasets (horizontal axis).** Test error (%) for the best model within each model family, maximizing validation accuracy over depth, width, and training and initialization hyperpameters. Except where indicated by parentheses, all models achieve 100% training accuracy. For SGD-trained CNNs, numbers in parentheses correspond to the same model family, but without restriction on training accuracy. CNN-GP achieves state of the art results on CIFAR10 for GPs without trainable kernels and outperforms SGD-trained models optimized with a small learning rate to 100% train accuracy. See Figure 7 for visualization and interpretation of these results. See §G.5 for experimental details.

# 7 ACKNOWLEDGEMENTS

We thank Sam Schoenholz, Vinay Rao, Daniel Freeman, Qiang Zeng, and Phil Long for frequent discussion and feedback on preliminary results.

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

# Appendices

## A    ADDITIONAL FIGURES

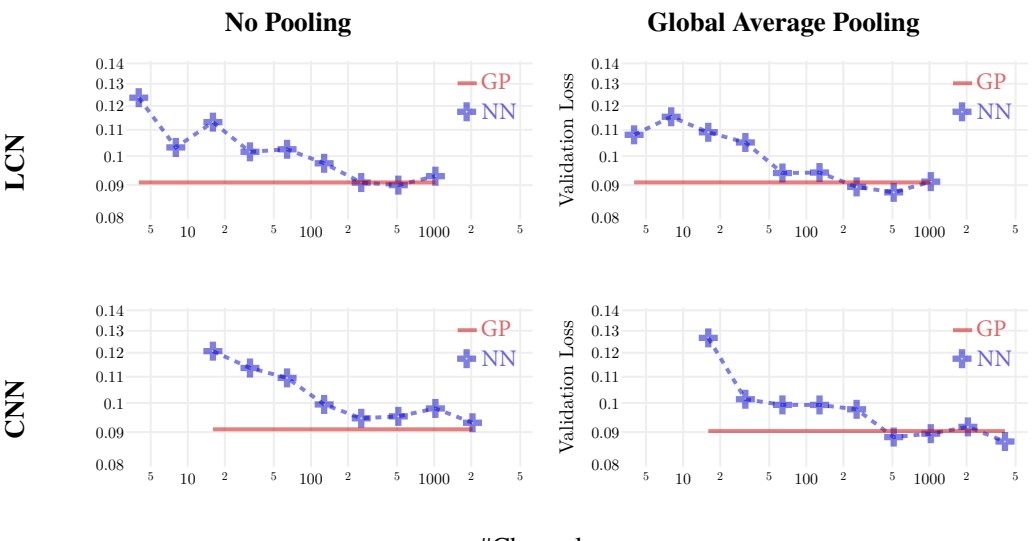

Figure 8: **Validation loss convergence.** Best validation loss (vertical axis) of **trained neural networks** (dashed line) as the number of channels increases (horizontal axis) approaches that of a respective **(MC-)CNN-GP** (solid horizontal line). See Figure 6 (b) for validation accuracy, Figure 9 for training loss and §G.1 for experimental details.

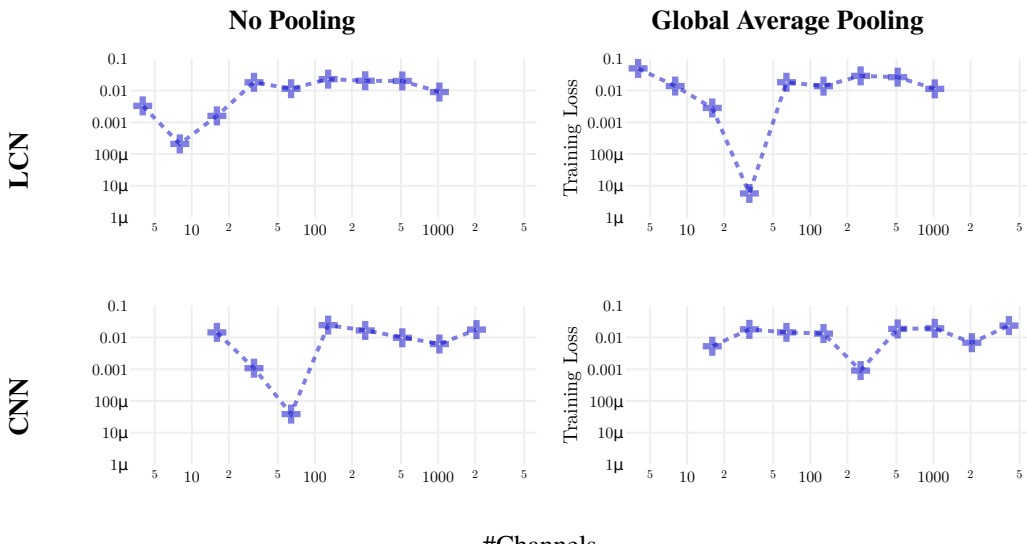

Figure 9: **No underfitting in small models.** **Training loss** (vertical axis) of best (in terms of validation loss) neural networks as the number of channels increases (horizontal axis). While perfect 0 loss is not achieved (but 100% accuracy is), we observe no consistent improvement when increasing the capacity of the network (left to right). This eliminates underfitting as a possible explanation for why small models perform worse in Figure 6 (b). See Figure 8 for validation loss and §G.1 for experimental details.

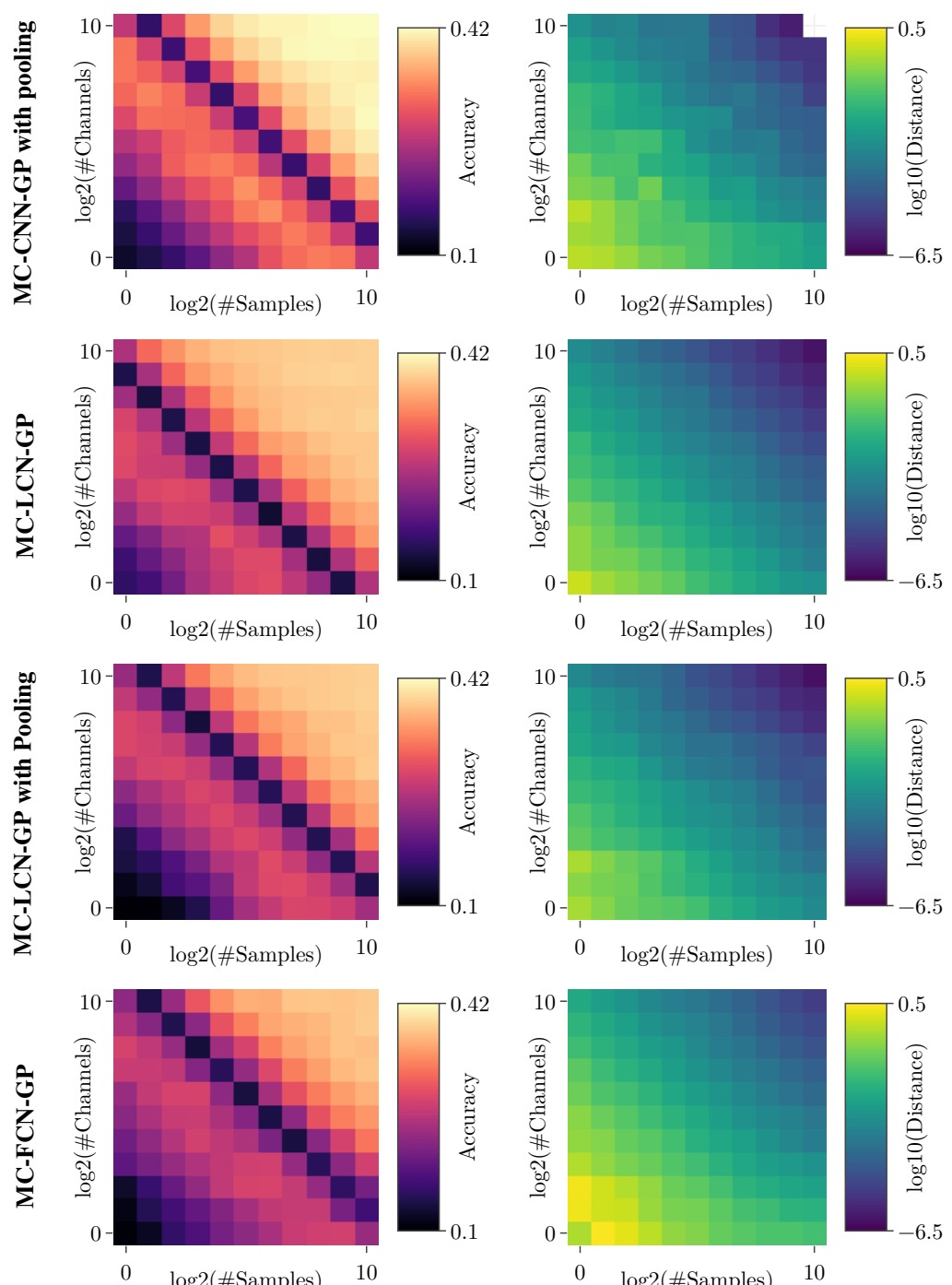

Figure 10: **Convergence of NN-GP Monte Carlo estimates.** As in Figure 5, **validation accuracy** (left) of MC-GPs increases with $n \times M$ (i.e. width times number of samples), while the **distance** (right) to the the respective exact GP kernel (or the best available estimate in the case of CNN-GP with pooling, top row) decreases. We remark that when using shared weights, convergence is slower as smaller number of independent random parameters are being used. See §G.2 for experimental details.

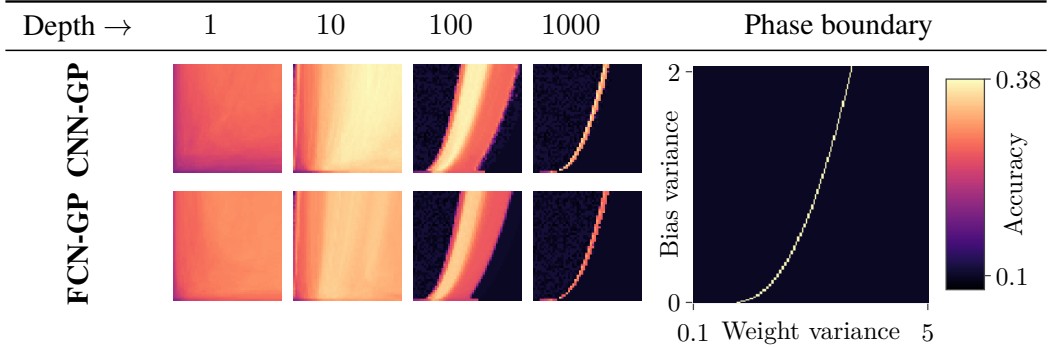

Figure 11: **Large depth performance of NN-GPs. Validation accuracy** of CNN- and FCN-GPs as a function of weight ($\sigma_\omega^2$, horizontal axis) and bias ($\sigma_b^2$, vertical axis) variances. As predicted in §B, the regions of good performance concentrate around the critical line (phase boundary, right) as the depth increases (left to right). All plots share common axes ranges and employ the erf nonlinearity. See §G.2 for experimental details.

## B  RELATIONSHIP TO DEEP SIGNAL PROPAGATION

The recurrence relation linking the GP kernel at layer $l+1$ to that of layer $l$ following from Equation 9 (i.e. $K_\infty^{l+1} = (\mathcal{C} \circ \mathcal{A})\left(K_\infty^l\right)$) is precisely the *covariance map* examined in a series of related papers on signal propagation (Xiao et al., 2018; Poole et al., 2016; Schoenholz et al., 2017; Lee et al., 2018) (modulo notational differences; denoted as $F$, $\mathcal{C}$ or e.g. $\mathcal{A} \star \mathcal{C}$ in Xiao et al. (2018)). In those works, the action of this map on hidden-state covariance matrices was interpreted as defining a dynamical system whose large-depth behavior informs aspects of trainability. In particular, as $l \to \infty$, $K_\infty^{l+1} = (\mathcal{C} \circ \mathcal{A})\left(K_\infty^l\right) \approx K_\infty^l \equiv K_\infty^*$, i.e. the covariance approaches a fixed point $K_\infty^*$. The convergence to a fixed point is problematic for learning because the hidden states no longer contain information that can distinguish different pairs of inputs. It is similarly problematic for GPs, as the kernel becomes pathological as it approaches a fixed point. Precisely, in both chaotic and ordered regimes, outputs of the GP become asymptically identically correlated. Either of these scenarios captures no information about the training data in the kernel and makes learning infeasible.

This problem can be ameliorated by judicious hyperparameter selection, which can reduce the rate of exponential convergence to the fixed point. For hyperpameters chosen on a critical line separating two untrainable phases, the convergence rates slow to polynomial, and very deep networks can be trained, and inference with deep NN-GP kernels can be performed – see Figure 11.

## C  STRIDED CONVOLUTIONS, AVERAGE POOLING IN INTERMEDIATE LAYERS, HIGHER DIMENSIONS

Our analysis in the main text can easily be extended to cover average pooling and strided convolutions (applied before the pointwise nonlinearity). Recall that conditioned on $K^l$ the pre-activation $z_j^l(x) \in \mathbb{R}^{d_1}$ is a zero-mean multivariate Gaussian. Let $B \in \mathbb{R}^{d_2 \times d_1}$ denote a linear operator. Then $Bz_j^l(x) \in \mathbb{R}^{d_2}$ is a zero-mean Gaussian, and the covariance is

$$\mathbb{E}_{\{\omega^l, b^l\}}\left[\left(Bz_j^l(x)\right)\left(Bz_j^l(x')\right)^T \middle| K^l\right] = B\mathbb{E}_{\{\omega^l, b^l\}}\left[z_j^l(x) z_j^l(x')^T \middle| K^l\right] B^T. \quad (20)$$

One can easily see that $\left\{Bz_j^l \middle| K^l\right\}_j$ are i.i.d. multivariate Gaussian as well.

**Strided convolution**. Strided convolution is equivalent to a non-strided convolution composed with subsampling. Let $s \in \mathbb{N}$ denote size of the stride. Then the strided convolution is equivalent to choosing $B$ as follows: $B_{ij} = \delta(is - j)$ for $i \in \{0, 1, \ldots (d_2 - 1)\}$.

**Average pooling**. Average pooling with stride $s$ and window size $ws$ is equivalent to choosing $B_{ij} = 1/ws$ for $i = 0, 1, \ldots (d_2 - 1)$ and $j = is, \ldots, (is + ws - 1)$.

**ND convolutions.** Note that our analysis in the main text (1D) easily extends to higher-dimensional convolutions by replacing integer pixel indices and sizes $d, \alpha, \beta$ with tuples (see also Figure 2). In Equation 2 $\beta$ values would have to span the hypercube $[\pm k]^N = \{-k, \ldots, k\}^N$ in the pre-activation definition. Similarly, in §3 the normalizing factor $d$ ($d^2$) should be the product (squared) of its entries, and summations over $\alpha, \beta$ should span the $[d_0] \times \cdots \times [d_N]$ hypercube as well. The definition of the kernel propagation operator $\mathcal{A}$ in Equation 5 will remain exactly the same, so long as $\beta$ is summed over the hypercube, and the variance weights remain respectively normalized $\sum_\beta v_\beta = 1$.

## D    REVIEW OF EXACT BAYESIAN REGRESSION WITH GPS

Our discussion in the paper has focused on model *priors*. A crucial benefit we derive by mapping to a GP is that Bayesian inference is straightforward to implement and can be done *exactly* for regression (Rasmussen & Williams, 2006, chapter 2), requiring only simple linear algebra. Let $\mathcal{X}$ denote training inputs $x_1, \ldots, x_{|\mathcal{X}|}$, $\mathbf{t}^T = \left(t_1, \ldots, t_{|\mathcal{X}|}\right)$ training targets, and collectively $\mathcal{D}$ for the training set. The integral over the posterior can be evaluated analytically to give a posterior predictive distribution on a test point $y_*$ which is Gaussian, $(z^*|\mathcal{D}, y^*) \sim \mathcal{N}\left(\mu_*, \sigma_*^2\right)$, with

$$\mu_* = \mathcal{K}\left(x^*, \mathcal{X}\right)^T \left(\mathcal{K}\left(\mathcal{X}, \mathcal{X}\right) + \sigma_\varepsilon^2 \mathbb{I}_{|\mathcal{X}|}\right)^{-1} \mathbf{t}, \tag{21}$$

$$\sigma_*^2 = \mathcal{K}\left(x^*, x^*\right) - \mathcal{K}\left(x^*, \mathcal{X}\right)^T \left(\mathcal{K}\left(\mathcal{X}, \mathcal{X}\right) + \sigma_\varepsilon^2 \mathbb{I}_{|\mathcal{X}|}\right)^{-1} \mathcal{K}\left(x^*, \mathcal{X}\right). \tag{22}$$

We use the shorthand $\mathcal{K}\left(\mathcal{X}, \mathcal{X}\right)$ to denote the $|\mathcal{X}| \times |\mathcal{X}|$ matrix formed by evaluating the GP covariance on the training inputs, and likewise $\mathcal{K}\left(x^*, \mathcal{X}\right)$ is a $|\mathcal{X}|$-length vector formed from the covariance between the test input and training inputs. Computationally, the costly step in GP posterior predictions comes from the matrix inversion, which in all experiments were carried out exactly, and typically scales as $\mathcal{O}\left(|\mathcal{X}|^3\right)$ (though algorithms scaling as $\mathcal{O}\left(|\mathcal{X}|^{2.4}\right)$ exist for sufficiently large matrices). Nonetheless, there is a broad literature on approximate Bayesian inference with GPs which can be utilized for efficient implementation (Rasmussen & Williams, 2006, chapter 8); (Quiñonero-Candela & Rasmussen, 2005; Titsias, 2009).

## E    EQUIVALENCE BETWEEN RANDOMLY INITIALIZED NNS AND GPS

In this section, we present two different approaches, the *sequential limit* (§E.3) and *simultaneous limit* (§E.4), to illustrate the relationship between many-channels Bayesian CNNs and GPs.

**Sequential limit** (§E.3) involves taking the infinite channel limit in hidden layers in a sequence, starting from bottom (closest to inputs) layers and going upwards (to the outputs), i.e. $n^1 \to \infty, \ldots, n^L \to \infty$. Note that this approach in fact only gives intuition into construction a GP using a NN architecture to define its covariance, and does not provide guarantees on actual convergence of large but finite Bayesian CNNs to GPs (which is of most practical interest), nor does it guarantee the existence of the specified GP on a given probability space. However, it has the following benefits:

1. Weak assumptions on the NN activation function $\phi$ and on the distribution of the NN parameters.

2. The arguments can be easily extended to more complicated network architectures, e.g. architectures with max pooling, dropout, etc.

3. A straightforward and intuitive way to compute the covariance of the Gaussian process without diving into mathematical details.

**Simultaneous limit** (§E.4) considers growing the number of channels in hidden layers uniformly, i.e. $\min \left\{n^1, \ldots, n^L\right\} \to \infty$. This approach establishes convergence of finite channel Bayesian CNNs to GPs and is thus a more practically relevant result. However, it makes stronger assumptions, and the proof is more involved.

We highlight that the GPs obtained by the two approaches are identical.

In both sections, we only provide the arguments for CNNs. It is straightforward (and in fact simpler) to extend them to LCNs and FCNs. Indeed, an FCN is a particular case of a CNN where the

inputs and filters have singular spatial dimensions ($d = 1$, $k = 0$). For LCNs, the proof goes through in an identical fashion if we replace $\mathcal{A}$ with $\mathcal{A}^{\text{LCN}}$ defined as $\left[\mathcal{A}^{\text{LCN}}(K)\right]_{\alpha,\alpha'}(x, x') \equiv \delta_{\alpha,\alpha'}\left[\mathcal{A}(K)\right]_{\alpha,\alpha'}(x, x')$.

## E.1  Setup

**Probability space.** Let $\mathscr{P}$ be a collection of countably many mutually independent random variables (R.V.s) defined on a probability space $(\Omega, \mathcal{F}, P)$, where $\mathcal{F}$ is a product Borel $\sigma$-algebra and $P$ is the probability measure. Here $\mathscr{P} \equiv \mathscr{W} \cup \mathscr{B} \cup \mathscr{H}$ is the collection of parameters used to define neural networks:

(i) **Weights.** $\mathscr{W} = \bigcup_{l \in \mathbb{N}} \mathscr{W}^l$ and $\mathscr{W}^l = \left\{\omega^l_{ij,\beta} : i, j \in \mathbb{N}, \beta \in [\pm k]\right\}$, where $[\pm k] \equiv [-k, k] \cap \mathbb{Z}$. We assume $\omega^l_{ij,\beta}$ are i.i.d. R.V.s with mean zero and finite variance $0 < \sigma^2_\omega < \infty$ (note the lack of scaling by input dimensionality, compensated for later in Equations 29 and 42; see also similar notation used in Matthews et al. (2018a)). When $l = 0$, we further assume they are Gaussian distributed.

(ii) **Biases.** $\mathscr{B} = \bigcup_{l \in \mathbb{N}} \mathscr{B}^l$ and $\mathscr{B}^l = \left\{b^l_j : j \in \mathbb{N}\right\}$. We assume $b^l_j$ are i.i.d. Gaussian with mean zero and variance $0 \leq \sigma^2_b < \infty$.

(iii) **Place-holder.** $\mathscr{H}$ is a place-holder to store extra (if needed) R.V.s , e.g. parameters coming from the final dense layer.

**Inputs.** We will consider a fixed $\mathcal{X} \subseteq \mathbb{R}^{n^0 \times d}$ to denote the inputs, with input channel count $n^0$, number of pixels $d$. Assume $x \neq 0$ for all $x \in \mathcal{X}$ and $|\mathcal{X}|$, the cardinality of the inputs, is finite.

However, our results can be straightforwardly extended to a countably-infinite input indexing spaces $\mathcal{X}$ for certain topologies via an argument presented in Matthews et al. (2018a, section 2.2), allowing to infer weak convergence on $\mathcal{X}$ from convergence on any finite subset (which is the case we consider in this text; see also Billingsley (1999, page 19) for details). For this reason, as in Matthews et al. (2018a), weak convergence of countably-infinite stochastic processes will be considered with respect to the topology generated by the following metric:

$$\nu(s, s') \equiv \sum_{k=1}^{\infty} 2^{-k} \min\left(1, |s_k - s'_k|\right), \quad \forall s, s' \in \mathbb{R}^{\mathbb{N}}.$$

**Notation, shapes, and indexing.** We adopt the notation, shape, and indexing convention similar to §2.1, which the reader is encouraged to review. We emphasize that whenever an index is omitted, the variable is assumed to contain all possible entries along the respective dimension (e.g. whole $\mathcal{X}$ if $x$ is omitted, or all $n^l$ channels if the channel $i$ is omitted).

## E.2  Preliminary

We will use the following well-known theorem.

**Theorem E.1.** Let $X, \{X_n\}_{n \in \mathbb{N}}$ be R.V.s in $\mathbb{R}^m$. The following are equivalent:

(i) $X_n \xrightarrow{\mathcal{D}} X$ (converges in distribution / converges weakly),

(ii) (Portmanteau Theorem) For all bounded continuous function $f : \mathbb{R}^m \to \mathbb{R}$,

$$\lim_{n \to \infty} \mathbb{E}\left[f(X_n)\right] = \mathbb{E}\left[f(X)\right], \tag{23}$$

(iii) (Lévy's Continuity Theorem) The characteristic functions of $X_n$, i.e. $\mathbb{E}\left[e^{\mathring{\imath} t^T X_n}\right]$, converge to those of $X$ pointwise, i.e. for all $t \in \mathbb{R}^m$,

$$\lim_{n \to \infty} \mathbb{E}\left[e^{\mathring{\imath} t^T X_n}\right] = \mathbb{E}\left[e^{\mathring{\imath} t^T X}\right], \tag{24}$$

where $\mathring{\imath}$ denotes the imaginary unit.

Using the equivalence between (i) and (iii), it is straightforward to show that

**Theorem E.2** (Cramer-Wold, (Billingsley (1995), Theorem 29.4))**.**

$$X_n \overset{\mathcal{D}}{\to} X \iff a^T X_n \overset{\mathcal{D}}{\to} a^T X \quad \text{for all } a \in \mathbb{R}^m. \tag{25}$$

In particular,

$$X_n \overset{\mathcal{D}}{\to} \mathcal{N}(0, \Sigma) \iff a^T X_n \overset{\mathcal{D}}{\to} \mathcal{N}(0, a^T \Sigma a) \quad \text{for all } a \in \mathbb{R}^m. \tag{26}$$

### E.3 Sequential Limit

In this section, we give intuition into constructing a Gaussian process using an infinite channel CNN with the limits taken sequentially. Informally, our argument amounts to showing that if the inputs $z^{l-1}$ to the layer $l$ are a multivariate Gaussian with covariance $\mathcal{A}\left(K^l\right) \otimes I_{n^l}$, then it's outputs $z^l$ converge in distribution to a multivariate Gaussian with covariance $\mathcal{A}\left(K^{l+1}\right) \otimes I_{n^{l+1}}$ as $n^l \to \infty$. This "allows" us to sequentially replace $z^1, z^2, \ldots, z^L$ with their respective limiting Gaussian R.V.s as we take $n^1 \to \infty, n^2 \to \infty, \ldots, n^L \to \infty$. However, since each convergence only holds in distribution and does not guarantee the existence of the necessary GPs on a given probability space, what follows merely gives intuition into understanding the relationship between wide Bayesian neural networks and GPs (contrary to §E.4, which presents a rigorous convergence proof for $\min\left\{n^1, \ldots, n^L\right\} \to \infty$).

Let $\Psi_1$ denote the space of functions with a uniformly bounded second moment, i.e. having

$$C_2(R, \phi) \equiv \sup_{1/R \le r \le R} \mathbb{E}_{x \sim \mathcal{N}(0, r)} |\phi(x)|^2 < \infty \quad \text{for every} \quad R \ge 1. \tag{27}$$

Let $\mathbb{N}^* = \mathbb{N} \backslash \{0\}$. We construct Gaussian processes with the following iterative in $l$ (from 0 to $L$) procedure:

(i) If $l > 0$, define random variables (a GP) $\left\{z_i^{l,\infty}\right\}_{i \in \mathbb{N}^*}$ in $(\Omega, \mathcal{F}, P)$ as i.i.d. Gaussian with mean zero and covariance $\mathcal{A}\left(K_\infty^l\right)$, so that they are also independent from any future events, i.e. independent from the $\sigma$-algebra generated by all R.V.s with layer index greater than $l$. Here we implicitly assume the probability space $(\Omega, \mathcal{F}, P)$ has enough capacity to fit in the R.V.s generated by the above procedure, if not we will extend the sample space $\Omega$ using product measures.

(ii) For $n \in \mathbb{N}^*$, define

$$y_{i,\alpha}^l(x) \equiv \begin{cases} x_{i,\alpha} & l = 0 \\ \phi\left(z_{i,\alpha}^{l-1,\infty}(x)\right) & l > 0 \end{cases}, \tag{28}$$

$$z_{i,\alpha}^{l,n}(x) \equiv \begin{cases} \frac{1}{\sqrt{n^0}} \sum_{j \in [n^0]} \sum_{\beta \in [\pm k]} \sqrt{v_\beta} \omega_{ij,\beta}^l y_{j,\alpha+\beta}^l(x) + b_i^l & l = 0 \\ \frac{1}{\sqrt{n}} \sum_{j \in [n]} \sum_{\beta \in [\pm k]} \sqrt{v_\beta} \omega_{ij,\beta}^l y_{j,\alpha+\beta}^l(x) + b_i^l & l > 0 \end{cases}, \tag{29}$$

where

$$[n] \equiv \{1, \ldots, n\} \quad \text{and} \quad [\pm k] \equiv \{-k, \ldots, 0, \ldots k\}. \tag{30}$$

(iii) Prove that for any finite $m \ge 1$, $\left[z_i^{l,n}\right]_{i \in [m]} \subseteq \mathbb{R}^{|\mathcal{X}|d}$ converges in distribution to a multivariate normal with mean zero and covariance $\mathcal{A}\left(K_\infty^l\right) \otimes I_m$ as $n \to \infty$, where $K_\infty^l$ is defined identically to Equation 9. As a consequence, per our remark in §E.1, $\left\{z_i^{l,n}\right\}_{i \in \mathbb{N}^*}$ converges weakly to the GP $\left\{z_i^{l,\infty}\right\}_{i \in \mathbb{N}^*}$.

In what follows, we use the central limit theorem to prove (iii).

**Theorem E.3.** If $\phi \in \Psi_1$, then for every $l \geq 0$ and every $m \geq 1$, $\left[z_i^{l,n}\right]_{i \in [m]} \subseteq \mathbb{R}^{|\mathcal{X}|d}$ converges in distribution to a multivariate normal with mean zero and covariance $\mathcal{A}\left(K_\infty^l\right) \otimes I_m$.

*Proof.* We proceed by induction. This is obvious in the base case $l = 0$, since the weights and biases are assumed to be independent Gaussian. Now we assume the theorem holds for $l - 1$. This implies $\left\{y_i^l\right\}_{i \in \mathbb{N}^*} = \left\{\phi\left(z_i^{l-1,\infty}\right)\right\}_{i \in \mathbb{N}^*}$ are i.i.d. random variables.

Choose a vector $a \equiv [a_i(x)]_{i \in [m], x \in \mathcal{X}}^T \in \mathbb{R}^{m|\mathcal{X}|}$. Then

$$\sum_{\substack{i \in [m] \\ x \in \mathcal{X}}} a_i(x) z_i^{l,n}(x) = \sum_{\substack{i \in [m] \\ x \in \mathcal{X}}} a_i(x) \frac{1}{\sqrt{n}} \left(\sum_{j \in [n]} \sum_{\beta \in [\pm k]} \sqrt{v_\beta} \omega_{ij,\beta}^l y_{j,\alpha+\beta}^l(x) + b_i^l\right) \tag{31}$$

$$\frac{1}{\sqrt{n}} \left(\sum_{j \in [n]} \sum_{\substack{i \in [m] \\ x \in \mathcal{X}}} a_i(x) \sum_{\beta \in [\pm k]} \sqrt{v_\beta} \omega_{ij,\beta}^l y_{j,\alpha+\beta}^l(x)\right) + \sum_{\substack{i \in [m] \\ x \in \mathcal{X}}} a_i(x) b_i^l \tag{32}$$

$$\equiv \frac{1}{\sqrt{n}} \sum_{j \in [n]} u_j + q. \tag{33}$$

It is not difficult to see that $u_j$ are i.i.d. and $q$ is Gaussian. Then we can use the central limit theorem to conclude that the above converges in distribution to a Gaussian, once we verify the second moment of $u_j$ is finite. Using the fact that $\left\{\omega_{ij,\beta}^l\right\}_{i,\beta}$ is a collection of independent R.V.s, and integrating over these R.V.s first, we get

$$\mathbb{E}u_j^2 = \mathbb{E}\left[\sum_{i \in [m], x \in \mathcal{X}} a_i(x) \sum_{\beta \in [\pm k]} \sqrt{v_\beta} \omega_{ij,\beta}^l y_{j,\alpha+\beta}^l(x)\right]^2 \tag{34}$$

$$= \sigma_\omega^2 \sum_{i \in [m]} \sum_{\beta \in [\pm k]} v_\beta \mathbb{E}\left[\sum_{x \in \mathcal{X}} a_i(x) y_{j,\alpha+\beta}^l(x)\right]^2 \tag{35}$$

$$= \sum_{i \in [m]} a_i^T \bar{\mathcal{A}}\left(K_\infty^l\right) a_i \tag{36}$$

where by $\bar{\mathcal{A}}$ we denote the linear transformation part of $\mathcal{A}$ from Equation 5, i.e. $\mathcal{A}$ without the translation term $\sigma_b^2$:

$$\left[\bar{\mathcal{A}}\left(K\right)\right]_{\alpha,\alpha'}(x, x') \equiv \sigma_\omega^2 \sum_\beta v_\beta [K]_{\alpha+\beta,\alpha'+\beta}(x, x'). \tag{37}$$

To prove finiteness of the second moment in Equation 36, it is sufficient to show that all the diagonal terms of $K_\infty^l$ are finite. This easily follows from the assumption $\phi \in \Psi_1$ and the definition of $K_\infty^l = (\mathcal{C} \circ \mathcal{A})^l\left(K^0\right)$. Together with the distribution of $v$ (whose covariance is straightforward to compute), the joint distribution of $\left[z_i^{l,n}\right]_{i \in [m]}$ converges weakly to a mean zero Gaussian with covariance matrix $\mathcal{A}\left(K_\infty^l\right) \otimes I_m$ by Theorem E.1 (Equation 26). $\qquad\square$

**Remark E.1.** The results of Theorem E.3 can be strengthened / extended in many directions.

(1) The same result for a countably-infinite input index set $\mathcal{X}$ follows immediately according to the respective remark in §E.1.

(2) The same analysis carries over (and the covariance matrix can be computed without much extra effort) if we stack a channel-wise deterministic affine transform after the convolution operator. Note that average pooling (not max pooling, which is not affine), global average pooling and the convolutional striding are particular examples of such affine tranforms. Moreover, valid padding (i.e. no padding) convolution can be regarded as a subsampling operator (a linear projection) composed with the regular circular padding.

(3) The same analysis applies to max pooling, but computing the covariance may require non-trivial effort. Let $\mathfrak{m}$ denote the max pooling operator and assume it is applied right after the activation function. The assumption $\left\{y_i^l\right\}_{i\in[n]} = \left\{\phi(z_i^{l-1,\infty})\right\}_{i\in[n]}$ are i.i.d. implies $\left\{\mathfrak{m}\left(y_i^l\right)\right\}_{i\in[n]}$ are also i.i.d. Then we can proceed exactly as above except for verifying the finiteness of second moment of $\mathfrak{m}(y_i^l)$ with the following trivial estimate:

$$\mathbb{E}\max\left\{y_{i,\alpha}^l\right\}_{\alpha\in[s]}^2 \le \mathbb{E}\sum_{\alpha\in[s]}\left|y_{i,\alpha}^l\right|^2 \tag{38}$$

where $s$ is the window size of the max pooling.

In general, one can stack a channel-wise deterministic operator $\mathfrak{op}$ on $\left\{y_i^l\right\}$ so long as the second moment of $\left\{\mathfrak{op}\left(y_i^l\right)\right\}$ is finite. One can also stack a stochastic operator (e.g. dropout), so long as the outputs are still channel-wisely i.i.d. and have finite second moments.

### E.4 SIMULTANEOUS LIMIT

In this section, we present a sufficient condition on the activation function $\phi$ so that the neural networks converge to a Gaussian process as all the widths approach infinity simultaneously. Precisely, let $t\in\mathbb{N}$ and for each $l\ge 0$, let $n^l : \mathbb{N}\to\mathbb{N}$ be the width function at layer $l$ (by convention $n^0(t) = n^0$ is constant). We are interested in the simultaneous limit $n^l = n^l(t)\to\infty$ as $t\to\infty$, i.e., for any fixed $L\ge 1$

$$\min\left\{n^1(t),\ldots,n^L(t)\right\}\xrightarrow[t\to\infty]{}\infty. \tag{39}$$

Define a sequence of finite channel CNNs as follows:

$$y_{i,\alpha}^{l,t}(x) \equiv \begin{cases} x_{i,\alpha}, & l=0 \\ \phi\left(z_{i,\alpha}^{l-1,t}(x)\right), & l>0 \end{cases}, \tag{40}$$

$$\tag{41}$$

$$z_{i,\alpha}^{l,t}(x) \equiv \begin{cases} \frac{1}{\sqrt{n^0}}\sum_{j\in[n^0]}\sum_{\beta\in[\pm k]}\sqrt{v_\beta}\omega_{ij,\beta}^l y_{j,\alpha+\beta}^{l,t}(x) + b_i^l, & l=0 \\ \sum_{j\in[n^l(t)]}\sum_{\beta\in[\pm k]}\sqrt{v_\beta}\omega_{ij,\beta}^l y_{j,\alpha+\beta}^{l,t}(x) + b_i^l, & l>0 \end{cases}. \tag{42}$$

This network induces a sequence of covariance matrices $K_t^l$ (which are R.V.s): for $l\ge 0$ and $t\ge 0$, for $x, x'\in\mathcal{X}$

$$\left[K_t^l\right]_{\alpha,\alpha'}(x,x') \equiv \frac{1}{n^l(t)}\sum_{i=1}^{n^l(t)} y_{i,\alpha}^{l,t}(x)y_{i,\alpha'}^{l,t}(x'). \tag{43}$$

We make an extra assumption on the parameters.

**Assumption**: all R.V.s in $\mathscr{W}$ are Gaussian distributed.

**Notation.** Let $\text{PSD}_m$ denote the set of $m\times m$ positive semi-definite matrices and for $R\ge 1$, define

$$\text{PSD}_m(R) \equiv \left\{\Sigma\in\text{PSD}_m : 1/R\le\Sigma_{\alpha,\alpha}\le R \quad\text{for}\quad 1\le\alpha\le m\right\}. \tag{44}$$

Further let $\mathcal{T}_\infty : \text{PSD}_2\to\mathbb{R}$ be a function given by

$$\mathcal{T}_\infty(\Sigma) \equiv \mathbb{E}_{(x,y)\sim\mathcal{N}(0,\Sigma)}\left[\phi(x)\phi(y)\right], \tag{45}$$

and $C_k\left(\phi, R\right)$ (may equal $\infty$) denotes the uniform upper bound for the $k$-th moment

$$C_k\left(\phi, R\right) \equiv \sup_{1/R\le r\le R}\mathbb{E}_{x\sim\mathcal{N}(0,r)}\left|\phi(x)\right|^k. \tag{46}$$

Let $\Psi$ denotes the space of measurable functions $\phi$ with the following properties:

1. **Uniformly bounded second moment:** for every $R \geq 1$, $C_2(\phi, R) < \infty$.

2. **Lipschitz continuity:** for every $R \geq 1$, there exists $\beta = \beta(\phi, R) > 0$ such that for all $\Sigma, \Sigma' \in \text{PSD}_2(R)$,

$$|\mathcal{T}_\infty(\Sigma) - \mathcal{T}_\infty(\Sigma')| \leq \beta \|\Sigma - \Sigma'\|_\infty; \tag{47}$$

3. **Uniform convergence in probability:** for every $R \geq 1$ and every $\varepsilon > 0$ there exists a positive sequence $\rho_n(\phi, \varepsilon, R)$ with $\rho_n(\phi, \varepsilon, R) \to 0$ as $n \to \infty$ such that for every $\Sigma \in \text{PSD}_2(R)$ and any $\{(x_i, y_i)\}_{i=1}^n$ i.i.d. $\sim \mathcal{N}(0, \Sigma)$

$$P\left(\left|\frac{1}{n}\sum_{i=1}^n \phi(x_i)\phi(y_i) - \mathcal{T}_\infty(\Sigma)\right| > \varepsilon\right) \leq \rho_n(\phi, \varepsilon, R). \tag{48}$$

We will also use $\Psi_1, \Psi_2$ and $\Psi_3$ to denote the spaces of measurable functions $\phi$ satisfying properties 1, 2, and 3, respectively. It is not difficult to see that for every $i$, $\Psi_i$ is a vector space, and so is $\Psi = \cap_i \Psi_i$.

Finally, we say that a function $f : \mathbb{R} \to \mathbb{R}$ is exponentially bounded if there exist $a, b > 0$ such that

$$|f(x)| \leq a e^{b|x|} \quad \text{a.e. (almost everywhere)} \tag{49}$$

We now prove our main result presented in §2.2.3 through the following three theorems.

**Theorem E.4.** If $\phi$ is absolutely continuous and $\phi'$ is exponentially bounded then $\phi \in \Psi$.

**Theorem E.5.** If $\phi \in \Psi$, then for $l \geq 0$, $K_t^l \xrightarrow{P} K_\infty^l$.

**Theorem E.6.** If for $l \geq 0$, $K_t^l \xrightarrow{P} K_\infty^l$ and $m \geq 1$, the joint distribution of $\left[z_j^{l,t}\right]_{j\in[m]}$ converges in distribution to a multivariate normal distribution with mean zero and covariance $\mathcal{A}\left(K_\infty^l\right) \otimes I_m$.

The proofs of Theorems E.4, E.5, and E.6 can be found in §E.7, §E.6, and §E.5 respectively. The proof of Theorem E.5 is slightly more technical and we will borrow some ideas from Daniely et al. (2016).

## E.5 PROOF OF THEOREM E.6

Per Equation 26, it suffices to prove that for any vector $[a_i(x)]_{i\in[m], x\in\mathcal{X}} \in \mathbb{R}^{m|\mathcal{X}|}$,

$$\sum_{\substack{i\in[m]\\x\in\mathcal{X}}} a_i(x)z_i^{l,t}(x) \xrightarrow{\mathcal{D}} \mathcal{N}\left(0, \sum_{i\in[m], x\in\mathcal{X}} a_i(x)^T \mathcal{A}\left(K_\infty^l\right) a_i(x)\right). \tag{50}$$

Indeed, the characteristic function

$$\mathbb{E}\exp\left(\mathbb{i}\sum_{i\in[m], x\in\mathcal{X}} a_i(x)z_i^{l,t}(x)\right) \tag{51}$$

$$=\mathbb{E}\exp\left(\mathbb{i}\sum_{\substack{i\in[m]\\x\in\mathcal{X}}} a_i(x)\left(\frac{1}{\sqrt{n^l(t)}}\sum_{j\in[n^l(t)]}\sum_{\beta\in[\pm k]}\sqrt{v_\beta}\omega_{ij,\beta}^l y_{j,\alpha+\beta}^{l,t}(x) + b_i^l\right)\right). \tag{52}$$

Note that conditioned on $\left\{y_j^{l,t}\right\}_{j\in[n^l(t)]}$, the exponent in the above expression is just a linear combination of independent Gaussian R.V.s $\left\{\omega_{ij,\beta}^l, b_i^l\right\}$, which is also a Gaussian. Integrating out these

R.V.s using the formula of the characteristic function of a Gaussian distribution yields

$$\mathbb{E}\exp\left(\mathrm{i}\sum_{i\in[m],\,x\in\mathcal{X}}a_i(x)z_i^{l,t}(x)\right) \tag{53}$$

$$=\mathbb{E}\exp\left(-\frac{1}{2}\sum_{i\in[m]}a_i^T\left(\mathcal{A}\left(K_t^l\right)\right)a_i\right) \tag{54}$$

$$\longrightarrow\exp\left(-\frac{1}{2}\sum_{i\in[m]}a_i^T\left(\mathcal{A}\left(K_\infty^l\right)\right)a_i\right)\quad\text{as}\quad t\to\infty\,, \tag{55}$$

where we have used $K_t^l\xrightarrow{P}K_\infty^l$ and Lipschitz continuity of the respective function in the vicinity of $K_\infty^l$ in the last step. Therefore, Equation 50 is true by Theorem E.1 (iii). As in §E.3, the same result for a countably-infinite input index set $\mathcal{X}$ follows immediately according to the respective remark in §E.1.

$\square$

**Remark E.2.** We briefly comment how to handle the cases when stacking an average pooling, a subsampling or a dense layer after flattening the activations in the last layer.

(i) **Global Average pooling / subsampling.** Let $B\in\mathbb{R}^{1\times d}$ be any deterministic *linear* functional defined on $\mathbb{R}^d$. The fact that

$$K_t^l\xrightarrow{P}K_\infty^l \tag{56}$$

implies that the empirical covariance of $\left\{By_j^{l,t}\right\}$

$$\frac{1}{n^l(t)}\sum_{j\in n^l(t)}B^{\otimes|\mathcal{X}|}y_j^{l,t}\left(B^{\otimes|\mathcal{X}|}y_j^{l,t}\right)^T\xrightarrow{P}B^{\otimes|\mathcal{X}|}K_\infty^l\left(B^{\otimes|\mathcal{X}|}\right)^T \tag{57}$$

where $B^{\otimes|\mathcal{X}|}\in\mathbb{R}^{1\times d|\mathcal{X}|}$, $|\mathcal{X}|$ copies of $B$. Invoking the same "characteristic function" arguments as above, it is not difficult to show that stacking a dense layer (assuming the weights and biases are drawn from i.i.d. Gaussian with mean zero and variances $\sigma_\omega^2$ and $\sigma_b^2$, and are properly normalized) on top of $\left\{By_j^{l,t}\right\}$ the outputs are i.i.d. Gaussian with mean zero and covariance $\sigma_\omega^2B^{\otimes|\mathcal{X}|}K_\infty^l\left(B^{\otimes|\mathcal{X}|}\right)^T+\sigma_b^2$. Taking $B=\left(\frac{1}{d},\ldots,\frac{1}{d}\right)\in\mathbb{R}^{1\times d}$ $\left(\text{or }B=e_\alpha\in\mathbb{R}^{1\times d}\right)$ implies the result of global average pooling (§3.2.1, Equation 17), or subsampling (§3.2.2, Equation 18).

(ii) **Vectorization and a dense layer.** Let $\left\{\omega_{ij,\alpha}^l\right\}_{i\in[m],j\in[n^l(t)],\alpha\in[d]}$ be the weights of the dense layer, $\omega_{ij,\alpha}^l$ represents the weight connecting the $\alpha$-th pixel of the $j$-channel to the $i$-th output. Note that the range of $\alpha$ is $[d]$ not $[\pm k]$ because there is no weight sharing. Define the outputs to be

$$f_i(x)=\frac{1}{\sqrt{n^l(t)d}}\sum_{\alpha\in[d]}\sum_{j\in[n^l(t)]}\omega_{ij,\alpha}^l y_{j,\alpha}^{l,t}(x)+b_i^l \tag{58}$$

Now let $[a_i(x)]_{i\in[m],x\in\mathcal{X}}\in\mathbb{R}^{m|\mathcal{X}|}$ and compute the characteristic function of

$$\sum_{i,x}a_i(x)f_i(x). \tag{59}$$

Using the fact $\mathbb{E}\omega_{ij,\alpha}\omega_{i'j',\alpha'} = 0$ unless $(ij,\alpha) = (i'j',\alpha')$ and integrating out the R.V.s of the dense layer, the characteristic function is equal to

$$\mathbb{E}\exp\left(-\frac{1}{2}\sum_{i\in[m]}\sum_{x,x'\in\mathcal{X}}a_i(x)a_i(x')\left(\frac{1}{n^l(t)d}\sum_{\substack{j\in[n^l(t)]\\\alpha\in[d]}}\sigma_\omega^2 y_{j,\alpha}^{l,t}(x)y_{j,\alpha}^{l,t}(x') + \sigma_b^2\right)\right) \quad (60)$$

$$= \mathbb{E}\exp\left(-\frac{1}{2}\sum_{i\in[m]}\sum_{x,x'\in\mathcal{X}}a_i(x)a_i(x')\bar{\text{tr}}\left(\sigma_\omega^2 K_t^l(x,x') + \sigma_b^2\right)\right) \quad (61)$$

$$\longrightarrow \exp\left(-\frac{1}{2}\sum_{i\in[m]}\sum_{x,x'\in\mathcal{X}}a_i(x)a_i(x')\bar{\text{tr}}\left(\sigma_\omega^2 K_\infty^l(x,x') + \sigma_b^2\right)\right), \quad (62)$$

where $\bar{\text{tr}}$ denotes the mean trace operator acting on the pixel by pixel matrix, i.e. the functional computing the mean of the diagonal terms of the pixel by pixel matrix. Therefore $[f_i]_{i\in[m]}$ converges weakly to a mean zero Gaussian with covariance $\left[\sigma_\omega^2\bar{\text{tr}}\left(K_\infty^l(x,x')\right) + \sigma_b^2\right]_{x,x'\in\mathcal{X}} \otimes I_m$ in the case of vectorization (§3.1).

## E.6 PROOF OF THEOREM E.5

*Proof.* We recall $K_t^L$ and $K_\infty^L$ to be random matrices in $\mathbb{R}^{d|\mathcal{X}|\times d|\mathcal{X}|}$, and we will prove convergence $K_t^L \xrightarrow[t\to\infty]{P} K_\infty^L$ with respect to $\|\cdot\|_\infty$, the pointwise $\ell^\infty$-norm (i.e. $\|K\|_\infty = \max_{x,x',\alpha,\alpha'}|K_{\alpha,\alpha'}(x,x')|$). Note that due to finite dimensionality of $K^L$, convergence w.r.t. all other norms follows.

We first note that the affine transform $\mathcal{A}$ is $\sigma_\omega^2$-Lipschitz and property 2 of $\Psi$ implies that the $\mathcal{C}$ operator is $\beta$-Lipschitz (both w.r.t. w.r.t. $\|\cdot\|_\infty$). Indeed, if we consider

$$\Sigma \equiv \begin{pmatrix} [K]_{\alpha,\alpha}(x,x) & [K]_{\alpha,\alpha'}(x,x') \\ [K]_{\alpha',\alpha}(x',x) & [K]_{\alpha',\alpha'}(x',x') \end{pmatrix}, \quad (63)$$

then $[\mathcal{C}(K)]_{\alpha,\alpha'}(x,x') = \mathcal{T}_\infty(\Sigma)$. Thus $\mathcal{C}\circ\mathcal{A}$ is $\sigma_\omega^2\beta$-Lipschitz.

We now prove the theorem by induction. Assume $K_t^l \xrightarrow{P} K_\infty^l$ as $t\to\infty$ (obvious for $l=0$).

We first remark that $K_\infty^l \in \text{PSD}_{|\mathcal{X}|d}$, since $\text{PSD}_{|\mathcal{X}|d} \ni \mathbb{E}\left[K_t^l\right] \to K_\infty^l$ and $\text{PSD}_{|\mathcal{X}|d}$ is closed. Moreover, due to Equation 4, $\mathcal{A}$ necessarily preserves positive semi-definiteness, and therefore $\mathcal{A}\left(K_\infty^l\right) \in \text{PSD}_{|\mathcal{X}|d}$ as well.

Now let $\varepsilon > 0$ be sufficiently small so that the $\frac{\varepsilon}{2\beta}$-neighborhood of $\mathcal{A}(K_\infty^l)$ is contained in $\text{PSD}_{|\mathcal{X}|d}(R)$, where we take $R$ to be large enough for $K_\infty^l$ to be an interior point of $\text{PSD}_{|\mathcal{X}|d}(R)$.[9] Since

$$\left\|K_\infty^{l+1} - K_t^{l+1}\right\|_\infty \leq \left\|K_\infty^{l+1} - \mathcal{C}\circ\mathcal{A}\left(K_t^l\right)\right\|_\infty + \left\|\mathcal{C}\circ\mathcal{A}\left(K_t^l\right) - K_t^{l+1}\right\|_\infty \quad (64)$$

$$= \left\|\mathcal{C}\circ\mathcal{A}\left(K_\infty^l\right) - \mathcal{C}\circ\mathcal{A}\left(K_t^l\right)\right\|_\infty + \left\|\mathcal{C}\circ\mathcal{A}\left(K_t^l\right) - K_t^{l+1}\right\|_\infty, \quad (65)$$

to prove $K_t^{l+1} \xrightarrow{P} K_\infty^{l+1}$, it suffices to show that for every $\delta > 0$, there is a $t^*$ such that for all $t > t^*$,

$$P\left(\left\|\mathcal{C}\circ\mathcal{A}\left(K_\infty^l\right) - \mathcal{C}\circ\mathcal{A}\left(K_t^l\right)\right\|_\infty > \frac{\varepsilon}{2}\right) + P\left(\left\|\mathcal{C}\circ\mathcal{A}\left(K_t^l\right) - K_t^{l+1}\right\|_\infty > \frac{\varepsilon}{2}\right) < \delta. \quad (66)$$

By our induction assumption, there is a $t^l$ such that for all $t > t^l$

$$P\left(\left\|K_\infty^l - K_t^l\right\|_\infty > \frac{\varepsilon}{2\sigma_\omega^2\beta}\right) < \frac{\delta}{3}. \quad (67)$$

---

[9]Such $R$ always exists, because the diagonal terms of $K_\infty^l$ are always non-zero. See (Long & Sedghi, 2019, Lemma 4) for proof.

Since $\mathcal{C} \circ \mathcal{A}$ is $\sigma_\omega^2 \beta$-Lipschitz, then

$$P\left(\left\|\mathcal{C} \circ \mathcal{A}\left(K_\infty^l\right) - \mathcal{C} \circ \mathcal{A}\left(K_t^l\right)\right\|_\infty > \frac{\varepsilon}{2}\right) < \frac{\delta}{3}. \tag{68}$$

To bound the second term in Equation 66, let $U(t)$ denote the event

$$U(t) \equiv \left\{\mathcal{A}\left(K_t^l\right) \in \text{PSD}_{|\mathcal{X}|d}(R)\right\} \tag{69}$$

and $U(t)^c$ its complement. For all $t > t^l$ its probability is

$$P\left(U(t)^c\right) < P\left(\left\|\mathcal{A}\left(K_\infty^l\right) - \mathcal{A}\left(K_t^l\right)\right\|_\infty > \frac{\varepsilon}{2\beta}\right) \qquad \text{[assumption on small } \varepsilon\text{]} \tag{70}$$

$$< P\left(\sigma_\omega^2 \left\|K_\infty^l - K_t^l\right\|_\infty > \frac{\varepsilon}{2\beta}\right) \qquad \left[\mathcal{A} \text{ is } \sigma_\omega^2\text{-Lipshitz}\right] \tag{71}$$

$$= P\left(\left\|K_\infty^l - K_t^l\right\|_\infty > \frac{\varepsilon}{2\sigma_\omega^2 \beta}\right) < \frac{\delta}{3}. \qquad \text{[Equation 67]} \tag{72}$$

Finally, denote

$$[V(t)]_{\alpha,\alpha'}(x,x') \equiv \left\{\left|\left[\mathcal{C} \circ \mathcal{A}\left(K_t^l\right)\right]_{\alpha,\alpha'}(x,x') - \left[K_t^{l+1}\right]_{\alpha,\alpha'}(x,x')\right| > \frac{\varepsilon}{2}\right\}, \tag{73}$$

i.e. the event that the inequality inside the braces holds. The fact

$$\left\{\left\|\mathcal{C} \circ \mathcal{A}\left(K_t^l\right) - K_t^{l+1}\right\|_\infty > \frac{\varepsilon}{2}\right\} \subseteq U(t)^c \bigcup \left(\bigcup_{x,x',\alpha,\alpha'} [V(t)]_{\alpha,\alpha'}(x,x') \bigcap U(t)\right)$$

implies

$$P\left(\left\{\left\|\mathcal{C} \circ \mathcal{A}\left(K_t^l\right) - K^{l+1}(t)\right\|_\infty > \frac{\varepsilon}{2}\right\}\right) \leq \frac{\delta}{3} + |\mathcal{X}|^2 d^2 \max_{x,x',\alpha,\alpha'} P\left([V(t)]_{\alpha,\alpha'}(x,x') \cap U(t)\right), \tag{74}$$

where the maximum is taken over all $(x,x',\alpha,\alpha') \in \mathcal{X}^2 \times [d]$.

Consider a fixed $\kappa \in \text{PSD}_{|\mathcal{X}|d}$, and define

$$\Sigma(\kappa,\alpha,\alpha',x,x') \equiv \begin{pmatrix} [\mathcal{A}(\kappa)]_{\alpha,\alpha}(x,x) & [\mathcal{A}(\kappa)]_{\alpha,\alpha'}(x,x') \\ [\mathcal{A}(\kappa)]_{\alpha',\alpha}(x',x) & [\mathcal{A}(\kappa)]_{\alpha',\alpha'}(x',x') \end{pmatrix}, \tag{75}$$

a deterministic matrix in $\text{PSD}_2$. Then

$$[\mathcal{C} \circ \mathcal{A}(\kappa)]_{\alpha,\alpha'}(x,x') = \mathcal{T}_\infty\left(\Sigma(\kappa,\alpha,\alpha',x,x')\right), \tag{76}$$

and, conditioned on $K_t^l$,

$$\left[K_t^{l+1} | K_t^l = \kappa\right]_{\alpha,\alpha'}(x,x') = \frac{1}{n^{l+1}(t)} \sum_{i=1}^{n^{l+1}(t)} \phi\left(z_{i,\alpha}^{l,t}(x)\right) \phi\left(z_{i,\alpha'}^{l,t}(x')\right), \tag{77}$$

where $\left\{\left(z_{i,\alpha}^{l,t}(x), z_{i,\alpha'}^{l,t}(x')\right) \Big| K_t^l = \kappa\right\}_{i \in [n^{l+1}(t)]}$ are i.i.d. $\sim \mathcal{N}(0, \Sigma(\kappa,\alpha,\alpha',x,x'))$.

Then if $\Sigma(\kappa,\alpha,\alpha',x,x') \in \text{PSD}_2(R)$ we can apply property 3 of $\Psi$ to conclude that:

$$P\left(\left([V(t)]_{\alpha,\alpha'}(x,x') \cap U(t)\right) \Big| K_t^l = \kappa\right) < \rho_{n^{l+1}(t)}\left(\phi, R, \frac{\varepsilon}{2}\right). \tag{78}$$

However, if $\Sigma(\kappa,\alpha,\alpha',x,x') \notin \text{PSD}_2(R)$, then necessarily $\mathcal{A}(\kappa) \notin \text{PSD}_{|\mathcal{X}|d}(R)$ (since $\Sigma(\kappa,\alpha,\alpha',x,x') \in \text{PSD}_2$), ensuring that $P\left(U(t) | K_t^l = \kappa\right) = 0$. Therefore Equation 78 holds for any $\kappa \in \text{PSD}_{|\mathcal{X}|d}$, and for any $(x,x',\alpha,\alpha') \in \mathcal{X}^2 \times [d]^2$.

We further remark that $\rho_{n^{l+1}(t)}\left(\phi, R, \frac{\varepsilon}{2}\right)$ is deterministic and does not depend on $(\kappa, x, x', \alpha, \alpha')$. Marginalizing out $K_t^l$ and maximizing over $(x, x', \alpha, \alpha')$ in Equation 78 we conclude that

$$\max_{x,x',\alpha,\alpha'} P\left([V(t)]_{\alpha,\alpha'}(x, x') \cap U(t)\right) < \rho_{n^{l+1}(t)}\left(\phi, R, \frac{\varepsilon}{2}\right). \tag{79}$$

Since $\rho_n\left(\phi, R, \frac{\varepsilon}{2}\right) \to 0$ as $n \to \infty$, there exists $n$ such that for any $n^{l+1}(t) \geq n$,

$$\max_{x,x',\alpha,\alpha'} P\left([V(t)]_{\alpha,\alpha'}(x, x') \cap U(t)\right) < \rho_{n^{l+1}(t)}\left(\phi, R, \frac{\varepsilon}{2}\right) \leq \frac{\delta}{3\left|\mathcal{X}\right|^2 d^2}, \tag{80}$$

and, substituting this bound in Equation 74,

$$P\left(\left\{\left\|\mathcal{C} \circ \mathcal{A}\left(K_t^l\right) - K_t^{l+1}\right\|_\infty > \frac{\varepsilon}{2}\right\} \cap U(t)\right) < \frac{2\delta}{3}. \tag{81}$$

Therefore we just need to choose $t^{l+1} > t^l$ so that $n^{l+1}(t) \geq n$ for all $t > t^{l+1}$. □

**Remark E.3.** We list some directions to strengthen / extend the results of Theorem E.5 (and thus Theorem E.6) using the above framework.

1. Consider stacking a deterministic channel-wise linear operator right after the convolutional layer. Again, strided convolution, convolution with no (valid) padding and (non-global) average pooling are particular examples of this category. Let $B \in \mathbb{R}^{d' \times d}$ denote a linear operator. Then the recurrent formula between two consecutive layers is

$$K_\infty^{l+1} = \mathcal{C} \circ \mathcal{B} \circ \mathcal{A}(K_\infty^l) \tag{82}$$

   where $\mathcal{B}$ is the linear operator on the covariance matrix induced by $B$. Conditioned on $K_t^l$, since the outputs after applying the linear operator $B$ are still i.i.d. Gaussian (and the property 3 is applicable), the analysis in the above proof can carry over with $\mathcal{A}$ replaced by $\mathcal{B} \circ \mathcal{A}$.

2. More generally, one may consider inserting an operator $\mathfrak{op}$ (e.g. max-pooling, dropout and more interestingly, normalization) in some hidden layer.

3. Gaussian prior on weights and biases might be relaxed to sub-Gaussian.

### E.7 PROOF OF THEOREM E.4

Note that absolutely continuous exponentially bounded functions contain all polynomials, and are closed under multiplication and integration in the sense that for any constant $C$ the function

$$\int_0^x \phi(t)dt + C \tag{83}$$

is also exponentially bounded. Theorem E.4 is a consequence of the following lemma.

**Lemma E.7.** The following is true:

1. for $k \geq 1$, $C_k(\phi, R) < \infty$ if $\phi$ is exponentially bounded.

2. $\phi \in \Psi_2$ if $\phi'$ exists a.e. and is exponentially bounded.

3. $\phi \in \Psi_3$ if $C_4(\phi, R) < \infty$.

Indeed, if $\phi$ is absolutely continuous and $\phi'$ is exponentially bounded, then $\phi$ is also exponentially bounded. By the above lemma, $\phi \in \Psi$.

*Proof of Lemma E.7.* **1.** We prove the first statement. Assume $|\phi(x)| \leq ae^{b|x|}$.

$$\mathbb{E}_{x \sim \mathcal{N}(0,r)} |\phi(x)|^k = \mathbb{E}_{x \sim \mathcal{N}(0,1)} \left|\phi\left(\sqrt{r}x\right)\right|^k \leq \mathbb{E}_{x \sim \mathcal{N}(0,1)} \left|ae^{\sqrt{r}b|\mathcal{X}|}\right|^k \leq 2a^k e^{k^2 b^2 r/2}. \tag{84}$$

Thus

$$C_k(\phi, R) = \sup_{1/R \le r \le R} \mathbb{E}_{x \sim \mathcal{N}(0,r)} |\phi(x)|^k \le 2a^k e^{k^2 b^2 R/2}. \tag{85}$$

**2.** To prove the second statement, let $\Sigma, \Sigma' \in \mathrm{PSD}_2(R)$ and define $A$ (similarly for $A'$):

$$A \equiv \begin{pmatrix} \sqrt{\Sigma_{11}} & 0 \\ \frac{\Sigma_{12}}{\sqrt{\Sigma_{11}}} & \sqrt{\frac{\Sigma_{22}\Sigma_{11} - \Sigma_{12}^2}{\Sigma_{11}}} \end{pmatrix}. \tag{86}$$

Then $AA^T = \Sigma$ (and $A'A'^T = \Sigma'$). Let

$$A(t) \equiv (1-t)A + tA', \quad t \in [0,1] \tag{87}$$

and

$$f(w) \equiv \phi(x)\phi(y) \quad \text{where} \quad w \equiv (x,y)^T. \tag{88}$$

Since $\phi'$ is exponentially bounded, $\phi$ is also exponentially bounded due to being absolutely continuous. In addition, $p(\|w\|_2) \|\nabla f(w)\|_2$ is exponentially bounded for any polynomial $p(\|w\|_2)$.

Applying the Mean Value Theorem (we use the notation $\lesssim$ to hide the dependence on $R$ and other absolute constants)

$$|\mathcal{T}_\infty(\Sigma) - \mathcal{T}_\infty(\Sigma')| = \frac{1}{2\pi} \left| \int (f(Aw) - f(A'w)) \exp\left(-\|w\|_2^2/2\right) dw \right| \tag{89}$$

$$= \frac{1}{2\pi} \left| \int \int\int_{[0,1]} (\nabla f(A(t)w))((A'-A)w) \exp\left(-\|w\|_2^2/2\right) dt dw \right| \tag{90}$$

$$\lesssim \int_{[0,1]} \int \|(A'-A)w\|_2 \|\nabla f(A(t)w)\|_2 \exp\left(-\|w\|_2^2/2\right) dw dt \tag{91}$$

$$\le \int_{[0,1]} \int \|A'-A\|_{\mathrm{op}} \|w\|_2 \|\nabla f(A(t)w)\|_2 \exp\left(-\|w\|_2^2/2\right) dw dt. \tag{92}$$

Note that the operator norm is bounded by the infinity norm (up to a multiplicity constant) and $\|w\|_2 \|\nabla f(A(t)w)\|_2$ is exponentially bounded. There is a constant $a$ (hidden in $\lesssim$) and $b$ such that the above is bounded by

$$\int_{[0,1]} \int \|A'-A\|_\infty \exp(b\|A(t)\|_\infty \|w\|_2) \exp\left(-\|w\|_2^2/2\right) dw dt \tag{93}$$

$$\lesssim \|A'-A\|_\infty \int_{[0,1]} \int \exp\left(b\sqrt{R}\|w\|_2 - \|w\|_2^2/2\right) dw dt \tag{94}$$

$$\lesssim \|A'-A\|_\infty \tag{95}$$

$$\lesssim \|\Sigma' - \Sigma\|_\infty. \tag{96}$$

Here we have applied the facts

$$\|A'-A\|_\infty \lesssim \|\Sigma - \Sigma'\|_\infty \quad \text{and} \quad \|A(t)\|_\infty \le \sqrt{R}. \tag{97}$$

**3.** Chebyshev's inequality implies

$$P\left( \left| \frac{1}{n} \sum_{i=1}^n \phi(x_i)\phi(y_i) - \mathcal{T}_\infty(\Sigma) \right| > \varepsilon \right) \tag{98}$$

$$\le \frac{1}{n\epsilon^2} \mathrm{Var}(\phi(x_i)\phi(y_i)) \le \frac{1}{n\epsilon^2} \mathbb{E} |\phi(x_i)\phi(y_i)|^2 \tag{99}$$

$$\le \frac{1}{n\epsilon^2} C_4(\phi, R) \to 0 \quad \text{as} \quad n \to \infty. \tag{100}$$

$$\square$$

**Remark E.4.** In practice the $1/n$ decay bound obtained by Chebyshev's inequality in Equation 100 is often too weak to be useful. However, if $\phi$ is linearly bounded, then one can obtain an exponential decay bound via the following concentration inequality:

**Lemma E.8.** If $|\phi(x)| \leq a + b|x|$ a.e., then there is an absolute constant $c > 0$ and a constant $\kappa = \kappa(a, b, R) > 0$ such that property 3 (Equation 48) holds with

$$\rho_n(\phi, \epsilon, R) = 2 \exp\left(-c \min\left\{\frac{n^2 \varepsilon^2}{(2\kappa)^2}, \frac{n\varepsilon}{2\kappa}\right\}\right). \tag{101}$$

*Proof.* We postpone the proof of the following claim.

**Claim E.9.** Assume $|\phi(x)| \leq a + b|x|$. Then there is a $\kappa = \kappa(a, b, R)$ such that for all $\Sigma \in \mathrm{PSD}_2(R)$ and all $p \geq 1$,

$$\left(\mathbb{E}_{(x,y)\sim\mathcal{N}(0,\Sigma)}|\phi(x)\phi(y)|^p\right)^{1/p} \leq \kappa p. \tag{102}$$

Claim E.9 and the triangle inequality imply

$$\left(\mathbb{E}_{(x,y)\sim\mathcal{N}(0,\Sigma)}|\phi(x)\phi(y) - \mathbb{E}\phi(x)\phi(y)|^p\right)^{1/p} \leq 2\kappa p. \tag{103}$$

We can apply Bernstein-type inequality (Vershynin, 2010, Lemma 5.16) to conclude that there is a $c > 0$ such that for every $\Sigma \in \mathrm{PSD}_2(R)$ and any $\{(x_i, y_i)\}_{i=1}^n$ i.i.d. $\sim \mathcal{N}(0, \Sigma)$

$$P\left(\left|\frac{1}{n}\sum_{i=1}^n \phi(x_i)\phi(y_i) - \mathcal{T}_\infty(\Sigma)\right| > \varepsilon\right) \leq 2 \exp\left(-c \min\left\{\frac{n^2\varepsilon^2}{(2\kappa)^2}, \frac{n\varepsilon}{2\kappa}\right\}\right). \tag{104}$$

It remains to prove Claim E.9. For $p \geq 1$,

$$\left(\mathbb{E}_{(x,y)\sim\mathcal{N}(0,\Sigma)}|\phi(x)\phi(y)|^p\right)^{1/p} \leq \left(\mathbb{E}_{x\sim\mathcal{N}(0,\Sigma_{11})}|\phi(x)|^{2p}\right)^{1/2p}\left(\mathbb{E}_{y\sim\mathcal{N}(0,\Sigma_{22})}|\phi(y)|^{2p}\right)^{1/2p} \tag{105}$$

$$\leq \left(a + b\left(\mathbb{E}|x|^{2p}\right)^{1/2p}\right)\left(a + b\left(\mathbb{E}|y|^{2p}\right)^{1/2p}\right) \tag{106}$$

$$\leq \left(a + b\sqrt{R}\left(\mathbb{E}_{u\sim\mathcal{N}(0,1)}|u|^{2p}\right)^{1/2p}\right)^2 \tag{107}$$

$$\leq \left(a + b\sqrt{R}\left(c'^{2p}p^p\right)^{1/2p}\right)^2 \tag{108}$$

$$\leq \left(a + bc'^2\sqrt{R}\right)^2 p \tag{109}$$

$$\equiv \kappa p. \tag{110}$$

We applied Cauchy-Schwarz' inequality in the first inequality, the triangle inequality in the second one, the fact $\Sigma_{11}, \Sigma_{22} \leq R$ in the third one, absolute moments estimate of standard Gaussian in the fourth one, where $c'$ is a constant such that

$$\left(\mathbb{E}_{u\sim\mathcal{N}(0,1)}|u|^p\right)^{1/p} \leq c'\sqrt{p}. \tag{111}$$

$\square$

# F  GLOSSARY

We use the following shorthands in this work:

1. NN - neural network;

2. CNN - convolutional neural network;

3. LCN - locally connected network, a.k.a. convolutional network without weight sharing;

4. FCN - fully connected network, a.k.a. multilayer perceptron (MLP);

5. GP - Gaussian process;

6. X-GP - a GP equivalent to a Bayesian infinitely wide neural network of architecture X (§2).

7. MC-(X-)-GP - a Monte Carlo estimate (§4) of the X-GP.

8. Width, (number of) filters, (number of) channels represent the same property for CNNs and LCNs.

9. Pooling - referring to architectures as "with" or "without pooling" means having a single global average pooling layer (collapsing the spatial dimensions of the activations $y^{L+1}$) before the final linear FC layer giving the regression outputs $z^{L+1}$.

10. Invariance and equivariance are always discussed w.r.t. translations in the spatial dimensions of the inputs.

## G    EXPERIMENTAL SETUP

Throughout this work we only consider $3 \times 3$ (possibly unshared) convolutional filters with stride $1$ and no dilation.

All inputs are normalized to have zero mean and unit variance, i.e. lie on the $d-$dimensional sphere of radius $\sqrt{d}$, where $d$ is the total dimensionality of the input.

All labels are treated as regression targets with zero mean. i.e. for a single-class classification problem with $C$ classes targets are $C-$dimensional vectors with $-1/C$ and $(C - 1)/C$ entries in incorrect and correct class indices respectively.

If a subset of a full dataset is considered for computational reasons, it is randomly selected in a balanced fashion, i.e. with each class having an equal number of samples. No data augmentation is used.

All experiments were implemented in Tensorflow (Abadi et al., 2016) and executed with the help of Vizier (Golovin et al., 2017).

All neural networks are trained using Adam (Kingma & Ba, 2015) minimizing the mean squared error loss.

### G.1    MANY-CHANNEL CNNS AND LCNS

Relevant Figures: 6 (b), 8, 9.

We use a training and validation subsets of CIFAR10 of sizes $500$ and $4000$ respectively. All images are bilinearly downsampled to $8 \times 8$ pixels.

All models have 3 hidden layers with an $\mathrm{erf}$ nonlinearity. No (valid) padding is used.

Weight and bias variances are set to $\sigma_\omega^2 \approx 1.7562$ and $\sigma_b^2 \approx 0.1841$, corresponding to the pre-activation variance fixed point $q^* = 1$ (Poole et al., 2016) for the $\mathrm{erf}$ nonlinearity.

NN training proceeds for $2^{19}$ gradient updates, but aborts if no progress on training loss is observed for the last 100 epochs. If the training loss does not reduce by at least $10^{-4}$ for 20 epochs, the learning rate is divided by 10.

All computations are done with 32-bit precision.

The following NN parameters are considered:[10]

1. Architecture: CNN or LCN.

2. Pooling: no pooling or a single global average pooling (averaging over spatial dimensions) before the final FC layer.

3. Number of channels: $2^k$ for $k$ from 0 to 12.

4. Initial learning rate: $10^{-k}$ for $k$ from 0 to 15.

5. Weight decay: $0$ and $10^{-k}$ for $k$ from 0 to 8.

6. Batch size: 10, 25, 50, 100, 200.

---

[10]Due to time and memory limitations certain large configurations could not be evaluated. We believe this did not impact the results of this work in a qualitative way.

For NNs, all models are filtered to only $100\%$-accurate ones on the training set and then for each configuration of {architecture, pooling, number of channels} the model with the lowest validation loss is selected among the configurations of {learning rate, weight decay, batch size}.

For GPs, the same CNN-GP is plotted against CNN and LCN networks without pooling. For LCN with pooling, inference was done with an appropriately rescaled CNN-GP kernel, i.e. $\left(\mathcal{K}^{\text{vec}}_\infty - \sigma_b^2\right)/d + \sigma_b^2$, where $d$ is the spatial size of the penultimate layer. For CNNs with pooling, a Monte Carlo estimate was computed (see §4) with $n = 2^{12}$ filters and $M = 2^6$ samples.

For GP inference, the initial diagonal regularization term applied to the training convariance matrix is $10^{-10}$; if the cholesky decompisition fails, the regularization term is increased by a factor of 10 until it either succeeeds or reaches the value of $10^5$, at which point the trial is considered to have failed.

### G.2 Monte Carlo Evaluation of Intractable GP Kernels

Relevant Figures: 5, 10.

We use the same setup as in §G.1, but training and validation sets of sizes $2000$ and $4000$ respectively.

For MC-GPs we consider the number of channels $n$ (width in FCN setting) and number of NN instantiations $M$ to accept values of $2^k$ for $k$ from 0 to 10.

Kernel distance is computed as:

$$\frac{\|\mathcal{K}_\infty - K_{n,M}\|_F^2}{\|\mathcal{K}_\infty\|_F^2},\tag{112}$$

where $\mathcal{K}_\infty$ is substituted with $K_{2^{10},2^{10}}$ for the CNN-GP pooling case (due to impracticality of computing the exact $\mathcal{K}^{\text{pool}}_\infty$). GPs are regularized in the same fashion as in §G.1, but the regularization factor starts at $10^{-4}$ and ends at $10^{10}$ and is multiplied by the mean of the training covariance diagonal.

### G.3 Transforming a GP over spatial locations into a GP over classes

Relevant Figure: 4.

We use the same setup as in §G.2, but rescale the input images to size of $31 \times 31$, so that at depth 15 the spatial dimension collapses to a $1 \times 1$ patch if no padding is used (hence the curve of the CNN-GP without padding halting at that depth).

For MC-CNN-GP with pooling, we use samples of networks with $n = 16$ filters. Due to computational complexity we only consider depths up to 31 for this architecture. The number of samples $M$ was selected independently for each depth among $\left\{2^k\right\}$ for $k$ from 0 to 15 to maximize the validation accuracy on a separate $500$-points validation set. This allowed us to avoid the poor conditioning of the kernel. GPs are regularized in the same fashion as in §G.1, but for MLP-GP the multiplicative factor starts at $10^{-4}$ and ends at $10^{10}$.

### G.4 Relationship to Deep Signal Propagation

Relevant Figure: 11.

We use a training and validation subsets of CIFAR10 of sizes $500$ and $1000$ respectively.

We use the erf nonlinearity. For CNN-GP, images are zero-padded (same padding) to maintain the spatial shape of the activations as they are propagated through the network.

Weight and bias variances (horizontal axis $\sigma_\omega^2$ and vertical axis $\sigma_b^2$ respectively) are sampled from a uniform grid of size $50 \times 50$ on the range $[0.1, 5] \times [0, 2]$ including the endpoints.

All computations are done with 64-bit precision. GPs are regularized in the same fashion as in §G.1, but the regularization factor is multiplied by the mean of the training covariance diagonal. If the experiment fails due to numerical reasons, $0.1$ (random chance) validation accuracy is reported.

## G.5    CNN-GP on full datasets

Relevant Figures 6 (a, c), 7 and Table: 1.

We use full training, validation, and test sets of sizes 50000, 10000, and 10000 respectively for MNIST (LeCun et al., 1998) and Fashion-MNIST (Xiao et al., 2017a), 45000, 5000, and 10000 for CIFAR10 (Krizhevsky, 2009). We use validation accuracy to select the best configuration for each model (we do not retrain on validation sets).

GPs are computed with 64-bit precision, and NNs are trained with 32-bit precision. GPs are regularized in the same fashion as in §G.4.

Zero-padding (same) is used.

The following parameters are considered:

1. Architecture: CNN or FCN.
2. Nonlinearity: erf or ReLU.
3. Depth: $2^k$ for $k$ from 0 to 4 (and up to $2^5$ for MNIST and Fashion-MNIST datasets).
4. Weight and bias variances. For erf: $q^*$ from $\{0.1, 1, 2, \dots, 8\}$. For ReLU: a fixed weight variance $\sigma_\omega^2 = 2 + 4e^{-16}$ and bias variance $\sigma_b^2$ from $\{0.1, 1, 2, \dots, 8\}$.

On CIFAR10, we additionally train NNs for $2^{18}$ gradient updates with a batch size of 128 with corresponding parameters in addition to[11]

1. Pooling: no pooling or a single global average pooling (averaging over spatial dimensions) before the final FC layer (only for CNNs).
2. Number of channels or width: $2^k$ for $k$ from 1 to 9 (and up to $2^{10}$ for CNNs with pooling in Figure 6, a).
3. Learning rate: $10^{-k} \times 2^{16} / (\text{width} \times q^*)$ for $k$ from 5 to 9, where width is substituted with the number of channels for CNNs and $q^*$ is substituted with $\sigma_b^2$ for ReLU networks. "Small learning rate" in Table 1 refers to $k \in \{8, 9\}$.
4. Weight decay: 0 and $10^{-k}$ for $k$ from 0 to 5.

For NNs, all models are filtered to only 100%-accurate ones on the training set (expect for values in parentheses in Table 1). The reported values are then reported for models that achieve the best validation accuracy.

## G.6    Model comparison on CIFAR10

Relevant Figure: 1.

We use the complete CIFAR10 dataset as described in §G.5 and consider 8-layer ReLU models with weight and bias variances of $\sigma_\omega^2 = 2$ and $\sigma_b^2 = 0.01$. The number of channels / width is set to $2^5$, $2^{10}$ and $2^{12}$ for LCN, CNN, and FCN respectively.

GPs are computed with 64-bit precision, and NNs are trained with 32-bit precision.

No padding (valid) is used.

NN training proceeds for $2^{18}$ gradient updates with batch size 64, but aborts if no progress on training loss is observed for the last 10 epochs. If the training loss does not reduce by at least $10^{-4}$ for 2 epochs, the learning rate is divided by 10.

Values for NNs are reported for the best validation accuracy over different learning rates ($10^{-k}$ for $k$ from 2 to 12) and weight decay values (0 and $10^{-k}$ for $k$ from 2 to 7). For GPs, validation accuracy is maximized over initial diagonal regularization terms applied to the training convariance matrix: $10^{-k} \times$ [mean of the diagonal] for $k$ among 2, 4 and 9 (if the cholesky decomposition fails, the regularization term is increased by a factor of 10 until it succeeds or $k$ reaches the value of 10).

---

[11]Due to time and compute limitations certain large configurations could not be evaluated. We believe this did not impact the results of this work in a qualitative way.

