# OpenReview forum: "Bayesian Deep Convolutional Networks with Many Channels are Gaussian Processes"
_ICLR.cc/2019/Conference_

### Official Review · AnonReviewer2 · 2018-10-30

**Rating:** 6
**Confidence:** 4

**Review:**

This paper extends the recent results concerning GP equivalence of infinitely wide FC nets to the convolutional case. This paper is generally of a high quality (notwithstanding the lack of keys on figures) and provides insights to an important class of model. I recommend that this paper be accepted, but I think it could be improved in a few ways.

Firstly, and rather mundanely: the figures. Fig 1 is not easy to read due to the density of plotting, and as there is no key it isn’t possible to tell what it shows. Figure 2 is rather is called a ‘graphical model’ but the variables (weights and biases) are not shown. It should be specified that this is the graphical model of the infinite limit, in which case the K variables should not be random. Also, the caption on this figure refers to variables that aren’t in the figure, and is grammatically incorrect (perhaps something like ‘the limit of an infinitely wide convolutional’ is missing?). Figure 3 has a caption which seems to be inconsistent with the coloring (for example green is center pixel in the text, but blue in the key). Figure 6 is also missing a key. In Figure 5, what does the tick symbol denote? Finally, the value some of Table 1 is questionable as so many entries are missing. For example, the Fashion-MNIST column has only two values, which seems to me of little use. [I would have given the paper a rating of 7 were it not for these issues]

Regarding the presentation of the content, I found this paper generally easy to follow and the arguments sound. Here are few points:

There is an important distinction between finite width Bayesian-CNNs and the infinite limit, and this distinction is indeed made in the paper but not clearly enough in my view. I would anticipate that some readers might come away after a cursory reading thinking that Bayesian-CNNs are fundamentally worse than their parametric counterparts, but this is emphatically not the message of the paper. It seems that the infinite limit that is the cause of two problems. The first problem (or perhaps benefit) is that the infinite limit gives Gaussian inner layers, just as in the fully connected case. The second problem (and I’d say this is definitely a problem this time) is that the infinite limit loses the covariance between the pixels, at least with a fully connected final layer. I would recall [Matthews 2018, long version] section 7, which discusses that point that taking the infinite limit in the fully connected is actually potentially undesirable. To quote Matthews 2018, “MacKay (2002, p. 547) famously reflected on what is lost when taking the Gaussian process limit of a single hidden layer network, remarking that Gaussian processes will not learn hidden features”. Some discussion of this would enhance the presented paper, in my view.

The discussion of eq (7) could be made more clear. Eq (7) is only defined on K, and not in composition with A. It is important that the alpha dependency is preserved by the A operation, and while I suppose this is obvious I would welcome a bit more detail. It would help to demonstrate the application of the results of [Cho and Saul 2009] to the convolution case explicitly (i.e. for C o A), in my view.

Regarding results, effort has clearly gone to keep the comparisons as fair as possible, but with these large datasets it is difficult to disentangle the many factors that might effect performance (as acknowledged on p9). It is a weakness of the paper that there is no toy example. An example demonstrating a situation which can only be solved with hierarchical features (e.g. features that are larger than the receptive field of a single layer) would be particularly interesting, as in this case I think the GP-CNN would fail, even with the average pooling, whereas the finite Bayesian-CNN would succeed (with a sufficiently accurate inference method).

It would improve readability to stress the 1D notation in the main text rather than in a footnote. On first reading I missed this detail and was confused as I was trying to interpret everything as a 2D convolution. On reflection I think notation is used in the paper is good, but I think the generalization to 2D should be elevated to something more than the footnote. Perhaps a paragraph explaining how the 2D case works would be appropriate, especially as all the experiments are in 2D cases.

Some further smaller points on specific [section, paragraph, line]s

1,2,4 I think ‘easily’ is a bit of an overstatement. In this work the kernel is itself defined via a recursive convolutional operation, which doesn’t seem to me much more interpretable than the parametric convolution. At least the filters can be examined in parametric case, which isn’t the case here. I do agree with the sentiment that a function prior is better than an implicit weight prior, however.

1,2,-1 This seems too vague to me, as at least to some extent, Matthew 2018 did indeed consider using NN-GPs to gain insight about equivalent NN models (e.g. section 5.3)

1.1,:,: I find it very surprising that there are no references to Cho and Saul 2009 in this section (one does appear in 2.2.2, however).

1.1,3,-2:-1 ‘Our work differs from all of these in that our GP corresponds exactly to a fully Bayesian CNN in the many channel limit’ I do not think this is completely true, as the deep convolution GP does correspond to an infinite limit of a Bayesian CNN, just not the same limit as the one taken in this paper. Similarly a DGP following the Danianou and Lawrence 2013 is an infinite limit of a NN, but one with bottlenecks between layers. It is important that readers appreciate that infinite limits can be taken in different ways, and the resulting models may be very different. This certain limit taken in this work has desirable computational properties, but arguably undesirable modelling implications.

1.1,-1,-2 It should be made more clear here that the SGD trained models are non-Bayesian.

Figure 3 The MC-CNN-GP appears to have performance that is nearly independent of the depth, even including 1 layer. Could this be explained?

2.2,2,: The z^l variables are zero mean Gaussian with a fixed covariance, not delta functions, as I understand it. They are independent of each other due to the deterministic K^l, certainly, but they are not themselves deterministic. Could this be clarified?

---

> ### Author Response · Authors · 2018-11-27
> **4/4 AnonReviewer2 Reply**
>
> ------------------------------------------------------------------------------------
> >>> 1,2,-1 This seems too vague to me, as at least to some extent, Matthew 2018 did indeed consider using NN-GPs to gain insight about equivalent NN models (e.g. section 5.3)
>
> To our best knowledge, we are the first to learn the role of architecture on the functions represented with the networks using NN-GP correspondence. Specifically, in our Table 1 (former 2), we disentangle the role of network topology and equivariance in CNNs. Previous works (both Lee et al 2018 and Matthews et al 2018) focused on establishing the correspondence and understanding the properties of the corresponding GPs. It would be helpful if you could clarify specific insights laid out in Matthews et al 2018 (section 5.3). As far as we can tell the section discusses how the Bayesian NNs would match NN-GPs.
>
> ------------------------------------------------------------------------------------
> >>> 1.1,:,: I find it very surprising that there are no references to Cho and Saul 2009 in this section (one does appear in 2.2.2, however).
>
> We have updated the related work section to address this point.
>
> ------------------------------------------------------------------------------------
> >>> 1.1,3,-2:-1 ‘Our work differs from all of these in that our GP corresponds exactly to a fully Bayesian CNN in the many channel limit’ I do not think this is completely true, as the deep convolution GP does correspond to an infinite limit of a Bayesian CNN, just not the same limit as the one taken in this paper. Similarly a DGP following the Danianou and Lawrence 2013 is an infinite limit of a NN, but one with bottlenecks between layers. It is important that readers appreciate that infinite limits can be taken in different ways, and the resulting models may be very different.
>
> Thank you for the comment, we have modified the text in this section to emphasize this distinction more strongly. However, this seems mostly to be a question of semantics in using the term “infinite limit”, i.e. whether one means to include or exclude bottleneck layers. We wish to point out, though, that the limit we take is nevertheless interesting and likely relevant to networks in which all layers widths are similarly large, which is arguably a rather large class of models used in the wild.
>
> ------------------------------------------------------------------------------------
> >>> This certain limit taken in this work has desirable computational properties, but arguably undesirable modelling implications.
>
> Good point, and we now emphasize this in the text as well.
>
> ------------------------------------------------------------------------------------
> >>> 1.1,-1,-2 It should be made more clear here that the SGD trained models are non-Bayesian.
>
> Done.
>
> ------------------------------------------------------------------------------------
> >>> Figure 3 The MC-CNN-GP appears to have performance that is nearly independent of the depth, even including 1 layer. Could this be explained?
>
> We believe there are two factors at play. Firstly, to the best of our knowledge, dependence of model (be it NN-GPs or SGD-trained NNs) performance on depth is poorly understood and is difficult to decouple (if at all possible) from the particular dataset and architectural decisions like pooling or residual connections. Therefore, we do not necessarily find the lack of a clear and interpretable dependence surprising. Secondly, performance of MC-GP is subject to approximation noise and bias (we only used 16 filters, see Appendix A.7.3) as well as poor conditioning (see dark bands in Figures 2 and 7 in the new revision). Therefore we conjecture that there could be a similar underlying depth dependence to the one observed in other curves on the plot in Figure 1 (former 3), yet it is mild enough (just like other curves don’t have steep slopes as well) to be overrun with the MC approximation imperfections.
>
> ------------------------------------------------------------------------------------
> >>> 2.2,2,: The z^l variables are zero mean Gaussian with a fixed covariance, not delta functions, as I understand it. They are independent of each other due to the deterministic K^l, certainly, but they are not themselves deterministic. Could this be clarified?
>
> You are correct and we have edited the text to clarify this fact.

---

> ### Author Response · Authors · 2018-11-27
> **3/4 AnonReviewer2 Reply**
>
> ------------------------------------------------------------------------------------
> >>> Regarding results, effort has clearly gone to keep the comparisons as fair as possible, but with these large datasets it is difficult to disentangle the many factors that might effect performance (as acknowledged on p9). It is a weakness of the paper that there is no toy example. An example demonstrating a situation which can only be solved with hierarchical features (e.g. features that are larger than the receptive field of a single layer) would be particularly interesting, as in this case I think the GP-CNN would fail, even with the average pooling, whereas the finite Bayesian-CNN would succeed (with a sufficiently accurate inference method).
>
> Thank you for your suggestion. To the best of our knowledge, while (as you have referenced earlier) arguments have been made in the literature against GPs due to lack of hierarchical representation learning present in CNNs (Matthews et al, 2018, section 7), (MacKay 2003, section 45.7), (Neal 1996, Chapter 5) the practical impact of these assertions in a supervised regression setting has not been carefully investigated empirically or theoretically. Moreover, it is unclear if these beliefs hold if we use a sufficiently powerful class of kernels, and we explicitly construct such a class in our work. Further, we believe it is important to decouple hierarchy and finite representations in this discussion. NN-GPs do have a hierarchical kernel, and CNN-GPs have a spatially-local hierarchical kernel with a receptive field (3x3 per layer in our work) smaller than the input images, and they do end up benefiting from hierarchy significantly (see Figure 1 (former 3); further, the best CNN-GP models in Table 2 (former 1) are at least 8 layers deep). Finally, we highlight the similarity in performance between the best finite SGD-trained fully- and locally-connected networks in our work (Tables 1, 2), Lee et al 2018 (Tables 1, 2), as well as the similarities between small Bayesian NNs and NN-GPs in Matthews et al 2018 (section 5.3). Considering all of the above, we believe construction of meaningful datasets that will decisively disentangle performance of finite-feature models from GPs in the context of regression to be a non-trivial research problem and lie beyond the scope of this work.
>
> ------------------------------------------------------------------------------------
> >>> It would improve readability to stress the 1D notation in the main text rather than in a footnote.
>
> Done, see beginning of section 2.1.
>
> ------------------------------------------------------------------------------------
> >>> On first reading I missed this detail and was confused as I was trying to interpret everything as a 2D convolution. On reflection I think notation is used in the paper is good, but I think the generalization to 2D should be elevated to something more than the footnote. Perhaps a paragraph explaining how the 2D case works would be appropriate, especially as all the experiments are in 2D cases.
>
> Done, see beginning of section 2.1 referencing section A.3 with an added paragraph “ND convolutions” at the end.
>
> ------------------------------------------------------------------------------------
> >>> 1,2,4 I think ‘easily’ is a bit of an overstatement. In this work the kernel is itself defined via a recursive convolutional operation, which doesn’t seem to me much more interpretable than the parametric convolution. At least the filters can be examined in parametric case, which isn’t the case here. I do agree with the sentiment that a function prior is better than an implicit weight prior, however.
>
> Thank you for this comment. Indeed, at that at the moment the kernel definition does not seem easily interpretable; in-depth investigation of its consequences is the subject of future work. We nonetheless think having compact expressions for the computation performed by a NN, both for the prior and posterior, can open up a novel route towards theoretical understanding. Note that examining filters in parametric space, to the best of our knowledge, can only be done after training and not analytically, the prior therefore remaining difficult to analyze. We have removed the word ‘easy’ from the text and added a footnote referencing filter visualization.

---

> ### Author Response · Authors · 2018-11-27
> **2/4 AnonReviewer2 Reply**
>
> ------------------------------------------------------------------------------------
> >>> Finally, the value some of Table 1 is questionable as so many entries are missing. For example, the Fashion-MNIST column has only two values, which seems to me of little use.
>
> Since running huge parameter sweeps for NNs (see Appendix A.7.5) is expensive, we have focused the full suite of experiments on CIFAR10, as the dataset benefiting from convolutional structure the most (among considered), thus allowing to gauge qualitatively the difference between different models (see e.g. Table 1 (former 2)). However, we still ran our (much smaller in the number of hyper-parameters) GP experiments on MNIST and Fashion-MNIST (a very recent dataset, hence no results from other work to report), to position our work among current and future SOTA results.
>
> ------------------------------------------------------------------------------------
> >>> There is an important distinction between finite width Bayesian-CNNs and the infinite limit, and this distinction is indeed made in the paper but not clearly enough in my view. I would anticipate that some readers might come away after a cursory reading thinking that Bayesian-CNNs are fundamentally worse than their parametric counterparts, but this is emphatically not the message of the paper. It seems that the infinite limit that is the cause of two problems. The first problem (or perhaps benefit) is that the infinite limit gives Gaussian inner layers, just as in the fully connected case. The second problem (and I’d say this is definitely a problem this time) is that the infinite limit loses the covariance between the pixels, at least with a fully connected final layer. I would recall [Matthews 2018, long version] section 7, which discusses that point that taking the infinite limit in the fully connected is actually potentially undesirable. To quote Matthews 2018, “MacKay (2002, p. 547) famously reflected on what is lost when taking the Gaussian process limit of a single hidden layer network, remarking that Gaussian processes will not learn hidden features”. Some discussion of this would enhance the presented paper, in my view.
>
> Thank you for your comment and your references. We have added a clear disclaimer at the end of introduction that we make no claims about finite width Bayesian networks and added a footnote 8 to expand the discussion section.
>
> ------------------------------------------------------------------------------------
> >>> The discussion of eq (7) could be made more clear. Eq (7) is only defined on K, and not in composition with A. It is important that the alpha dependency is preserved by the A operation, and while I suppose this is obvious I would welcome a bit more detail. It would help to demonstrate the application of the results of [Cho and Saul 2009] to the convolution case explicitly (i.e. for C o A), in my view.
>
> Thank you for your comment, we have clarified the application of the A operation by a) referencing the specific derivation of Equation (7) in Xiao et al, 2018 (Lemma A.1), b) defining  A’s domain and codomain in section 2.2.1, and c) adding a section 2.1. “Shapes and indexing” to make our matrix/vector notation more precise.

---

> ### Author Response · Authors · 2018-11-27
> **1/4 AnonReviewer2 Reply**
>
> Thank you for the very thorough and insightful review. We are glad you found our research useful. We have adjusted the text to address your suggestions. Please see below our responses to specific comments:
>
> ------------------------------------------------------------------------------------
> >>> Firstly, and rather mundanely: the figures. Fig 1 is not easy to read due to the density of plotting, and as there is no key it isn’t possible to tell what it shows.
> >>> Figure 6 is also missing a key
>
> Thank you for the suggestion. We have added (partial) keys to figures, increased figures axes / ticks / title fonts, and increased the size of Figures 1, 5, 6 (now Figure 3, a, b, c) to make them more legible. Please note that displaying a full key is not practical in Figures 1 and 6 (now Figure 3, a, c) since each line / point respectively corresponds to one of many distinct setting of hyper-parameters like weight and bias variances, non-linearity, and depth. Instead, these plots serve to give a general picture across many different configurations, various styles serving merely to make different entries more visually separate.
>
> ------------------------------------------------------------------------------------
> >>> Figure 2 is rather is called a ‘graphical model’ but the variables (weights and biases) are not shown. It should be specified that this is the graphical model of the infinite limit, in which case the K variables should not be random. Also, the caption on this figure refers to variables that aren’t in the figure, and is grammatically incorrect (perhaps something like ‘the limit of an infinitely wide convolutional’ is missing?).
>
> Please note that the graphical model does represent a finite CNN with random  covariance matrices K^l after marginalizing out the weights and biases, and we believe it to be accurate. Otherwise, we agree that in the infinite channel/width limit the model also remains correct, and covariance matrices K^l indeed become deterministic. However, in the revised version we have replaced the justification in section 2.2 in terms of marginalizing over {K^l} with a more rigorous approach, and this figure no longer appears in the text.
>
> ------------------------------------------------------------------------------------
> >>> Figure 3 has a caption which seems to be inconsistent with the coloring (for example green is center pixel in the text, but blue in the key).
>
> Thank you for noticing this. We have fixed it in the updated version (now Figure 1).
>
> ------------------------------------------------------------------------------------
> >>> In Figure 5, what does the tick symbol denote?
>
> We added keys to the figure to clarify the symbols. All plots share x and y axes where each denote number of channels and accuracy. Note that the x-axis is in log scale. Crosses are displayed at #channel values for which NN experiments were run.

---

> ### Author Response · Authors · 2018-12-05
> **AnonReviewer2 message**
>
> Dear Reviewer,
>
> Thank you again for your very thorough and insightful review. We believe we have effectively implemented most of your very helpful suggestions, by expanding the discussion in the text and improving the clarity of figures and exposition. Both reviewers 1 and 3 have raised their scores on the strength of our rebuttal and paper improvements. We are wondering if you also feel that we have significantly improved our paper, and if so whether you would be willing to increase your score as a result.
>
> Thank you for your consideration!

---

### Official Review · AnonReviewer3 · 2018-11-02

**Rating:** 7
**Confidence:** 5

**Review:**

****Reply to authors' rebuttal****

Dear Authors,

I greatly appreciate the effort you have put into the rebuttal. The changes you have made have addressed most of my concerns and I believe that the few outstanding ones can be fixed without significantly affecting the main message of the paper. I will thus be recommending acceptance of the paper.

Best wishes,
Rev 3


Several remarks on the updated version:

- (p.20, A.5.1) To ensure the random variables are well-defined, please state explicitly which sigma algebra is F (I am assuming the product Borel sigma-algebra + the relevant definitions of the random variables). This is important for the reader to understand what convergence in distribution on this particular space does and does not imply. Some readers might also appreciate if you used the mentioned "infinite width, finite fan-out, networks" (Matthews et al.) construction (or similar) which would ensure that the collection of random variables {z_i^l}_{i \in N*} is well-defined for any network width and l, which currently does not seem to be the case according to Eqs. (28-29). If the full countably infinite vectors of random variables are not defined for all networks in the sequence, it is not possible to prove their convergence in distribution to the relevant GPs.

- (p.21, A.5.3) Thank you for clarifying the definition of elements of the sequential limit. If possible, I would further recommend first fixing the probability space and then defining the random variables (the argument just before Theorem A.2 seems somewhat circular as R.V.s should first be defined on some space, and not put on a probability space post-hoc; perhaps some product space with the product sigma-algebra would work here?!). Furthermore, if I understand correctly, there are now L sequences of neural networks (one sequence for networks with 0, ..., L-1 "infinite layers"), rather than a single sequence, and the "infinite layers" are squashed into a single "infinite layer" which is represented by z_i^\infty? In other words, all the infinite layers are replaced by iid samples from a particular GP and only the finite layers have the standard neural network structure? If I am mistaken (or not), perhaps a further explanatory footnote would help the reader.

- (p.21, A.5.3 & p.23, A.5.4) Thank you for improving the discussion of joint convergence. Please clarify that proving convergence for any finite m is sufficient for proving convergence in distribution of the countably infinite vector {z_i}_{i \in N*} for the **product Borel sigma-algebra** (e.g. using an argument like the one on p.19 of Billingsley (1999)).

- (p.21) "Uniformly square-integrable": to me, this phrase suggests that the collection of squares of the functions has to be uniformly integrable but the definition in Eq. (27) only states one of the conditions in definition of uniform integrability. Please clarify that "uniform square-integrability" here is not related to the standard notion of "uniform integrability" in the literature.





****Summary****

This paper extends recent results on convergence of Bayesian fully connected networks (FCNs) to Gaussian processes (GPs), to the equivalent relationship between convolutional neural networks (CNNs) and GPs. This is currently an area of high interest, with Xiao et al. (2018) examining the same relationship from a mean-field perspective, and two other concurrent papers making contributions:

https://arxiv.org/abs/1808.05587
https://arxiv.org/abs/1810.10798

Thus the scope of the paper fits well within the aims of the conference.

I really appreciate that the authors did not shy away from studying the effect of pooling layers, and find the connection to locally connected networks they describe intriguing and insightful. On the experimental side, the investigation of the relative importance of compositionality, equivariance and invariance on performance of CNNs is very interesting.

These experiments and investigations are however based on a theoretical foundation which suffers from several issues. The main problems are an incorrect proof of convergence of the joint distribution of filters, and an improper use of convergence in probability in cases where random variables do not share a common underlying probability space. Unfortunately, either of these by itself invalidates the main theoretical claims which is why I am recommending rejection of the paper.

However, I believe that the argument in (A.4.3) can potentially be rectified, and, as I detail below, is of greater interest to the community relative to the ones in (A.4.1) and (A.4.2). If this is accomplished and the proofs in (A.4.1) and (A.4.2) are either also fixed or left out (A.4.3 is sufficient to justify the claims in the main body), I am willing to significantly improve my rating of this paper and potentially recommend acceptance. For this reason, a "detailed comments" section is appended at the end of the standard review where the technical issues are described in much greater detail.


****General comments****

**Bayesian vs. infinite neural networks**

The main theoretical claims concerning the relationship between Bayesian CNNs and GPs are within Section 2. Therein on top of page 4, the authors say "In Appendix A.4 we give several **alternative** derivations of the correspondence" (emphasis mine), and then progress to outline the skeleton of the argument (A.4.2) in Sections 2.2.1-2.2.3. Section 2.2.3 is concluded by statement of the main theoretical result of this paper, Eq. (10), which comes from (A.4.3) and can only be linked to the rest of Section 2 through the claim of equivalence between the "alternative derivations" (A.4.1), (A.4.2) and (A.4.3). The problem is that the equivalence claim does not hold, as explained below:

The most important distinction here is between what I will call a "sequential" and a "simultaneous" limit. In the "sequential" case (A.4.1 & A.4.2, Sections 2.2.1-2.2.3), layers are taken to infinity one by one, whereas in the "simultaneous" case (A.4.3, used to obtain the result concluding Section 2.2.3) all layers are **finite** for **all** members of the sequence, growing in width simultaneously.

The "simultaneous" limit (A.4.3) is in my view more interesting as it tells us that **finite** BNNs do indeed converge to GPs in distribution, i.e. that for each expectation of a continuous bounded function of the outputs of the limiting GP, there exists a BNN with a **finite** number of neurons in **each** layer for which the expectation of the same function is arbitrarily close. From a practical perspective, "simultaneous" limit tells us that inference algorithms for BNNs (which can be inaccurate and/or computationally expensive) can sometimes be replaced by exact or approximate inference algorithms for the limiting GP (cf. Section 5 in (Matthews et al., 2018, extended version)).

The "sequential" limit (A.4.1 & A.4.2) on the other hand does not establish existence of finite BNNs arbitrarily close to a particular GP, or justify use of the GP limit as approximation for finite BNNs as above. This is because the width of individual layers goes to infinity in a sequence from first to last. This means that most of the networks that constitute the sequence converging to the GP will have **one or more infinitely wide layers** and thus do not correspond to the finite BNNs we usually work with. In other words, "sequential" limit can only ever establish that there exists a network with **all but the final hidden layer infinite** that is arbitrarily close to the limiting GP. The only case where "sequential" and "simultaneous" limits agree is thus in the single hidden layer case first studied by Neal (1996). I will call the networks with one or more infinite layers "infinite networks", inspired by the work of Williams (1997) and others. Notice that infinite networks cannot be described by Eqs. (1) and (2) as the weights would be zero with probability one and thus output of the network would only depend on biases. It is not immediately obvious how to formally replace Eqs. (1) and (2) in the case of infinite networks which is one of the technical issues with the approaches in (A.4.1) and (A.4.2) (see the detailed comments section for further discussion).

Others may of course disagree and find "sequential" limits more interesting, but if the authors wish to keep the description of (A.4.2) in the main paper (Sections 2.2.1-2.2.3), it would be highly beneficial if readers were given the opportunity to understand the differences between the two types of limits so that they can form their own judgement. The authors should then also make clearer that the approach described in Sections 2.2.1-2.2.3 cannot be used to obtain the final result, Eq. (10). I would rather recommend reworking Sections 2.2.1-2.2.3 based on the "simultaneous" limit argument in (A.4.3) which unlike the current one can justify the result in Eq. (10) stated at the end.


**Other comments**

- (p.2, top) You say your results are "strengthening and extending the result of Matthews et al. (2018)" which is somewhat confusing. Matthews et al. prove a result for FCNs whereas this paper focuses on CNNs. Extension of (A.4.3) to FCNs may well be possible but is not included in this paper. Results in (A.4.1) and (A.4.2) are for the "sequential" whereas Matthews et al. study the "simultaneous" limit. Further differences:
	- Matthews et al. prove convergence for any countable rather than only finite input sets.
	- In Matthews et al.'s work, Gaussianity is obtained through use of a particular version of CLT, whereas this work exploits Gaussianity of the prior over weights and biases. Going forward, an extension to more general priors/initialisations (like uniform or any sub-Gaussian) is likely to be easier using the CLT approach.
	- Matthews et al.'s assumption on the activation functions is independent of the input set (p.7, Definition 1), whereas this work uses an assumption that is explicitly dependent on input (Eq. (37)) which might be potentially difficult to check.

- (p.15, A.2 end) Should also mention Titsias (2009), "Variational Learning of Inducing Variables in Sparse Gaussian Processes", as a classical reference for approximate GP inference.


****Questions****

- (Section 4) Can you please provide more details on the MC approximation? Specifically, is only the last kernel approximated, or rather all of them, sequentially resampling from the Gaussian with empirical covariance in each layer? In case you tried, is there any qualitative or quantitative difference between the two approaches?

- (Section 4 and Appendix A) Daniely et al. (2016) assume that the inputs to the neural network are l^2 normalised. You mention that the inputs have been normalised in the experiments (A.6). Is this assumption used in any of your proofs? Have you observed that l^2 normalisation improves empirical performance?

- (p.8, Figure 6) How was "the best CNN with the same parameters" selected? If training error is zero for all, was it selected by validation accuracy? I was assuming that what is plotted is an estimate of the **expected** generalisation error, whereas the above selection procedure would be estimating supremum of the support of the generalisation error estimator which does not seem like a fair comparison. Can you please clarify?

- (p.8 and A.6) Why only neural networks with zero training loss were allowed as benchmarks? How did the ones with non-zero training error fared in comparison? Can you please expand on footnote 3?

- (p.8, last sentence) "an observation specific to CNNs and FCNs or LCNs": Matthews et al. (2018, extended version) observed in Section 5.2 that BNNs and their corresponding GP limits do not always perform the same even in the FCN case (cf. their Figure 8). Their paper unfortunately does not compare to equivalent FCNs trained by SGD. Have you experimented with or have an intuition for whether the cases where SGD trained models prevail coincide with the cases where BNNs+MCMC posterior inference outperform their GP limit?

- (p.15, Table 3) The description says you were using erf activation (instead of the more standard ReLU): why? Have you observed any significant differences? Further, how big a proportion of the values in the image is black due to the numerical issues mentioned in A.6.4?

- (p.18, just after Eq. 39) Use of PSD_{|X|d} in (A.4.3) suggests this proof assumes "same" padding is used?! Does the proof generalise to any padding/changing dimensions of filters inside the network?

- (A.6) Can you comment on the pros & cons of "label regression" for classification and how does it compare with approximate inference when softmax is put on top of a GP (perhaps illustrating by a simple experiment on a toy dataset)?


[end of standard review]

















[detailed comments]

****Technical concerns****

Notation-wise, I would strongly encourage incorporating the dependence on network width into your notation, at the very least throughout the appendix. It would greatly reduce the amount of mental book-keeping the reader currently has to do, and significantly increase clarity at several places.

One of my main concerns is that the random variables and their underlying probability space are never formally set-up. This is problematic because convergence in probability is only defined for random variables sharing the same underlying space. At the moment, networks with different widths are not set-up to share a probability space. The practical implication for the approaches relying on convergence in probability of the empirical covariance matrices K is that the convergence in probability is not well-defined exactly because the empirical covariance matrices are not set-up on the same underlying probability space. A possible way to address this issue is to use an approach akin to what Matthews et al. (2018, extended version) call "infinite width, finite fan-out, networks" on page 20. This puts the networks on the same underlying space and because the empirical covariance matrices are measurable functions of thus defined random variables, they will also share the same underlying probability space.

Also regarding convergence in probability, please state explicitly with respect to which metric is the convergence considered when first mentioned (A.4.3 is explicitly using l^\infty; A.4.2 perhaps l^2 or l^\infty?), and make any necessary changes (e.g. show continuity of the mapping C in A.4.2).

At several places within the paper, you state that the law of large numbers (LLN) or the central limit theorem (CLT) can be applied. Apart from other concerns detailed later, these come with conditions on finiteness of certain expectations (usually the first one or two moments of the relevant random variables). Please provide proofs that these expectations are indeed finite and make any assumptions that you need explicit in the main text.

Another major concern is that none of (A.4.1), (A.4.2) and (A.4.3) successfully proves joint convergence of the filters at the top layer as claimed in the main text (e.g. Eq. (10)), and instead only focuses on marginal convergence of each filter which is not sufficient (cf. the comment on joint vs. pairwise Gaussianity below). This is perhaps sufficient if a single filter is the output of the network, but insufficient otherwise, especially when proving convergence with additional layers added on top of the last convolutional layer (as in Section 3) whenever the number filters is taken to infinity.

It would be nice, but not necessary for acceptance of the paper, to extend the proofs to uncountable index sets. I think you could use the same argument as described towards the end of Section 2.2  in (Matthews et al., 2018, extended version) and references therein.


**Other comments**

- I would strongly encourage distinguishing more clearly between probability distributions and density functions. For example, I would infer that lower case p refers to the probability distribution from Eq. (6); however, in Eqs. (8) and (9) the same notation is used for density functions (whilst integrating against the Lebesgue measure). This is quite confusing in this context as the two objects are not the same (see next two comments). I would suggest using capital P when referring to distribution, and lower case p when referring to its density.

- (p.4, Eq. 6) If p is a density, it cannot be equal to a delta distribution. If it is a probability distribution then I am similarly confused - convergence in probability is a statement about behaviour of random variables, not probability distributions; in that case possibly Eq. (6) is trying to say that the empirical distribution of K^l (which is a random variable) conditioned on K^{l-1} converges weakly to the delta distribution on the RHS in probability? Please clarify.

- (p.5, Eq. 10) I would recommend stating explicitly the mode of convergence. If p is the density then even assuming A.4.3 can be fixed to prove weak convergence of the **joint** distribution of filters is not enough not justify Eq. (10) - convergence in distribution does not imply pointwise convergence of the density function. If p is the distribution, then I would possibly use the more standard notation '\otimes' instead of '\prod'.

- (p.17, end of A.4.2) You say "Note that addition of various layers on top (as discussed in Section 3) does not change the proof in a qualitative way". Can you please provide the formal details? At the very least, joint convergence of filters will have to be established if fully connected layers are added on top. This is the main reason why joint convergence of filters in the top layer is important.


****Specific comments & issues for individual proofs****

**Approaches suited infinite networks ("sequential limit")**

As mentioned in the beginning, it is not entirely clear how to formalise infinite networks in a way analogous to Eqs. (1) and (2) in your paper. This is important because you are ultimately proving statements about random variables, like convergence in probability, and this is not possible if those random variables are not formally defined. This section only comments on technical issues with the approaches described in (A.4.1) and (A.4.2). From now on, I assume that the authors' were able to formally define all the mentioned random variables in a way that fits with (A.4.1) and (A.4.2).


(i) Hazan and Jaakola type approach (A.4.1)

This approach essentially iteratively applies a version of the recursion first studied by Hazan and Jaakola (2015), "Steps Toward Deep Kernel Methods from Infinite Neural Networks".

- (p.16, A.4.1) Please provide reference for the claim that "pairwise independent Gaussian implies joint independent Gaussian". This seems to assume that the variables are jointly Gaussian which is, as far as I can see, not established here.
	- see second part of the linked answer for a nice example of three random variables with pairwise standard normal marginals, but joint not the multivariate standard normal:

	https://stats.stackexchange.com/questions/180708/x-i-x-j-independent-when-i%E2%89%A0j-but-x-1-x-2-x-3-dependent/180727#180727

- (p.16, A.4.1) The application of the multivariate CLT is slightly more complicated than the text suggests. Except for the necessity of proving finiteness of the relevant moments, multivariate CLT does not out-of-the-box apply to infinite dimensional random variables like {z_j^{l+1}}_{1 \leq j \leq \infty} as claimed. Hence joint convergence is not proved which will be problematic for the reasons explained earlier.


(ii) Lee et al. type approach (A.4.2)

This type of approach follows the technique used by Lee et al. (2018), "Deep Neural Networks as Gaussian Processes".

Application of the weak law of large numbers (wLLN): As mentioned before, convergence in probability is only possible between random variables on the same underlying space. This is usually not a problem when wLLN is applied as the random variables converge to a constant random variable. Because every constant random variable generates the trivial sigma-algebra, it is measurable for any underlying probability space and thus convergence in probability is well-defined. The situation here is more complicated because the target is constant only conditionally on the previous layer, i.e. is not constant. As a side note, even the conditioning is only well-defined if all random variables live on the same space (conditioning on a random variable is technically conditioning on the sub-sigma-algebra it generates on the shared space).

Assuming the problem with all K^{l, t} (t denotes the dependence on network width), for all l \in {1, ... L} and t \in {1, 2, 3, ...}, being on the same underlying probability space is solved, the next point is application of the wLLN itself. You claim "we can apply the law of large numbers and conclude that [Eq. (6)]" (p.4) which is not entirely correct here. Focusing on the application when the sizes of all the previous layers are held fixed, the two conditions that have to be checked here are: (i) the conditional expectation of the iid summands in Eq. (3) is finite; (ii) the sequence of iid variables is fixed. Please provide an explicit proof of (i). Regarding (ii), I am specifically concerned with the fact that with changing t (and thus network widths), the sequence of random variables changes (because the previous K^{l-1,t} matrix changes) which means that completely different size of the current layer may be necessary to get sufficiently close to the target (which has itself changed with t). In other words, instead of having a fixed infinite sequence of iid random variables, you currently have a sequence of growing finite sets of random variables which are iid only within the finite sets, but not between members of the sequence (different t). The direct implication is that this type of proof is not applicable to the "simultaneous limit" case as claimed in the main text (Section 2.2 says all proofs are equivalent and lead to Eq. (10) which explicitly takes the simultaneous limit), since the application would require some form of uniform convergence in probability akin to (A.4.3). I think that the approach taken in (A.4.3) is a correct way to address this issue and would thus recommend focusing on (A.4.3) and leaving (A.4.2) out. The appendix seems to acknowledge that (A.4.2) does not work for the "simultaneous limit" - please adapt the main text accordingly.

A note on convergence in probability: In Eq. (3), the focus is on convergence in probability of individual entries of the K matrices. This in general does not imply convergence of all entries jointly. However, the type of convergence studied here is convergence to a constant random variable which is fortunate because simultaneous convergence of all entries in probability can be obtained for free in this case (thanks to having a **finite** number of entries of K). I think it might be potentially beneficial for the reader if this was explicitly stated as a footnote with an appropriate reference included.

A note on marginal vs joint probability: As you say above Eq. (23), you are only proving convergence of a single filter marginally, instead of the full sequence {z_j^L}_{1 \leq j \leq \infty} jointly. Convergence of the marginals does not imply convergence of the joint, which will be problematic for the reasons explained earlier.


**Approaches for BNNs ("simultaneous limit")**

(iii) The proof in (A.4.3)

My biggest concern about this approach is that it only establishes convergence of a single filter marginally, instead of the full sequence {z_j^L}_{1 \leq j \leq \infty} jointly. Convergence of the marginals does not imply convergence of the joint, which will be problematic for the reasons explained earlier.

Other comments:

- (p.17) You say "Using Theorem A.1 and the arguments in the above section, it is not difficult to see that a sufficient condition is that the empirical covariance converges in probability to the analytic covariance".
	- Can you please provide more detail as it is unclear what exactly do you have in mind?
	- I will be assuming from now on that you show that a particular combination of the Portmanteau theorem and convergence of K^L in probability to get pointwise convergence of the characteristic function is sufficient.

- (p.18) Condition on activation function: The class \Omega(R) is dependent on the considered input set X through the constant R. This seems slightly cumbersome as it would be desirable to know whether a particular activation function can be used without any reference to the data. It would be nice (but not necessary) if you can derive a condition on \phi which would not rely on the constant R but allows ReLU.

- (p.19, Eq. 48) I see where Eq. (48) is coming from, i.e. from Eq. (44) and the assumption of \bar{\varepsilon} ball around A(K_\infty^l) being in PSD(R), but it would be nicer if you could be a bit more verbose here and also write out the bound explicitly (caveat: I did not check if the definition of \bar{\varepsilon} matches up but assume a potential modification would not affect the proof in a significant way).

- (p.19) The second part of the proof is a little confusing, especially after Eq. (49) - please be more verbose here. For example, just after Eq. (49), it is said that because the two random variables have the same distribution, property (3) of \Omega(R)'s definition can be applied. However the two random variables are not identical and importantly are not constructed on the same underlying probability space. Property (3) is a statement about the the set of random variables {T_n (Sigma)}_{Sigma \in PSD_2(R)} and not about the different 2x2 submatrices of K^{l+1}, but it needs to be applied to the latter. When this is clarified, the next point that could be made clearer is in the following sentence where changing t will affect the 2x2 submatrices of K^{l+1,t} as well as the bound through U(t) and V(t); it is not immediately obvious that the proof goes through as claimed so please be a bit more verbose.


****Typos and other minor remarks****

- (p.2, top) "hidden layers go to infinity uniformly": The use of word uniformly is non-standard in this context. Please clarify.

- (p.3, Eq. 2) Using x for both inputs and post-activations is slightly confusing.

- (p.4, Eq. 5) Should v_\beta multiply \sigma_\beta^2 ?

- (p. 4) The summands in Equation (3) are iid -> "conditionally iid" (please also specify the conditioning variables/sigma-algebra).

- (p.4, Eq. 4) Eq. (4) is slightly confusing given you mention that K is a 4D object on the previous page.
	- I only understood K is "flattened" into |X|d x |X|d matrix when I reached (A.4.3) - this should be stated in main text as otherwise the above confusion arises.

- (p.5, 3 and 3.1) The introduction of "curly" K is slighlty confusing. Please provide more detail when introducing the notation, e.g. state in what space the object lives.

- (p.5, before Eq. (11)) Is R^{n^(l+1)} the right space for vec(z^L) ? It seems that the meaning of z changes here as compared to the definition in Eq. (2). If z is still defined as in Eq. (2), how exactly is the vec operator defined here? Please clarify.

- (p.16, A.4.2) "law of large number" -> "weak law of large numbers"

- (p.17) T_n is technically not a function from PSD_2 only but also from some underlying probability space into a measurable space (i.e. can be viewed as a random variable from the product space of PSD_2 and some other measurable space).

- (p.18, Eq. 38) Missing dot at the end. Also the K matrix either should or shouldn't have the superscript "l" (now mixed); it does have the superscript in Eq. (39) so probably "should".

- (p.18, Eq. 39) Slightly confusing notation. Please clarify that both K and A(K) should have diagonal within the given range.

- (p.18) "squared integrable" -> "square integrable" or "square-integrable"

- (p.18) Last display before Eq. (43): second inequality can be replaced by equality?!

- (p.19, Eq. 47) The absolute value should be sup norm.

- (p.19, Eq. 49) LHS is a scalar, RHS a 2x2 matrix (typo).

- (p.19, last sentence of the proof) It does not seem the inequalities need to be strict.

---

> ### Author Response · Authors · 2018-11-27
> **8/8 AnonReviewer3 Reply**
>
> ------------------------------------------------------------------------------------
> >>> ****Typos and other minor remarks****
> - (p.2, top) "hidden layers go to infinity uniformly": The use of word uniformly is non-standard in this context. Please clarify.
>
> Done. The “uniform” qualifier was used by analogy of uniform function convergence.
>
> ------------------------------------------------------------------------------------
> >>>- (p.3, Eq. 2) Using x for both inputs and post-activations is slightly confusing.
>
> Changed post-activations (called activations in the text) to “y”.
>
> ------------------------------------------------------------------------------------
> >>>- (p.4, Eq. 5) Should v_\beta multiply \sigma_\beta^2 ?
>
> It should not, thank you, fixed.
>
> ------------------------------------------------------------------------------------
> >>> - (p. 4) The summands in Equation (3) are iid -> "conditionally iid" (please also specify the conditioning variables/sigma-algebra).
>
> Done, thank you.
>
> ------------------------------------------------------------------------------------
> >>> - (p.4, Eq. 4) Eq. (4) is slightly confusing given you mention that K is a 4D object on the previous page.
> 	- I only understood K is "flattened" into |X|d x |X|d matrix when I reached (A.4.3) - this should be stated in main text as otherwise the above confusion arises.
>
> Thank you, fixed and clarified (section 2.1.Shapes and indexing).
>
> ------------------------------------------------------------------------------------
> >>>- (p.5, 3 and 3.1) The introduction of "curly" K is slighlty confusing. Please provide more detail when introducing the notation, e.g. state in what space the object lives.
>
> Done (see also new section 2.1.Shapes and indexing).
>
> ------------------------------------------------------------------------------------
> >>> - (p.5, before Eq. (11)) Is R^{n^(l+1)} the right space for vec(z^L) ? It seems that the meaning of z changes here as compared to the definition in Eq. (2). If z is still defined as in Eq. (2), how exactly is the vec operator defined here? Please clarify.
>
> Note that it’s n^{L+1} times d, yet you are correct that it should’ve been the dimension of z^L(x), not z^L. We have fixed the error and substantially improved the clarity of this section and clarified the notation in section 2.1.Shapes and indexing.
>
> ------------------------------------------------------------------------------------
> >>> - (p.16, A.4.2) "law of large number" -> "weak law of large numbers"
>
> Done.
>
> ------------------------------------------------------------------------------------
> >>> - (p.17) T_n is technically not a function from PSD_2 only but also from some underlying probability space into a measurable space (i.e. can be viewed as a random variable from the product space of PSD_2 and some other measurable space).
>
> We no longer use T_n notation in the new revision.
>
> ------------------------------------------------------------------------------------
> >>> - (p.18, Eq. 38) Missing dot at the end. Also the K matrix either should or shouldn't have the superscript "l" (now mixed); it does have the superscript in Eq. (39) so probably "should".
>
> Done.
>
> ------------------------------------------------------------------------------------
> >>> - (p.18, Eq. 39) Slightly confusing notation. Please clarify that both K and A(K) should have diagonal within the given range.
>
> Done (no such confusing notation in the new revision).
>
> ------------------------------------------------------------------------------------
> >>>- (p.18) "squared integrable" -> "square integrable" or "square-integrable"
>
> Done.
>
> ------------------------------------------------------------------------------------
> >>> - (p.18) Last display before Eq. (43): second inequality can be replaced by equality?!
>
> Thank you, done.
>
> ------------------------------------------------------------------------------------
> >>> - (p.19, Eq. 47) The absolute value should be sup norm.
>
> We believe the expression is correct.
>
> ------------------------------------------------------------------------------------
> >>> - (p.19, Eq. 49) LHS is a scalar, RHS a 2x2 matrix (typo).
>
> Both are scalars (\Tau_\infty is defined as a scalar).
>
> ------------------------------------------------------------------------------------
> >>> - (p.19, last sentence of the proof) It does not seem the inequalities need to be strict.
>
> Thank you, fixed for n^{l+1}.

---

> ### Author Response · Authors · 2018-11-27
> **7/8 AnonReviewer3 Reply**
>
> ------------------------------------------------------------------------------------
> >>> (ii) Lee et al. type approach (A.4.2)
> [...]
>
> Per your suggestion we have removed the section A.4.2. in this revision.
>
> ------------------------------------------------------------------------------------
> >>> A note on convergence in probability: In Eq. (3), the focus is on convergence in probability of individual entries of the K matrices. This in general does not imply convergence of all entries jointly. However, the type of convergence studied here is convergence to a constant random variable which is fortunate because simultaneous convergence of all entries in probability can be obtained for free in this case (thanks to having a **finite** number of entries of K). I think it might be potentially beneficial for the reader if this was explicitly stated as a footnote with an appropriate reference included.
>
> We have added a footnote 4 clarifying this step (we believe the limit being a constant not necessary though as long as their number is finite).
>
> ------------------------------------------------------------------------------------
> >> A note on marginal vs joint probability: As you say above Eq. (23), you are only proving convergence of a single filter marginally, instead of the full sequence {z_j^L}_{1 \leq j \leq \infty} jointly. Convergence of the marginals does not imply convergence of the joint, which will be problematic for the reasons explained earlier.
>
> We now prove joint convergence in the new revision.
>
> ------------------------------------------------------------------------------------
> >>> **Approaches for BNNs ("simultaneous limit")**
> (iii) The proof in (A.4.3)
> My biggest concern about this approach is that it only establishes convergence of a single filter marginally, instead of the full sequence {z_j^L}_{1 \leq j \leq \infty} jointly. Convergence of the marginals does not imply convergence of the joint, which will be problematic for the reasons explained earlier.
>
> Done, see Theorem A.6.
>
> ------------------------------------------------------------------------------------
> >>> Other comments:
> - (p.17) You say "Using Theorem A.1 and the arguments in the above section, it is not difficult to see that a sufficient condition is that the empirical covariance converges in probability to the analytic covariance".
> 	- Can you please provide more detail as it is unclear what exactly do you have in mind?
>
> Done, see Theorem A.6.
>
> ------------------------------------------------------------------------------------
> >>> - (p.18) Condition on activation function: The class \Omega(R) is dependent on the considered input set X through the constant R. This seems slightly cumbersome as it would be desirable to know whether a particular activation function can be used without any reference to the data. It would be nice (but not necessary) if you can derive a condition on \phi which would not rely on the constant R but allows ReLU.
>
> Done, there’s no more dataset dependency.
>
> ------------------------------------------------------------------------------------
> >>>- (p.19, Eq. 48) I see where Eq. (48) is coming from, i.e. from Eq. (44) and the assumption of \bar{\varepsilon} ball around A(K_\infty^l) being in PSD(R), but it would be nicer if you could be a bit more verbose here and also write out the bound explicitly (caveat: I did not check if the definition of \bar{\varepsilon} matches up but assume a potential modification would not affect the proof in a significant way).
>
> Done (now Equations 70-72).
>
> ------------------------------------------------------------------------------------
> >>> - (p.19) The second part of the proof is a little confusing, especially after Eq. (49) - please be more verbose here. For example, just after Eq. (49), it is said that because the two random variables have the same distribution, property (3) of \Omega(R)'s definition can be applied. However the two random variables are not identical and importantly are not constructed on the same underlying probability space. Property (3) is a statement about the the set of random variables {T_n (Sigma)}_{Sigma \in PSD_2(R)} and not about the different 2x2 submatrices of K^{l+1}, but it needs to be applied to the latter.
>
> Done (Equations 76-77 + see new modified revision of property 3 (now Equation 48)).
>
> ------------------------------------------------------------------------------------
> >>> When this is clarified, the next point that could be made clearer is in the following sentence where changing t will affect the 2x2 submatrices of K^{l+1,t} as well as the bound through U(t) and V(t); it is not immediately obvious that the proof goes through as claimed so please be a bit more verbose.
>
> Done, we have substantially expanded that part of the proof (starting from Equation 75).

---

> ### Author Response · Authors · 2018-11-27
> **6/8 AnonReviewer3 Reply**
>
> ------------------------------------------------------------------------------------
> >>> It would be nice, but not necessary for acceptance of the paper, to extend the proofs to uncountable index sets. I think you could use the same argument as described towards the end of Section 2.2  in (Matthews et al., 2018, extended version) and references therein.
>
> Thank you, indeed our proof extends to the case of countably many inputs with the metric referenced in Matthews et al. 2018, and we now mention it in section A.5.1.
>
> ------------------------------------------------------------------------------------
> >>> **Other comments**
> - I would strongly encourage distinguishing more clearly between probability distributions and density functions. For example, I would infer that lower case p refers to the probability distribution from Eq. (6); however, in Eqs. (8) and (9) the same notation is used for density functions (whilst integrating against the Lebesgue measure). This is quite confusing in this context as the two objects are not the same (see next two comments). I would suggest using capital P when referring to distribution, and lower case p when referring to its density.
>
> Done, we believe the new revision should not have any confusing notation.
>
> ------------------------------------------------------------------------------------
> >>> - (p.4, Eq. 6) If p is a density, it cannot be equal to a delta distribution. If it is a probability distribution then I am similarly confused - convergence in probability is a statement about behaviour of random variables, not probability distributions; in that case possibly Eq. (6) is trying to say that the empirical distribution of K^l (which is a random variable) conditioned on K^{l-1} converges weakly to the delta distribution on the RHS in probability? Please clarify.
>
> Thank you, we no longer use delta-function notation in the main text and are clear about modes of convergence.
>
> ------------------------------------------------------------------------------------
> >>> - (p.5, Eq. 10) I would recommend stating explicitly the mode of convergence. If p is the density then even assuming A.4.3 can be fixed to prove weak convergence of the **joint** distribution of filters is not enough not justify Eq. (10) - convergence in distribution does not imply pointwise convergence of the density function. If p is the distribution, then I would possibly use the more standard notation '\otimes' instead of '\prod'.
>
> Thank you for pointing this out, we now always state modes explicitly and do not imply convergence of probability densities.
>
> ------------------------------------------------------------------------------------
> >>> - (p.17, end of A.4.2) You say "Note that addition of various layers on top (as discussed in Section 3) does not change the proof in a qualitative way". Can you please provide the formal details? At the very least, joint convergence of filters will have to be established if fully connected layers are added on top. This is the main reason why joint convergence of filters in the top layer is important.
>
> Done, see Theorem A.6.
>
> ------------------------------------------------------------------------------------
> >>> ****Specific comments & issues for individual proofs****
> **Approaches suited infinite networks ("sequential limit")**
> As mentioned in the beginning, it is not entirely clear how to formalise infinite networks in a way analogous to Eqs. (1) and (2) in your paper. This is important because you are ultimately proving statements about random variables, like convergence in probability, and this is not possible if those random variables are not formally defined. This section only comments on technical issues with the approaches described in (A.4.1) and (A.4.2). From now on, I assume that the authors' were able to formally define all the mentioned random variables in a way that fits with (A.4.1) and (A.4.2).
>
> Done. Specifically, we provide a definition in A.5.3 (former A.4.1, “Sequential limit”; note that, we don’t make any convergence in probability statements here, only in distribution). A.4.2 is left out.
>
> ------------------------------------------------------------------------------------
> >>> (i) Hazan and Jaakola type approach (A.4.1)
> [...]
> - (p.16, A.4.1) The application of the multivariate CLT is slightly more complicated than the text suggests. Except for the necessity of proving finiteness of the relevant moments, multivariate CLT does not out-of-the-box apply to infinite dimensional random variables like {z_j^{l+1}}_{1 \leq j \leq \infty} as claimed. Hence joint convergence is not proved which will be problematic for the reasons explained earlier.
>
> We have significantly revamped this section (now A.5.3, “Sequential limit”), including proving joint convergence and finiteness of the moments.

---

> ### Author Response · Authors · 2018-11-27
> **5/8 AnonReviewer3 Reply**
>
> ------------------------------------------------------------------------------------
> >>>[detailed comments]
> ****Technical concerns****
> Notation-wise, I would strongly encourage incorporating the dependence on network width into your notation, at the very least throughout the appendix. It would greatly reduce the amount of mental book-keeping the reader currently has to do, and significantly increase clarity at several places.
>
> Done, we now use “_t” subscript to show dependence on n^1(t), ..., n^L(t) in the appendix.
>
> ------------------------------------------------------------------------------------
> >>> One of my main concerns is that the random variables and their underlying probability space are never formally set-up. This is problematic because convergence in probability is only defined for random variables sharing the same underlying space. At the moment, networks with different widths are not set-up to share a probability space. The practical implication for the approaches relying on convergence in probability of the empirical covariance matrices K is that the convergence in probability is not well-defined exactly because the empirical covariance matrices are not set-up on the same underlying probability space. A possible way to address this issue is to use an approach akin to what Matthews et al. (2018, extended version) call "infinite width, finite fan-out, networks" on page 20. This puts the networks on the same underlying space and because the empirical covariance matrices are measurable functions of thus defined random variables, they will also share the same underlying probability space.
>
> Done, we now define the probability space in section A.5.1. Networks of different widths now do share the underlying probability space, and hence {K^l} covariances as well.
>
> ------------------------------------------------------------------------------------
> >>> Also regarding convergence in probability, please state explicitly with respect to which metric is the convergence considered when first mentioned (A.4.3 is explicitly using l^\infty; A.4.2 perhaps l^2 or l^\infty?), and make any necessary changes (e.g. show continuity of the mapping C in A.4.2).
>
> Convergence is w.r.t.  l^\infty and we now state it explicitly in section A.5.6. However, note that due to finite dimensionality all norms are equivalent. While we no longer have section A.4.2, continuity of map C follows from Lemma A.6.2 in the new revision.
>
> ------------------------------------------------------------------------------------
> >>> At several places within the paper, you state that the law of large numbers (LLN) or the central limit theorem (CLT) can be applied. Apart from other concerns detailed later, these come with conditions on finiteness of certain expectations (usually the first one or two moments of the relevant random variables). Please provide proofs that these expectations are indeed finite and make any assumptions that you need explicit in the main text.
>
> We now prove finiteness of the necessary moments (see Theorem A.2).
> ------------------------------------------------------------------------------------
> >>> Another major concern is that none of (A.4.1), (A.4.2) and (A.4.3) successfully proves joint convergence of the filters at the top layer as claimed in the main text (e.g. Eq. (10)), and instead only focuses on marginal convergence of each filter which is not sufficient (cf. the comment on joint vs. pairwise Gaussianity below). This is perhaps sufficient if a single filter is the output of the network, but insufficient otherwise, especially when proving convergence with additional layers added on top of the last convolutional layer (as in Section 3) whenever the number filters is taken to infinity.
>
> Done, we now explicitly prove joint convergence wherever applicable.

---

> ### Author Response · Authors · 2018-11-27
> **4/8 AnonReviewer3 Reply**
>
> ------------------------------------------------------------------------------------
> >>> - (p.8, last sentence) "an observation specific to CNNs and FCNs or LCNs": Matthews et al. (2018, extended version) observed in Section 5.2 that BNNs and their corresponding GP limits do not always perform the same even in the FCN case (cf. their Figure 8). Their paper unfortunately does not compare to equivalent FCNs trained by SGD. Have you experimented with or have an intuition for whether the cases where SGD trained models prevail coincide with the cases where BNNs+MCMC posterior inference outperform their GP limit?
>
> We have not explored BNNs+MCMC experiments in this work. As mentioned in the Discussion (section 5.3), we attribute the observation (SGD-trained finite CNNs outperforming their GPs) to the loss of pixel-pixel covariances. This happens in infinite Bayesian (contrary to finite SGD-trained) models, and we do not have strong intuitions at the moment on whether to attribute this to Bayesian treatment or infinite width (or both). However, as we have mentioned in the conclusion, we enthusiastically agree that this is a very interesting question to answer in future work!
>
> ------------------------------------------------------------------------------------
> >>>- (p.15, Table 3) The description says you were using erf activation (instead of the more standard ReLU): why? Have you observed any significant differences?
>
> We did not have a particular reason, and have produced some preliminary results for ReLU below:
> https://www.dropbox.com/s/d3lmb84o9b06syt/infoprop_relu.pdf?dl=0,
> where we see a qualitatively similar trend that is in agreement with Lee et al. 2018 (Figure 4.b, Figure 9, bottom row; rightmost phase diagram is borrowed from their paper as well in our plot).
>
> ------------------------------------------------------------------------------------
> >>>Further, how big a proportion of the values in the image is black due to the numerical issues mentioned in A.6.4?
>
> Total of 13%, 2792 out of total 20000 trials (2500 per plot in the table) failed.
> % of failures per each plot:
>
> ----------------------------------------
> Depth:    | 1 | 10 | 100 | 1000 |
> ----------------------------------------
> CNN-GP | 0 | 0   | 9     | 44     |
> ----------------------------------------
> FCN-GP | 0 | 0   | 13   | 45     |
> ----------------------------------------
>
> Please note that the line between a numerical failure and poor performance is blurry and depends on the specific experimental setup (see A.7.4). Indeed, not all numerical issues result in failures and sometimes will simply produce poor / random results.
>
> ------------------------------------------------------------------------------------
> >>> - (p.18, just after Eq. 39) Use of PSD_{|X|d} in (A.4.3) suggests this proof assumes "same" padding is used?! Does the proof generalise to any padding/changing dimensions of filters inside the network?
>
> We now state that we use circular padding and the spatial shape indeed is considered to remain fixed for simplicity (see section 2.1). While we do not consider changing padding / dimensions inside the network, we believe the proof to generalize to such cases easily (by introducing a different A^l operator for each layer, which will still be affine and Lipschitz-continuous).
>
> ------------------------------------------------------------------------------------
> >>> - (A.6) Can you comment on the pros & cons of "label regression" for classification and how does it compare with approximate inference when softmax is put on top of a GP (perhaps illustrating by a simple experiment on a toy dataset)?
>
> In order to establish and understand correspondence to the GPs we focused on cases where exact inference on GP side was possible (a benefit of label regression) while working on realistic well known dataset for CNNs.
>
> Apparent downsides of label regression are: (a) the independent prior on different output classes, which discards our prior knowledge about them being mutually-exclusive and (b) complications in interpreting GP predictions and their uncertainty on categorical outputs. However, the practical impact of softmax on best achieved accuracy in classification tasks is, to the best of our knowledge, not clear due to how well our MSE-trained NNs perform in this work (Table 2 (former 1); we believe FCN results to be close to SOTA using cross-entropy loss, and CNN results to be decent yet unfortunately hard to compare to SOTA due to architecture limitations), and due to FCN- and CNN-GPs performing similarly to the best considered FCNs and LCN. Therefore, while we certainly believe there to be a difference between label regression and proper classification, we do not think a simple toy task can fully illustrate it.
>
> We still think it is interesting future work to implement and investigate the effects of softmax output using cross entropy loss.

---

> ### Author Response · Authors · 2018-11-27
> **3/8 AnonReviewer3 Reply**
>
> ------------------------------------------------------------------------------------
> >>> - (Section 4 and Appendix A) Daniely et al. (2016) assume that the inputs to the neural network are l^2 normalised. You mention that the inputs have been normalised in the experiments (A.6). Is this assumption used in any of your proofs? Have you observed that l^2 normalisation improves empirical performance?
>
> The assumption is not used in the new revision. We did not try other (or no) normalization approaches, and normalized inputs mainly as a common preprocessing practice in machine learning.
>
> ------------------------------------------------------------------------------------
> >>> - (p.8, Figure 6) How was "the best CNN with the same parameters" selected? If training error is zero for all, was it selected by validation accuracy?
>
> Yes, we state this in experimental details (A.7.5), and now also in that caption (now Figure 3, c).
>
> ------------------------------------------------------------------------------------
> >>> I was assuming that what is plotted is an estimate of the **expected** generalisation error, whereas the above selection procedure would be estimating supremum of the support of the generalisation error estimator which does not seem like a fair comparison. Can you please clarify?
>
> If we understand you correctly (please let us know if not), your concern is with us reporting validation and not test accuracy. This is indeed not a fair comparison, and is slightly biased in favor of NNs over GPs. We have replace it with test accuracy (now Figure 3, c), which is extremely similar.
>
> ------------------------------------------------------------------------------------
> >>> - (p.8 and A.6) Why only neural networks with zero training loss were allowed as benchmarks?
>
> Please note that for practical benchmarking purposes we have presented Table 2 (former 1), where non-zero accuracy (not loss - exactly zero loss was not achieved by our trained NNs) results are presented in parentheses and were emphasized in the caption. Otherwise, we wanted to put the two classes of models in as similar conditions as was practically possible; since the GP without regularization perfectly fits the training set, we filtered for this condition in the networks with SGD training.
>
> Relatedly, note that NN-GP correspondence could be obtained by Sample-then-optimize procedure of [1], where one train only the read-out weights to convergence (infinite steps) using gradient descent training. For realizable problems (over-parameterized) the trained networks will obtain zero loss. Therefore, trained networks that would correspond to NN-GP necessarily should have zero loss (or close to zero loss if only finite training steps were taken).
>
>  In our NN experiments with SGD, we relaxed this requirement but still required models to produce 100% accurate train set predictions, and believe that controlling for perfect accuracy allowed us to make arguably more interesting conclusions. E.g. one of the results of this paper is an observation that SGD-trained CNNs can significantly outperform equivalent CNN-GPs. Without controlling for train accuracy the difference may come from CNNs benefitting from underfitting. However the fact that SGD-trained CNNs significantly outperform CNN-GPs even with conditioning for zero error indicates an interesting and more specific mechanism of breakdown of NN-GP correspondence in SGD training.
>
> [1] Alexander G. de. G Matthews,  Jiri Hron,  Richard E. Turner,  and Zoubin Ghahramani.   Sample-then-optimize posterior sampling for bayesian linear models.   In NIPS Workshop on Advances in Approximate Bayesian Inference, 2017
>
> ------------------------------------------------------------------------------------
> >>> How did the ones with non-zero training error fared in comparison?
>
> As can be seen in Table 2 (former 1) and noted in the caption, underfitting tends to improve generalization for CNNs. Further, we have produced the analogous plots without the 100% accuracy requirement (NNs can underfit):
> https://www.dropbox.com/s/vxuhzyfj9we9pj2/underfit.pdf?dl=0
> As we can see, on full CIFAR10 (top) now the majority of models perform better in the NN case, suggesting that properly tuned underfitting can be a contributing factor of good generalization. However, on the smaller task (bottom), while the trend is altered, the plots are qualitatively similar, potentially due to underfitting on a small dataset being unlikely and hence not playing a significant role.
>
> ------------------------------------------------------------------------------------
> >>> Can you please expand on footnote 3?
> Please see our comments above + we have added a sentence emphasizing that underfitting can lead to better generalization in the footnote.

---

> ### Author Response · Authors · 2018-11-27
> **2/8 AnonReviewer3 Reply**
>
> ------------------------------------------------------------------------------------
> >>>However, I believe that the argument in (A.4.3) can potentially be rectified, and, as I detail below, is of greater interest to the community relative to the ones in (A.4.1) and (A.4.2). If this is accomplished and the proofs in (A.4.1) and (A.4.2) are either also fixed or left out (A.4.3 is sufficient to justify the claims in the main body), I am willing to significantly improve my rating of this paper and potentially recommend acceptance. For this reason, a "detailed comments" section is appended at the end of the standard review where the technical issues are described in much greater detail.
>
> Thank you, we believe to have addressed all of your concerns in the new revision (section A.5). We have left the section A.4.2 out as you advised.
>
> ------------------------------------------------------------------------------------
> >>> ****General comments****
> **Bayesian vs. infinite neural networks**
> [...] Others may of course disagree and find "sequential" limits more interesting, but if the authors wish to keep the description of (A.4.2) in the main paper (Sections 2.2.1-2.2.3), it would be highly beneficial if readers were given the opportunity to understand the differences between the two types of limits so that they can form their own judgement. The authors should then also make clearer that the approach described in Sections 2.2.1-2.2.3 cannot be used to obtain the final result, Eq. (10). I would rather recommend reworking Sections 2.2.1-2.2.3 based on the "simultaneous" limit argument in (A.4.3) which unlike the current one can justify the result in Eq. (10) stated at the end.
>
> Thank you, we have both revamped the presentation in section 2.2, and added a discussion about different limit approaches in section A.5.
>
> ------------------------------------------------------------------------------------
> >>> **Other comments**
> - (p.2, top) You say your results are "strengthening and extending the result of Matthews et al. (2018)" which is somewhat confusing. Matthews et al. prove a result for FCNs whereas this paper focuses on CNNs. Extension of (A.4.3) to FCNs may well be possible but is not included in this paper.
>
> We added a clarification on how to apply our results to LCNs and FCNs, see section A.5.
>
> ------------------------------------------------------------------------------------
> >>> Results in (A.4.1) and (A.4.2) are for the "sequential" whereas Matthews et al. study the "simultaneous" limit.
>
> The emphasis in our work (and in section 2.2 in particular) is now on the simultaneous limit as well.
>
> ------------------------------------------------------------------------------------
> >>> Further differences:
> 	- Matthews et al. prove convergence for any countable rather than only finite input sets.
>
> Thank you for pointing this out, our proof indeed generalizes to the same setting as we now mention in section A.5.1.
>
> ------------------------------------------------------------------------------------
> >>>	- In Matthews et al.'s work, Gaussianity is obtained through use of a particular version of CLT, whereas this work exploits Gaussianity of the prior over weights and biases. Going forward, an extension to more general priors/initialisations (like uniform or any sub-Gaussian) is likely to be easier using the CLT approach.
>
> We have partially relaxed our assumptions on the priors, see section A.5.1 in the new revision. However, please also note that Matthews et al explicitly assume Gaussian priors in their work to the best of our knowledge.
>
> ------------------------------------------------------------------------------------
> >>> - Matthews et al.'s assumption on the activation functions is independent of the input set (p.7, Definition 1), whereas this work uses an assumption that is explicitly dependent on input (Eq. (37)) which might be potentially difficult to check.
>
> We no longer have this dependency in the new revision.
>
> ------------------------------------------------------------------------------------
> >>> - (p.15, A.2 end) Should also mention Titsias (2009), "Variational Learning of Inducing Variables in Sparse Gaussian Processes", as a classical reference for approximate GP inference.
>
> Thank you, done.
>
> ------------------------------------------------------------------------------------
> >>> ****Questions****
> - (Section 4) Can you please provide more details on the MC approximation? Specifically, is only the last kernel approximated, or rather all of them, sequentially resampling from the Gaussian with empirical covariance in each layer? In case you tried, is there any qualitative or quantitative difference between the two approaches?
>
> We have only tried to approximate the last kernel, i.e. sampling random networks and averaging their top-level activations.

---

> ### Author Response · Authors · 2018-11-27
> **1/8 AnonReviewer3 Reply and Summary**
>
> Thank you for your _extremely_ detailed and insightful review. Your suggestions have allowed us to significantly improve on the quality of our submission and we are very grateful for your hard work. Please find below a summary of our changes, as well as responses to your specific comments.
>
> ------------------------------------------------------------------------------------
> ****Summary****
> We believe the simultaneous limit proof in section A.4.3 (now A.5.4) to be largely correct, however, as you rightly pointed out, it was lacking in terms of explicit treatment of various aspects and suffered from typos / notational inconsistencies. We believe the current revision addresses all the relevant issues.
> We have made sure to have a more explicit, consistent, and rigorous notation throughout the paper. We especially encourage you to review the new section 2.1. “Shapes and indexing” where we describe our notation in detail.
> In response to your valid concerns we have omitted section A.4.2 and have rewritten section 2.2 in the main text to reference results from A.4.3 (now A.5.4). Section A.4.1 (now A.5.3) was revamped to rigorously define a sequential limit NN-GP and show that it results in the same covariance as the simultaneous limit (A.4.3, now A.5.4).
>
> ------------------------------------------------------------------------------------
> >>> These experiments and investigations are however based on a theoretical foundation which suffers from several issues. The main problems are an incorrect proof of convergence of the joint distribution of filters, and an improper use of convergence in probability in cases where random variables do not share a common underlying probability space. Unfortunately, either of these by itself invalidates the main theoretical claims which is why I am recommending rejection of the paper.
>
> We now formally define an underlying probability space (see A.5.1). Note that random variables {K^l} have constant dimensionality (|X|d x |X|d, see Equation 4) that does not change with widths. Same convention was implied in the previous revision, however we acknowledge that the notation was not explicit enough and may have been a source of confusion, especially in conjunction with the derivations in A.4.2. Further, we derive the joint convergence (wherever applicable) which can be obtained by coupling the convergence of the covariance in probability to deterministic quantities and an argument using characteristic function. Please see Theorems A.2 and A.5.

---

> ### Author Response · Authors · 2018-12-05
> **AnonReviewer3 update reply**
>
> Thank you for promptly reviewing our revision!
>
> ------------------------------------------------------------------------------------
> >>> - (p.20, A.5.1) To ensure the random variables are well-defined, please state explicitly which sigma algebra is F (I am assuming the product Borel sigma-algebra + the relevant definitions of the random variables). This is important for the reader to understand what convergence in distribution on this particular space does and does not imply.
>
> Will do.
>
> ------------------------------------------------------------------------------------
> >>> Some readers might also appreciate if you used the mentioned "infinite width, finite fan-out, networks" (Matthews et al.) construction (or similar) which would ensure that the collection of random variables {z_i^l}_{i \in N*} is well-defined for any network width and l, which currently does not seem to be the case according to Eqs. (28-29). If the full countably infinite vectors of random variables are not defined for all networks in the sequence, it is not possible to prove their convergence in distribution to the relevant GPs.
>
> Thank you, we agree that currently the construction process in A.5.3 is not explicit enough to define the countably-infinite collection {z_i^{l, \infty}}_{i \in N*} (as you point out below in more detail), and we will make it so in the next revision.
>
> ------------------------------------------------------------------------------------
> >>> - (p.21, A.5.3) Thank you for clarifying the definition of elements of the sequential limit. If possible, I would further recommend first fixing the probability space and then defining the random variables (the argument just before Theorem A.2 seems somewhat circular as R.V.s should first be defined on some space, and not put on a probability space post-hoc; perhaps some product space with the product sigma-algebra would work here?!).
>
> Thank you for the suggestion. One can define {z_i^{l, \infty}}_{i \in N*} in the place-holder (A.5.1.iii) before defining the neural networks. This avoids reconstructing the probability space / apparent circularity. We will make sure to be more explicit about it in the next revision.
>
> ------------------------------------------------------------------------------------
> >>> Furthermore, if I understand correctly, there are now L sequences of neural networks (one sequence for networks with 0, ..., L-1 "infinite layers"), rather than a single sequence, and the "infinite layers" are squashed into a single "infinite layer" which is represented by z_i^\infty? In other words, all the infinite layers are replaced by iid samples from a particular GP and only the finite layers have the standard neural network structure? If I am mistaken (or not), perhaps a further explanatory footnote would help the reader.
>
> You are correct, and we will elaborate on this more in the next revision. This is an inconvenience of the sequential limit approach, since the outputs of any hidden layers only converge in distribution and not necessarily almost surely (point-wisely). Thus we have to re-define/construct them. We believe this inconvenience to be present in all prior / concurrent work using the sequential limit. It might be possible to circumvent this issue with the help of Skorokhod’s Representation Theorem.
>
> ------------------------------------------------------------------------------------
> >>> - (p.21, A.5.3 & p.23, A.5.4) Thank you for improving the discussion of joint convergence. Please clarify that proving convergence for any finite m is sufficient for proving convergence in distribution of the countably infinite vector {z_i}_{i \in N*} for the **product Borel sigma-algebra** (e.g. using an argument like the one on p.19 of Billingsley (1999)).
>
> Will do, thank you for pointing this out.
>
> ------------------------------------------------------------------------------------
> >>> - (p.21) "Uniformly square-integrable": to me, this phrase suggests that the collection of squares of the functions has to be uniformly integrable but the definition in Eq. (27) only states one of the conditions in definition of uniform integrability. Please clarify that "uniform square-integrability" here is not related to the standard notion of "uniform integrability" in the literature.
>
> Thanks. Will do.

---

> ### Author Response · Authors · 2019-01-03
> **Suggesting authorship**
>
> Dear AnonReviewer3,
>
> We would like to recognize your very detailed and useful review by including you as a co-author in the next revision (among other pending changes addressing your and other reviewers' final remarks). If you are interested, please let us know your name, email and affiliation.
>
> Thank you!

---

### Official Review · AnonReviewer1 · 2018-11-02
**BAYESIAN CONVOLUTIONAL NEURAL NETWORKS WITH MANY CHANNELS ARE GAUSSIAN PROCESSES**

**Rating:** 7
**Confidence:** 2

**Review:**

Overall Score: 7/10.
Confidence Score: 3/10. (This paper includes so many ideas that I have not been able to prove that are right due to
my limited knowledge, but I think that there are correct).

Summary of the main ideas: This paper establishes a theoretical correspondence between BCNN with many channels and GP and
propsoes a Monte Carlo method to estimate the GP corresponding to a NN architecture. It is a very strong and complete
paper since its gives theoretical contents and experiments content. I think that it is a really good result that should
be read by anyone interested in Neural Network and GP equivalences, and that Machine Learning in general needs these kind
of papers that establish this complicated equivalences.

Related to: The work by Lee and G. Matthews (2018) regarding equivalence between Deep Neural Networks and GPs and the
Convolutional Neural Network framework.

Strengths:
Theoretical content, Experiments and methodology content (even a Monte Carlo approach) makes it a very complete paper.
Having been able to establish complicated and necessary equivalences.

Weaknesses:
Very difficult for newcomers or non expert technical readers.

Does this submission add value to the ICLR community? : Yes, it adds, and a lot.

Quality:
Is this submission technically sound?: Yes it is, it is a necessary step in GP-NN equivalence research.
Are claims well supported by theoretical analysis or experimental results?: Yes, quite sure.
Is this a complete piece of work or work in progress?: Complete piece of work.
Are the authors careful and honest about evaluating both the strengths and weaknesses of their work?: Yes, they are.

Clarity:
Is the submission clearly written?: Yes, but I suggest giving formal introductions to some concepts in the introduction
and include a figure with the ideas given or the equivalences.
Is it well organized?: Yes, although sometimes section feel a little but put one after the another. More cohesion would be
added if they are introduce before.
Does it adequately inform the reader?: Yes.

Originality:
Are the tasks or methods new?: The monte carlo is new, the other methods not but the task of the equivalence is new.
Is the work a novel combination of well-known techniques?: It is kind of a combination, but the proposed ideas are new, it is very theoretical.
Is it clear how this work differs from previous contributions?: Yes, authors bother in explaining it clearly.
Is related work adequately cited?: Yes, this is a huge positive point of the paper.

Significance:
Are the results important?: From my point of view, yes they are.
Are others likely to use the ideas or build on them?: I think so, because the topic is hot right now.
Does the submission address a difficult task in a better way than previous work?: It is a new task.
Does it advance the state of the art in a demonstrable way?: Yes, clearly.
Does it provide unique data, unique conclusions about existing data, or a unique theoretical or experimental approach?: Yes, the theoretical approach is sound.


Arguments for acceptance: It is a paper that provides theory, methodology and experiments regarding a very difficult and challenging task that add value to the community and makes progress in the area of the equivalence between NN and GPs.

Arguments against acceptance: I do not have.

Typos:

-> Define the channel concept in introduction.
-> Put in bold best results of the experiments.
-> Why not put "deep" in the title?
-> In the introduction, introduce formally a CNN. (brief)
-> Define the many channel limit.
-> Put a figure with the equivalences and with the contents of the paper explaining a bit.



After rebuttal:
=============

Authors have addressed many topics that not only I but rev 3 address and hence I score this paper with a 7 and recommend it for publication.

---

> ### Author Response · Authors · 2018-11-27
> **AnonReviewer1 Reply**
>
> Thank you for your detailed and encouraging review! We are glad you found our research interesting. Please find below our replies to your specific comments:
>
> ------------------------------------------------------------------------------------
> >>> -> Put in bold best results of the experiments.
>
> Thank you for the suggestion. Tables 1 and 2 are updated.
>
> ------------------------------------------------------------------------------------
> >>> -> Why not put "deep" in the title?
>
> Good suggestion, we have updated our title.
>
> ------------------------------------------------------------------------------------
> >>> -> Define the channel concept in introduction.
> >>> -> In the introduction, introduce formally a CNN. (brief)
>
> We have formally defined convolutional operation with convolution filters in Section 2.1 (preliminaries).
>
> ------------------------------------------------------------------------------------
> >>> -> Define the many channel limit.
>
> The revised introduction describes the many channel limit more concretely (point 1 under the contributions), also see the end of the new section 2.1 “Shapes and indexing”.
>
> ------------------------------------------------------------------------------------
> -> Put a figure with the equivalences and with the contents of the paper explaining a bit.
>
> Thank you for the suggestion. We have added Figure 4 to better explain the notation and different concepts used in the paper.

---

> ### Author Response · Authors · 2018-12-05
> **AnonReviewer1 message**
>
> Dear Reviewer,
>
> Thank you again for your thoughtful reading and review. We believe you lowered your score due to the justified technical concerns raised by Reviewer 3. However, we have now updated the paper to address those specific issues. Reviewer 3 is satisfied with our response, and has raised their score by 4 points. Would you now be willing to restore your original score, since we have addressed all the open technical concerns, and both other reviewers are now voting for acceptance?
>
> Thank you very much for your consideration!

---

### Official Review · AnonReviewer4 · 2018-12-06
**REVIEW OF DEEP BAYESIAN CONVOLUTIONAL NETWORKS WITH MANY CHANNELS ARE GAUSSIAN PROCESSES**

**Rating:** 7
**Confidence:** 3

**Review:**

The paper establishes a connection between  infinite channel Bayesian convolutional neural network and Gaussian processes. The authors prove that taking the number of channels in a Bayesian CNN to infinite leads to a GP with a specific Kernel (GP-CNN) and provide a Monte Carlo approach to evaluate the kernels when it is intractable. They show that without pooling the kernel fails to maintain the equivariance property that is achievable with a CNN without pooling.  GP-CNN with pooling maintains the invariance property. They make extensive  experimental comparison with CNN, demonstrating that as the number of channels become large, CNN achieve performance close to a GP-CNN. A discussion on reasons for best CNN to give a performance better than GP-CNN (especially with pooling), and a experimental comparison with finite width Bayesian CNN would have made the paper more concrete.  The paper has both strong theoretical and experimental contribution, and is also very relevant to the ICLR conference.

Quality

The paper provides a theoretical connection between Bayesian CNN with infinite wide channels and Gaussian processes with a recursive kernel (GP-CNN). The derivations and arguments seem correct. The experiments are conducted comparing the performance of SGD trained CNN with  GP-CNN, and other models on mainly on CIFAR-10 data set.
However, some discussion and clarity on the following points will be  useful to improve the paper.

- (Page 5) on convergence of K^l :  From Equations (3) and (4), it can be seen that  K^l converges to  C(K^{l-1}), with C(K^{l-1}) defined slightly different from the paper, in that the expectation over z  is taken w.r.t z~ N(0;A(K)) instead of z ~ N(0; K). Is this equivalent to the expressions (7) and (8) described in the paper for a non-linear function \phi ?
- Experimental comparison with Bayesian CNN, demonstrating the effect of increasing the number of channels.
- (Page 7) GP-CNN with pooling :  Paper proposes subsampling one particular pixel to improve computational efficiency. Has some experiments been performed to evaluate the performance of this approach ? How accurate is this approach ?
- Discussion on the positive semi-definiteness of the recursive GP-CNN kernel
- More explanations on why the best SGD-trained CNN gives a better performance than GP-CNN, especially with pooling. Does the Monte-Carlo approximation of GP-CNN kernel  computation could impact this performance? I suppose hyper-parameters of the GP-CNN kernel are not learnt from the data, could this result in a lower accuracy  ?
- Discussion on learning the hyper-parameters of the GP-CNN kernel and its impact on the performance of the model.
- Demonstrate  through some sample figures that GP-CNN with pooling achieves invariance while GP-CNN with out pooling fail to capture it.
- Is the best result on CIFAR-10  achieved using the proposed method ? See Deep convolutional Gaussian processes by Kenneth Blomqvist, Samuel Kaski, Markus Heinonen
- Include the results with CNN-GP both with pooling and without pooling in Table 1 and Table 2.
- Provide the results of best SGD trained CNN against CNN-GP, both with pooling, as in Figure 3.c. Is the same trend observed in this case also ?
- Experimental comparison and results on other Image datasets, specifically MNIST. Does the same observations hold on MNIST too ?

Clarity

The paper is relatively well written and clearly provides main ideas leading to the results. However, notations could have made more succinct, and figures could have been more legible( Axis labels are missing for some figures in Figure 3, and provide legends wherever possible). The is also an ambiguity in what CNN-GP refers to, with pooling to without pooling.
- The term CNN-GP is overloaded in many places in the experimental section. I guess in Table 1, its CNN-GP without pooling, while in Table 2, its CNN-GP with pooling. Kindly make the distinction clear in the nomenclature itself, by calling one of them by a different name. Its also not clear when they mention SGD trained CNN, if it is with pooling or without pooling.
- What is the difference between the top and bottom pair of figures in Figure 3 (b). Why is the GP performance different in top and bottom cases.?
- What does 10, 100, 1000 correspond to in Figure 3 ? Please explain it in caption.

Originality

Previous works of  Lee and G. Matthews (2018) had shown the equivalence between Deep Neural Networks and GPs. This paper has extended it  to deep convolutional  neural network setting, but is interesting in its own way. The have come up with an equivalent kernel corresponding to infinite wide Bayesian convolution neural network and provided a monte-carlo approach to compute it. Along with the theoretical contribution, they have also provided extensive experimental comparison.

Significance

The paper has made significant contributions connecting the Bayesian convolutional neural networks with Gaussian processes, in deriving the equivalent kernel for GPs, and in demonstrating the performance of the proposed approach on Image datasets

---

> ### Author Response · Authors · 2018-12-15
> **2/2 AnonReviewer4 Reply**
>
> ------------------------------------------------------------------------------------
> >>> - Demonstrate  through some sample figures that GP-CNN with pooling achieves invariance while GP-CNN with out pooling fail to capture it.
>
> Thank you for the suggestion, we are working on and are planning to include covariance visualizations on toy data in the next revision.
>
> ------------------------------------------------------------------------------------
> >>> - Is the best result on CIFAR-10  achieved using the proposed method ? See Deep convolutional Gaussian processes by Kenneth Blomqvist, Samuel Kaski, Markus Heinonen
>
> We cite them in related work and point out that their model (as are other deep GPs) is not a GP but a more complex and expressive probabilistic model. To the best of our knowledge, our result is SOTA on CIFAR10 for GPs without trainable kernels.
>
> ------------------------------------------------------------------------------------
> >>> - Include the results with CNN-GP both with pooling and without pooling in Table 1 and Table 2.
>
> As mentioned above, we do not have these results since running a CNN-GP with pooling is prohibitively expensive on such large datasets, especially for a large-scale grid search as was done for Table 2 (see A.7.5)
>
> ------------------------------------------------------------------------------------
> >>> - Provide the results of best SGD trained CNN against CNN-GP, both with pooling, as in Figure 3.c. Is the same trend observed in this case also ?
>
> Please see our comments above - we believe evaluating CNN-GP with pooling on complete datasets for such a large grid to lie beyond the scope of this work.
>
> ------------------------------------------------------------------------------------
> >>> - Experimental comparison and results on other Image datasets, specifically MNIST. Does the same observations hold on MNIST too ?
>
> We have only run our large-scale grid searches on CIFAR10 since this is the dataset that benefits from the convolutional architecture the most (among considered) and allows to confidently distinguish the performance of different models (see e.g. Table 1). We expect the general trends to generalize to other image datasets.
>
> ------------------------------------------------------------------------------------
> >>> [...] ( Axis labels are missing for some figures in Figure 3,
>
> Since all plots share common axes and ranges, we only displayed the title of the x-axis (“#Channels”) at the bottom once, and the title of the y-axis (“Validation accuracy”) in the center to avoid clutter. We can fix it in the next revision.
>
> ------------------------------------------------------------------------------------
> >>> and provide legends wherever possible).
>
> Please note that it is not practical to have a complete legend in Figures 3.a and 3.c due to each point representing one of many different hyper-paramater settings (if you refer to other Figures, please let us know which ones).
>
> ------------------------------------------------------------------------------------
> >>> The is also an ambiguity in what CNN-GP refers to, with pooling to without pooling.
> - The term CNN-GP is overloaded in many places in the experimental section. I guess in Table 1, its CNN-GP without pooling, while in Table 2, its CNN-GP with pooling. Kindly make the distinction clear in the nomenclature itself, by calling one of them by a different name. Its also not clear when they mention SGD trained CNN, if it is with pooling or without pooling.
>
> Throughout the work, pooling is only used if explicitly mentioned (e.g. “CNN-GP w/ pooling”, “CNN w/ pooling” etc). Otherwise CNN(-GP) is without pooling. We will make it more explicit in the next revision.
>
> ------------------------------------------------------------------------------------
> >>> - What is the difference between the top and bottom pair of figures in Figure 3 (b). Why is the GP performance different in top and bottom cases.?
>
> As per text labels to the left of the table, top are LCNs (locally-connected networks, CNNs without weight sharing), while bottom are regular CNNs. If pooling is present, LCNs and CNNs result in different respective GPs, hence different performance (see discussion in section 5.1). We will make the text labels more noticeable in the next revision.
>
> ------------------------------------------------------------------------------------
> >>> - What does 10, 100, 1000 correspond to in Figure 3 ? Please explain it in caption.
>
> The numbers are depth, as indicated by the label in top-left, and the caption mentions it as well. We will make it more explicit in the next revision.

---

> ### Author Response · Authors · 2018-12-15
> **1/2 AnonReviewer4 Reply**
>
> Thank you for your very comprehensive and encouraging review! Please find our replies to your specific comments below.
>
> ------------------------------------------------------------------------------------
> >>> A discussion on reasons for best CNN to give a performance better than GP-CNN (especially with pooling), and a experimental comparison with finite width Bayesian CNN would have made the paper more concrete.
>
> Please note that we have only observed this difference in performance for CNN-GP without pooling, which we explain in discussion (sections 5.1 and 5.3). We don’t make any claims regarding comparisons CNN-GP with pooling and the respective CNN. We emphasize that CNN-GP throughout the paper refers to a no-pooling architecture, and in the rare cases where we evaluate it with pooling we say so explicitly (e.g. “MC-CNN-GP with pooling”, “Global average pooling”). We will make this more clear in the next revision.
>
> ------------------------------------------------------------------------------------
> >>> [...]- (Page 5) on convergence of K^l :  From Equations (3) and (4), it can be seen that  K^l converges to  C(K^{l-1}), with C(K^{l-1}) defined slightly different from the paper, in that the expectation over z  is taken w.r.t z~ N(0;A(K)) instead of z ~ N(0; K). Is this equivalent to the expressions (7) and (8) described in the paper for a non-linear function \phi ?
>
> You are correct, it is exactly equivalent (\circ represents composition). We will make this step more clear in the next revision.
>
> ------------------------------------------------------------------------------------
> >>> - Experimental comparison with Bayesian CNN, demonstrating the effect of increasing the number of channels.
>
> Thank you for the suggestion, we agree such experiments would be highly relevant. However, the computational requirements of training Bayesian CNN prohibits us from performing experiments in the many channel setting, which is on the other hand tractable with SGD. Additionally, the connection between SGD and Bayesian inference is an area of active and sometimes contradictory research in the ML community currently, and we believe our experimental results comparing the NN-GP to SGD trained network will therefore be of significant interest.
>
> ------------------------------------------------------------------------------------
> >>> - (Page 7) GP-CNN with pooling :  Paper proposes subsampling one particular pixel to improve computational efficiency. Has some experiments been performed to evaluate the performance of this approach ? How accurate is this approach ?
>
> Please note that this approach (section 3.2.2) is only related to pooling (section 3.2.1) in that both approaches are particular cases of projection (section 3.2). Subsampling the center pixel is instead more similar to vectorization (section 3.1) in terms of both performance and compute, and is compared to other methods in Figure 1 (blue curve).
>
> ------------------------------------------------------------------------------------
> >>> - Discussion on the positive semi-definiteness of the recursive GP-CNN kernel
>
> Thank you for the suggestion, we will include explicit derivation of this property in the next revision.
>
> ------------------------------------------------------------------------------------
> >>> - More explanations on why the best SGD-trained CNN gives a better performance than GP-CNN, especially with pooling. Does the Monte-Carlo approximation of GP-CNN kernel  computation could impact this performance? I suppose hyper-parameters of the GP-CNN kernel are not learnt from the data, could this result in a lower accuracy  ?
>
> Please see our comment above - we did not evaluate the CNN-GP with pooling on the whole CIFAR10 dataset, since it was prohibitively expensive. The explanation for the difference in performance between the best CNN and best CNN-GP (both without pooling) is given in sections 5.1 and 5.3. Whenever we evaluate a CNN-GP with pooling, it is stated explicitly.
>
> ------------------------------------------------------------------------------------
> >>> - Discussion on learning the hyper-parameters of the GP-CNN kernel and its impact on the performance of the model.
>
> Thank you for the suggestion. This work was primarily focused on comparing CNNs and their respective CNN-GPs, hence we only considered CNN-GP parameters that follow directly from the respective CNN architecture, and are learned only via a grid search (non-linearity, depth, weight and bias variance, pooling). It would indeed be very interesting in future work to do gradient descent of the GP/NN likelihood w.r.t. weight and bias variance, as well as parameterizing the nonlinearity in a differentiable way, and comparing these models.
>
> Relatedly, we would like to also draw your attention to the discussion in Appendix A.2, where we link the hyperparameters of the CNN-GP kernel to previous work in deep information propagation.

---

### Meta-Review · Area_Chair1 · 2018-12-15
**Interesting work taking recent advances one step further**

**Confidence:** 5
**Recommendation:** Accept (Poster)

**Metareview:**

There has been a recent focus on proving the convergence of Bayesian fully connected networks to GPs. This work takes these ideas one step further, by proving the equivalence in the convolutional case.

All reviewers and the AC are in agreement that this is interesting and impactful work. The nature of the topic is such that experimental evaluations and theoretical proofs are difficult to carry out in a convincing manner, however the authors have done a good job at it, especially after carefully taking into account the reviewers’ comments.